# Loop current fluctuations
# and quantum critical transport

Zhengyan Darius Shi[1], Dominic V. Else[2,*], Hart Goldman[1,*], and T. Senthil[1]

[1]*Department of Physics, Massachusetts Institute of Technology, Cambridge, MA 02139, USA*

[2]*Department of Physics, Harvard University, Cambridge, MA 02138, USA*

August 10, 2022

## Abstract

We study electrical transport at quantum critical points (QCPs) associated with loop current ordering in a metal, focusing specifically on models of the "Hertz-Millis" type. At the infrared (IR) fixed point and in the absence of disorder, the simplest such models have infinite DC conductivity and zero incoherent conductivity at nonzero frequencies. However, we find that a particular deformation, involving $N$ species of bosons and fermions with random couplings in flavor space, admits a finite incoherent, frequency-dependent conductivity at the IR fixed point, $\sigma(\omega > 0) \sim \omega^{-2/z}$, where $z$ is the boson dynamical exponent. Leveraging the non-perturbative structure of quantum anomalies, we develop a powerful calculational method for transport. The resulting "anomaly-assisted large $N$ expansion" allows us to extract the conductivity systematically. Although our results imply that such random-flavor models are problematic as a description of the physical $N = 1$ system, they serve to illustrate some general conditions for quantum critical transport as well as the anomaly-assisted calculational methods. In addition, we revisit an old result that irrelevant operators generate a frequency-dependent conductivity, $\sigma(\omega > 0) \sim \omega^{-2(z-2)/z}$, in problems of this kind. We show explicitly, within the scope of the original calculation, that this result does not hold for any order parameter.

---

\* These authors contributed equally to the development of this work.

# 1 Introduction

In the last few decades a growing number of metallic systems have been studied which show striking departures from Fermi liquid physics. Examples include the normal metallic state of the cuprate and iron-based high temperature superconductors [1–4], various heavy fermion metals near zero temperature magnetic phase transitions [5, 6], and metallic states in various two dimensional Moiré heterostructures [7–10]. A common striking property is an electrical resistivity that varies linearly in temperature, down to scales much lower than the natural energy scales of the system. Many such non-Fermi liquids (NFLs) also appear to display a scale-invariant form of the optical conductivity at small frequencies and temperatures [11–13],

$$\sigma(\omega, T) = \frac{1}{T^\alpha} \Sigma(\omega/T) , \qquad (1.1)$$

for some scaling function $\Sigma(x)$ and exponent $\alpha$. When $\alpha = 1$, the famous $T$-linear DC resistivity is recovered. Such a scale-invariant form of the conductivity is often viewed as a signature of quantum criticality.

Nevertheless, finding models of NFLs that actually display scale-invariant transport has proven to be a challenging problem. One reason is that in general, due to emergent conserved quantities such as spatial momentum, one expects that in the IR scaling limit the real part of the conductivity is the sum of a "coherent" Drude peak and an "incoherent" scale-invariant contribution,

$$\mathrm{Re}\,\sigma(\omega) = \mathcal{D}\,\delta(\omega) + \frac{1}{T^\alpha} \Sigma(\omega/T) . \qquad (1.2)$$

The first question, then, is to understand how to suppress the Drude weight. This is particularly challenging if one exclusively considers clean systems. A focus on the clean limit is well motivated phenomenologically: For example, in the cuprate strange metal phase, the slope of the famous linear resistivity (per $Cu - O$ layer) is roughly the same across different cuprate materials, each with wildly different levels of disorder [14] (see also Ref. [15] for additional discussion). Recently, two of us [15, 16] addressed this question by showing that a vanishing Drude weight in a clean, compressible system is possible only if there is a diverging susceptibility for an observable that is odd under time reversal and inversion symmetries, is at zero crystal momentum, and transforms as a vector under lattice rotations. This mechanism was dubbed "critical drag" and we will broadly refer to such observables as "loop current" order parameters [17, 18].

However, the situation is actually even more acute than what we described above. As we showed using general arguments in Ref. [19], the most commonly studied "Hertz-Millis" mod-

els of non-Fermi liquid metals [20–35] *only* have the delta function conductivity in Eq. (1.2) in the IR limit. These models are constructed by coupling a Fermi surface to a bosonic order parameter at zero momentum near a quantum critical point, or to a fluctuating gauge field (in the latter case, the conductivity vanishes altogether). Hence, we are led to the guiding questions of this work: *What clean models of compressible non-Fermi liquid metals display any incoherent conductivity at all in the scaling limit? And can we explicitly calculate the incoherent conductivity within some controlled theory of the NFL state?* Indeed, calculations of transport are notoriously difficult compared to, say, thermodynamic quantities. Thus, it is important to develop methods to systematically calculate transport given a controlled access to the infrared (IR) NFL fixed point.

The mechanism of critical drag, introduced in Refs. [15, 16] and refined for Hertz-Millis models in Ref. [19], suggests that good starting points are models involving loop current order parameters. Such order parameters have been subject to much discussion in the cuprate materials, with many reports of static loop currents in the underdoped regime, along with some controversies [36–46]. Here we will not wade into the experimental situation, except to the extent that the possible observation of loop currents may be viewed as further motivation for studying related models. Our main interest will be in the loop current ordering transition in a metallic system, which is a quantum phase transition from a symmetry preserving Fermi liquid metal to a different Fermi liquid metal with static loop current order. This phase transition separates two electronic Fermi liquid metals, but the critical point itself will be a non-Fermi liquid metal with a sharp "critical Fermi surface" but no Landau quasiparticle.

In a provocative body of work dedicated to this phase transition, Varma and collaborators [47–57] have explored a description in terms of a dissipative quantum XY model. The loop current order parameter is described by the XY field (with some anisotropy that is argued to be irrelevant), and the dissipation comes from coupling to the fermions. These authors have emphasized that many experimental features of strange metals are captured by this dissipative quantum XY model. However, there are important open theoretical questions regarding the form of the assumed dissipation in these models. Thus, it is appropriate to consider other more conventional formulations of the quantum critical point associated with the onset of loop current order. The results reviewed in Ref. [54] will thus be in a different universality class from the one studied in this paper, although they may describe the same phase transition (field theoretic examples where one phase transition admits multiple universality classes are explored in Ref. [58]).

We focus this work on theories of metallic loop current criticality formulated within the *conventional* Hertz-Millis framework, where the bosonic loop current order parameter is cou-

pled to the Fermi surface via a Yukawa coupling. We begin with a general discussion of these models and their physical properties. Most importantly, we demonstrate that strict critical drag, where the Drude weight vanishes due to critical fluctuations, can occur only through fine tuning of the Fermi velocity and/or the momentum-dependence of the boson-fermion coupling around the Fermi surface. This conclusion was already reached as a special case in Ref. [19], which focused on general classes of order parameters. Despite an apparent contrast with the earlier conclusions of Refs. [15, 16], we show that this result is still consistent with the divergence of the order parameter susceptibility. The essential reason for this is that the theory of a Fermi surface coupled to a loop current order parameter actually has infinitely many emergent conserved quantities: The charge at each point on the Fermi surface is conserved in the low energy limit. The diverging susceptibility of the order parameter is only sufficient to suppress the contribution of the Drude weight of finitely many linear combinations of these quantities. Thus, in the absence of fine-tuning, the weight of the delta function in the frequency dependent conductivity, $\sigma(\omega)$, will not go to zero, although in general it will be reduced from its free fermion value by critical fluctuations. Moreover, as mentioned above, Ref. [19] demonstrated that generic theories of this type have *vanishing* incoherent conductivity in the scaling limit; hence it appears that the scale invariant conductivity, Eq. (1.1) is not within reach in this class of models.

The result of Ref. [19] relied crucially on a general non-perturbative property–known as an *anomaly*–of the IR fixed point that the standard Hertz-Millis theory is believed to flow to. In the quest to find models with non-trivial incoherent conductivity, we should then modify the anomaly structure so as to evade the restrictions of Ref. [19]. Here we make a first step towards this goal by finding a deformation of the Hertz-Millis model that *does* exhibit a modified anomaly structure and an associated incoherent conductivity at the IR fixed point. In this deformation, one introduces $N$ flavors of both the fermions, $\psi_I$, and the critical boson, $\phi_I$, with a mutual coupling that is random in flavor space,

$$S_{\text{int}} = \int_{k,q} f_a(\boldsymbol{k}) \frac{g_{IJK}}{N} \psi_I^\dagger(k+q/2) \, \psi_J(k-q/2) \, \phi_K^a(q) \,, \tag{1.3}$$

$$\overline{g_{IJK}^* \, g_{I'J'K'}} = g^2 \, \delta_{II'} \, \delta_{JJ'} \, \delta_{KK'} \,, \qquad \overline{g_{IJK}} = 0 \,. \tag{1.4}$$

Here $f_a(\boldsymbol{k})$ is a form factor specifying the order parameter symmetry: the coupling for a loop current order parameter changes sign across the Fermi surface, $f_a(-\boldsymbol{k}) = -f_a(\boldsymbol{k})$. We also use the notation, $\int_p \equiv \int \frac{d^2\boldsymbol{p} \, dp_0}{(2\pi)^3}$. These models are translation invariant: The quenched randomness is in flavor space, not real space. In the $N \to \infty$ limit, these models have been argued to be self averaging and yield exactly solvable non-Fermi liquid fixed points [59–61]. Although we will also show that the large-$N$ model has a number of problematic features

that do not allow viewing it as a controlled description of the $N = 1$ theory, it serves our purpose of finding models that answer the guiding questions posed above.

We show how to explicitly compute the incoherent conductivity in the random-flavor large $N$ model. To that end, we introduce a new calculational technique that leverages the anomaly structure of the model. We dub this method *the anomaly-assisted large $N$ expansion*, and we expect that it will be useful more broadly in other controlled approximations for accessing NFL fixed points. Because this approach works at large but finite $N$, we are able to study the physical order of limits where the low frequency (IR) limit is taken prior to the large $N$ limit. Indeed, we demonstrate that these limits *do not commute* due to the presence of slow modes whose relaxation rates vanish as $N \to \infty$. We therefore focus on computing the conductivity using the memory matrix approach, which naturally handles these slow modes. In our anomaly-assisted large $N$ expansion, this calculation organizes itself into a perturbative expansion in the relaxation rates of the slow modes, which we find go like $1/N$. The existence of an anomaly in turn provides non-trivial constraints on the susceptibilities, which greatly simplify the calculation.

Our final result for the conductivity at $T = 0$ is of the form,

$$
\begin{aligned}
\sigma(\omega) &= N \frac{i}{\omega} \left[ \frac{\mathcal{D}_0}{\pi} - \frac{\delta\mathcal{D}}{\pi \left[ 1 - i\pi N \mathcal{C}_z \, \omega^{1-2/z} / \delta\mathcal{D} \right]} + \dots \right] \\
&= N \frac{i}{\omega} \frac{\mathcal{D}_0 - \delta\mathcal{D}}{\pi} + N^2 \mathcal{C}_z \, \omega^{-2/z} + \dots, \text{ for } \omega^{1-2/z} \ll \frac{1}{N} \ll 1,
\end{aligned}
\tag{1.5}
$$

where $z$ is the boson dynamical exponent, $\mathcal{D}_0$ is the Drude weight of a free Fermi gas, and $\delta\mathcal{D}$ and $\mathcal{C}_z$ are constants that vanish for inversion-even order parameters (such as Ising-nematic) but do not vanish for inversion-odd order parameters. For the commonly considered case of a $z = 3$ boson, the incoherent conductivity scales as $\sigma(\omega > 0) \sim \omega^{-2/3}$. Notice that in the naïve order of limits where $N \to \infty$ before $\omega \to 0$, the conductivity in Eq. (1.5) reduces to a free Fermi gas Drude weight, in agreement with [62, 63], along with a $\mathcal{O}(1/N)$ correction with a different frequency dependence. Therefore, the Drude weight reduction $\delta\mathcal{D}$ and the nontrivial incoherent conductivity in Eq. (1.5) are non-perturbative effects (in $1/N$) that are invisible to the naïve large $N$ limit. Moreover, the structure of Eq. (1.5) indicates that taking the low frequency limit first leads to an enhancement of the incoherent conductivity by a power of $N$, reminiscent of the phenomenon discovered in the ordinary (random phase approximation, or RPA) large $N$ expansion which renders it uncontrolled [28]. Finally, we find that at finite temperature, the conductivity becomes dominated by thermal rather than quantum fluctuations, and we are unable to verify that the incoherent conductivity found above fits into a scale-invariant form at the lowest temperatures, even though Eq. (1.5) is a

property of the IR fixed point of the model.

Physically, the existence of an incoherent conductivity indicates that the random-flavor large $N$ model is *qualitatively distinct* from the $N = 1$ model studied at length in Ref. [19]. When $N$ is small but not 1, the functional form of the incoherent conductivity is inaccessible using our methods, though we still expect a nontrivial answer because the theory for any $N > 1$ does not satisfy the same anomaly constraints as the $N = 1$ model.

From an RG perspective, the theory for $N > 1$ describes a multicritical point with $N^2$ relevant operators allowed by symmetry, while the $N = 1$ model is an ordinary critical point. For finite $N$, without fine tuning each disorder realization, the model then generically flows in the IR to a *different* critical point with $N$ fermions and a single gapless boson, which is the traditionally studied large $N$ theory. Despite its resistance to controlled study, we established in Ref. [19] that this latter theory is subject to the same constraints on optical transport as the physical $N = 1$ theory. While there may have been some hope that these fixed points merge as $N$ is turned down to unity, our transport results demonstrate that this cannot be the case.

Given our results for the random-flavor model and the conclusions of our earlier non-perturbative work at $N = 1$ in Ref. [19], we were also driven to bring fresh eyes to some of the more traditional approaches to calculating optical conductivity in this class of non-Fermi liquid metals. In particular, a classic and widely-quoted paper on this subject by Kim, Furusaki, Wen, and Lee [64] obtains an incoherent conductivity, $\sigma(\omega > 0) \sim \omega^{2(2-z)/z}$, where $z$ is the dynamical boson critical exponent. This calculation is based on an RPA-style expansion up to 2-loop order, and it relies on the presence of irrelevant operators, such as the curvature of the dispersion and the momentum-dependence of the boson-fermion coupling. This result should therefore be viewed as a correction to scaling, rather than a genuine quantum critical conductivity. Interestingly, by carefully performing the same type of calculation for generic order parameter symmetry, we show that the coefficient of $\omega^{2(2-z)/z}$ actually *vanishes* for arbitrary order parameters due to surprising diagrammatic cancellations! To our knowledge, this is the first time these coefficients have been determined for generic order parameter symmetry and Fermi surface geometry within the RPA expansion. We note that following the initial appearance of this work, [63] was updated to reflect similar calculations that were performed independently (though they did not demonstrate the cancellation for general order parameter symmetry and Fermi surface geometry).

We proceed as follows. In Section 2, we review the Hertz-Millis framework for the onset of broken symmetries in metals, focusing on unique features of the loop current critical point that motivate us to consider its transport properties. In Section 3, we study transport in

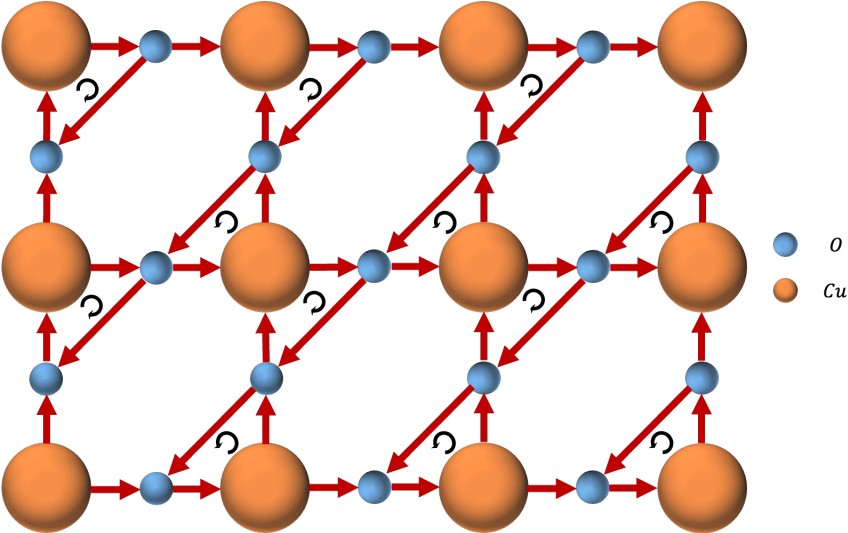

**Figure 1:** The standard Varma loop ordering pattern in a single CuO plane of cuprate materials. The red and black arrows indicate the direction of current flow.

the simplest Hertz-Millis model for loop current criticality and re-derive some of the results of our earlier work, Ref. [19], from a different perspective. We also summarize the effects of irrelevant operators by revisiting the calculation in Ref. [64]. In Section 4, we compute the optical conductivity of the random-flavor model using our anomaly-assisted large $N$ approach. We conclude in Section 5.

# 2 Loop current criticality in a metal

## 2.1 Hertz-Millis paradigm for onset of loop order

The loop current ordered state is characterized by a zero-momentum order parameter that transforms as a vector under lattice rotations and is odd under time reversal and inversion. See Figure 1 for the standard ordering pattern proposed in the context of cuprates [17, 18]. In this work, we are interested in the quantum critical point (QCP) describing the onset of loop current order in a metallic system. We first briefly review the standard Hertz-Millis paradigm [20, 21] for addressing phase transitions of this kind and highlight some of its general properties. Then we compare loop current criticality to criticality associated with the onset of other broken symmetries and note a number of similarities and differences. This discussion will set the stage for our analysis of transport later on.

The basic philosophy of the Hertz-Millis paradigm is that universal properties of the phase

transition can be captured by an effective model that couples the electronic quasiparticles near the Fermi surface to long wavelength and low energy fluctuations of the loop current order parameter field. Introducing the fermion field, $\psi$, and the loop current order parameter field, $\vec{\phi} = (\phi_x, \phi_y)$, we can write the Euclidean action of the Hertz-Millis model in two spatial dimensions as

$$S = S_\psi + S_{\text{int}} + S_\phi \tag{2.1}$$

$$S_\psi = \int_{\omega,\boldsymbol{k}} \psi^\dagger (i\omega - \epsilon(\boldsymbol{k}))\psi \,, \tag{2.2}$$

$$S_{\text{int}} = \int_{\Omega,\boldsymbol{q}} \int_{\omega,\boldsymbol{k}} \vec{g}(\boldsymbol{k}) \cdot \vec{\phi}(\boldsymbol{q}, \Omega)\, \psi^\dagger(\boldsymbol{k} + \boldsymbol{q}/2, \omega + \Omega)\, \psi(\boldsymbol{k} - \boldsymbol{q}/2, \omega) \,, \tag{2.3}$$

$$S_\phi = \frac{1}{2} \int_{\tau,\boldsymbol{x}} \left[ m_c^2 |\vec{\phi}|^2 + \lambda |\partial_\tau \vec{\phi}|^2 + J\left(\boldsymbol{\nabla}\phi_i\right) \cdot \left(\boldsymbol{\nabla}\phi^i\right) + \cdots \right] \,. \tag{2.4}$$

Here we have used the notation, $\int_{\omega,\boldsymbol{k}} \equiv \int \frac{d^2\boldsymbol{k}\, d\omega}{(2\pi)^3}$ and $\int_{\tau,\boldsymbol{x}} \equiv \int d\tau\, d^2\boldsymbol{x}$, which we will use throughout the manuscript. We will generally use boldface to denote spatial vectors except for the cases of $\vec{g}$ and $\vec{\phi}$, which here are also spatial vectors. Later on, we will consider more general kinds of inversion-odd order parameters where $\vec{g}, \vec{\phi}$ are allowed to have any number of components.

The first term, $S_\psi$, describes the electronic degrees of freedom with some dispersion $\epsilon(\boldsymbol{k})$, while $S_\phi$ describes the order parameter fluctuations. The interaction term, $S_{\text{int}}$, describes the coupling of the order parameter with particle-hole pairs of the electronic fluid. Importantly, this coupling involves the form factor, $\vec{g}(\boldsymbol{k})$, which incorporates the symmetries of the loop current order parameter. It is a vector under lattice rotations and satisfies

$$\vec{g}(\boldsymbol{k}) = -\vec{g}(-\boldsymbol{k}) \,. \tag{2.5}$$

The Hertz-Millis model for the onset of other broken symmetries is described by a similar action but with a different form factor structure. For instance, in the popular case of Ising nematic criticality, the order parameter is a scalar, has a form factor that is even under $\boldsymbol{k} \to -\boldsymbol{k}$, and has a characteristic angular variation in $\boldsymbol{k}$-space.

The exact IR properties of the Hertz-Millis model are challenging to determine. But there is a large literature of approximate treatments that agree on the general structure of low energy singularities. The simplest (albeit uncontrolled) treatment of the model is based on a random phase approximation (RPA) and shows that the dynamics of the boson is determined by the Landau damping term, $\Pi(\omega \ll \boldsymbol{q}) \sim \frac{|\omega|}{|\boldsymbol{q}|}$, generated through the boson-fermion coupling. The fermions themselves acquire a self-energy, $\Sigma(\omega) \sim i|\omega|^{2/3} \operatorname{sgn}(\omega)$,

showing that this quantum critical point (QCP) is a non-Fermi liquid. Thermodynamic properties also deviate from the predictions of Fermi liquid theory: For example, the low-$T$ heat capacity $C_v \sim T^{2/3}$ at the QCP. These scaling properties signal a divergence of the effective mass of the Landau quasiparticles throughout the Fermi surface as we tune to the QCP. Nevertheless, the electronic compressibility – defined by the change of charge density as the chemical potential is varied while staying on the critical line – and spin susceptibility (if spin is included) stay finite at the QCP. Approaching from the Fermi liquid side, this may be understood in terms of diverging Landau parameters that compensate for the diverging effective mass in these susceptibilities.

Although the RPA is uncontrolled, the physical picture it paints is impressively preserved by a number of more sophisticated treatments of the model [30, 31, 33]. Such treatments typically involve a deformation of the model that allows for a perturbative expansion about a tractable limit, facilitating access to a controlled non-Fermi liquid fixed point. Unfortunately, each such existing approach has some unsatisfactory feature that complicates the extrapolation to the original (i.e undeformed) model. Nevertheless the study of these different controlled expansions has yielded much insight, and we will rely on these insights in this Section to draw some qualitative conclusions about the non-Fermi liquid QCP driven by critical loop current fluctuations.

First, an important insight that we will return to soon is that the universal critical properties can be described by a "patch" approach that discretizes the Fermi surface into a large number of small patches. Boson fluctuations with a momentum $\boldsymbol{q}$ primarily couple strongly to patches whose normal is perpendicular to $\boldsymbol{q}$. For simple Fermi surface shapes, this means that a pair of antipodal patches couple strongly to a single common boson field. The ultimate low energy properties are obtained by sewing together different pairs of antipodal patches.

The behavior of the form factor under $\boldsymbol{k} \to -\boldsymbol{k}$ is thus crucial, as it determines how the boson couples to the fermions in the two antipodal patches. In the Ising-nematic case (or other inversion-even order parameters), the boson couples with the same sign to the two antipodal patches. This is to be contrasted with the loop current case, where the boson couples with opposite signs (the latter also happens in the related problem of fermions coupled to a dynamical $U(1)$ gauge field). An immediate consequence of this sign difference is an enhancement (suppression) of $2k_F$–singularities at the loop current (Ising-nematic) QCP compared to the ordinary Fermi liquid. The relative sign of the form factor also determines the possible instability of the critical non-Fermi liquid metal to pairing. In the Ising-nematic case, within the controlled expansion of [30], the pairing instability is enhanced compared

to the ordinary Fermi liquid metal [65], and the putative metallic QCP is pre-empted by the occurence of a superconducting dome[1]. Thus, the QCP is ultimately avoided at the Ising-nematic transition. In the loop current case, by contrast, the sign difference in the boson-fermion coupling between antipodal patches renders the quantum critical metal *stable* to BCS couplings. Hence, the loop current QCP is a rare opportunity[2] to study a "naked" quantum critical metal without the complications of superconductivity. Such naked QCPs have been considered before in certain large $N$ models, but here it occurs in a physical setting without the need for appealing to such deformations.

The suppression of superconductivity at the loop current QCP is perhaps not surprising if we realize that the corresponding loop current ordered metal breaks time reversal and inversion symmetries. Thus the usual BCS instability is absent in the ordered state (as fermion states at $\boldsymbol{k}$ and $-\boldsymbol{k}$ are not degenerate) and it is natural that superconductivity is also absent at the QCP itself. On the disordered side of the transition, the IR fixed point is determined by the competition between attractive interactions induced by the BCS coupling and repulsive interactions mediated by the order parameter field. As a detailed RG analysis in Ref. [65] shows[3] that the symmetry-preserving Fermi liquid phase remains stable in a small neighborhood around the QCP and we end up with an interesting phase diagram shown in Figure 2.

In the rest of this paper, we focus on electrical transport at such naked QCPs. As anticipated in the introduction, distinct form factor symmetries lead to dramatically different $\omega, T$ scalings of the infrared conductivity. We make some general model-independent state-

---

[1]The question of pairing instability of the non-Fermi liquid fixed point was considered in Ref. [66] through a different expansion where the Fermi surface co-dimension is generalized to $d_c = 1.5 - \epsilon$. However for co-dimension $d_c > 1$ the Fermi surface density of states vanishes in the free theory; hence any possible weak coupling pairing instability is suppressed even if the low energy physics is controlled by a free fermion fixed point (as happens for $d_c \geq 1.5$), and this situation persists for small $\epsilon > 0$. Thus, as acknowledged in Ref. [66], addressing the pairing instability requires extrapolation to $\epsilon = 0.5$ where the co-dimension expansion loses control.

[2]In Ref. [65], it was suggested that at metallic QCPs driven by order parameter fluctuations - in contrast to those associated with phenomena like the Mott or Kondo breakdown transitions - superconductivity would generally be enhanced. The present observations show that this suggestion should be refined further to be generally valid. Specifically only fluctuations of inversion and time reversal symmetric order parameters enhance superconductivity at quantum critical points while other kinds of quantum criticality - both those beyond Hertz-Millis and those within but involving inversion and time-reversal odd order parameters - likely have suppressed superconductivity.

[3]Ref. [65] analysed the vicinity of a continuous Mott transition [67] between a Fermi liquid metal and a quantum spin liquid insulator with a spinon fermi surface. Though other details of the transition are different from the loop current criticality, the suppression of superconductivity relies on the same RG analysis.

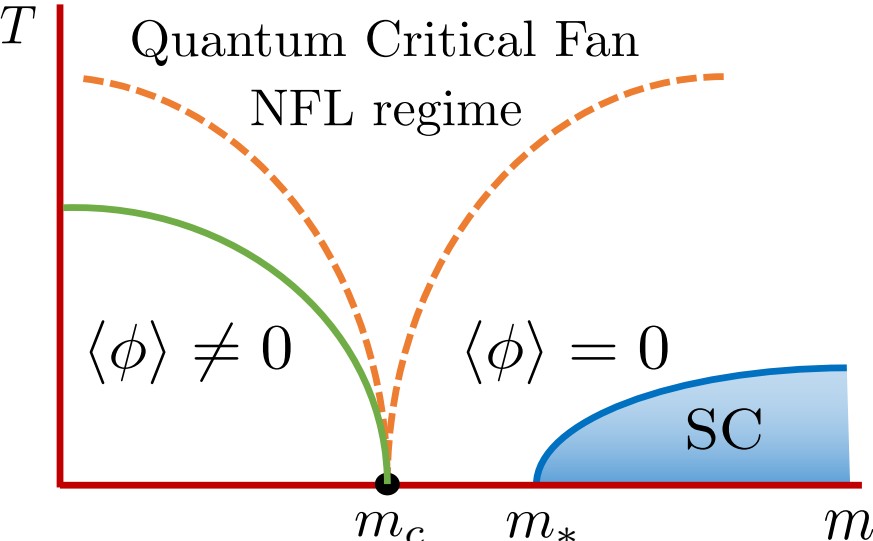

**Figure 2:** The phase diagram for a metal in the vicinity of a loop current ordering transition as a function of temperature $T$ and tuning parameter $m$. Unlike the Ising-nematic case, the loop current QCP at $m_c$ and its immediate neighborhood are stable to weak BCS attraction and a superconducting instability only develops deep in the disordered phase where $m > m_*$. The thermal phase transitions occur along the green and blue curves.

ments about transport at QCPs in the next subsection, which set the stage for calculations in concrete models that follow.

## 2.2 Drude weight suppression from critical drag

The starting point of our analysis is a discussion of general constraints on the Drude weight in a clean metallic QCP. Following the framework of Refs. [15, 16, 68], there is an obstacle to suppressing the Drude weight in a clean, compressible system. This is because compressibility is always associated with the existence of emergent conserved quantities (of the IR fixed point theory) that overlap with the current. Emergent conserved quantities are usually expected to lead to a nonzero Drude weight, as they prevent the current from relaxing.

For simplicity, we first consider the case where there is exactly one conserved quantity, $M$, that overlaps with the current. Then one can show from general thermodynamic arguments [69, 70] (see Ref. [71] for a review) that the Drude weight, $\mathcal{D}^{ij}$, should be given by

$$\frac{\mathcal{D}^{ij}}{\pi} = \frac{\chi_{J^i M} \chi_{J^j M}}{\chi_{MM}}.$$  (2.6)

Here we have defined the generalized susceptibility of two Hermitian operators, $A$ and $B$,

according to

$$\chi_{AB} = \chi_{BA} = \frac{d}{ds} \langle A \rangle_{sB} \bigg|_{s=0} = \int_0^\beta \langle AB(i\lambda) \rangle \, d\lambda - \beta \langle A \rangle \langle B \rangle, \qquad (2.7)$$

where the expectation values and imaginary time evolution are defined with respect to the appropriate thermal equilibrium ensemble, except for $\langle \cdot \rangle_{sB}$, where the ensemble is modified by adding a term $-sB$ to the Hamiltonian. The susceptibility $\chi_{JM}$ should be viewed as a measure of the overlap of $M$ with the current. If $\chi_{JM} \neq 0$, then the only way for the Drude weight to go to zero would be if $\chi_{MM}$ diverges. We should generally expect such a divergence at a QCP at which there are quantum fluctuations of an operator with the same symmetry transformation properties as $M$. At such a QCP, the Drude weight is thus driven to zero by the critical fluctuations, a mechanism that was referred to as "critical drag" [15, 16].

The situation becomes somewhat more complicated when there are $N_c$ conserved quantities overlapping with the current, $M_1, \cdots, M_{N_c}$. In that case, Eq. (2.6) generalizes to

$$\frac{\mathcal{D}^{ij}}{\pi} = \chi_{J^i M_a} \, \chi_{J^j M_b} \, (\chi^{-1})^{ab}, \qquad (2.8)$$

where we sum over the repeated indices and defined the susceptibility matrix $\chi_{ab} = \chi_{M_a M_b}$, which must be positive-definite by thermodynamic stability. For the Drude weight to vanish, we then need either: (a) $\chi^{-1} = 0$, meaning that *all* of the eigenvalues of $\chi$ diverge; or (b) at least one of the eigenvalues of $\chi$ diverges, and $\chi_{J^i M_a}$ satisfies an additional condition.

Now suppose that the null space of $\chi^{-1}$ has dimension $N_s$ at the QCP. If $N_c > N_s$, then $\chi^{-1}$ cannot be zero. Therefore, if there are too many conserved quantities, quantum criticality in the right symmetry channel is no longer a *sufficient* condition for the Drude weight to vanish, although it is still a *necessary* condition (otherwise $\chi$ would not have *any* diverging eigenvalues). In the concrete models of metallic quantum criticality that we study in this work, the infinite dimensional emergent symmetry group of the IR fixed point puts us precisely in the regime where $N_c \gg N_s$. As a result, we find that the Drude weight of the loop current QCP is nonzero but significantly reduced relative to QCPs associated with inversion-even order parameters.

# 3 Electrical transport in a model of loop current criticality

## 3.1 The mid-IR theory

We now move on to a detailed study of electrical transport in the Hertz-Millis model of fermions, $\psi$, interacting with a general order parameter field, $\vec{\phi} = (\phi_x, \phi_y)$, defined via

Eq. (2.1). Following Ref. [19], we focus primarily on a version of the model in which certain couplings are switched off. The expectation is that the resulting "mid-IR theory" will flow to the same IR fixed-point as the orginal microscopic model. The properties of this mid-IR theory will prove considerably easier to compute than of the original microscopic model.

We start by using the standard decomposition of the Fermi surface into patches, prohibiting any interaction terms that would scatter a fermion from one patch into another. Such inter-patch scattering terms are believed to be irrelevant in the renormalization group (RG) sense[4]. Summing over discrete patches labelled by $\theta = 2\pi/N_{\text{patch}}, \ldots, 2\pi$ results in the mid-IR action,

$$S = S_\phi + \sum_\theta S_{\text{patch}}(\theta) \,, \tag{3.1}$$

where $S_\phi$ includes all quadratic terms in the microscopic action. We discard additional self-interaction terms, i.e. the ... in Eq. (2.4), as they are believed to be irrelevant.

$$S_{\text{patch}}(\theta) = \int_{t,\boldsymbol{x}} \left[ \psi_\theta^\dagger \Big\{ i\partial_t + iv_F(\theta)[\boldsymbol{w}(\theta) \cdot \boldsymbol{\nabla}] + \kappa_{ij}(\theta)\partial_i\partial_j \Big\} \psi_\theta + g^a(\theta)\, \phi_a\, \psi_\theta^\dagger \psi_\theta \right]. \tag{3.2}$$

Here, for each patch $\theta$, we defined the Fermi velocity $\boldsymbol{v}_F(\theta) = v_F(\theta)\boldsymbol{w}(\theta)$, the order parameter form factor $g^a(\theta) \equiv g^a(\boldsymbol{k}_F(\theta))$. Note that we use $a, b, \ldots$ for the boson indices because although they are spatial indices in the problem of physical interest, one may also be interested in more general types of order parameters that are still odd under inversion and time reversal. The matrix $\kappa_{ij}(\theta)$ is the projection of the curvature tensor, $\partial_{k_i}\partial_{k_j}\epsilon(\boldsymbol{k})|_{\boldsymbol{k}_F(\theta)}$, onto the direction parallel to the Fermi surface, i.e. $w^i(\theta)\kappa_{ij}(\theta) = 0$. We drop the curvature of the fermion dispersion in the direction perpendicular to the Fermi surface since it is irrelevant compared to the term linear in $\boldsymbol{w}(\theta) \cdot \boldsymbol{\nabla}$, which has fewer derivatives. To avoid introducing redundant fermionic degrees of freedom, we impose a momentum cutoff, $\Lambda(\theta)$, for the fermion $\psi_\theta$ in the direction parallel to the Fermi surface, such that $\sum_\theta \Lambda(\theta)$ is the length in momentum space of the Fermi surface. At the end of each calculation, we take $\Lambda(\theta) \to 0$ and $N_{\text{patch}} \to \infty$.

In passing to the mid-IR theory, we have dropped interaction terms irrelevant under a tree-level scaling scheme within each patch that preserves the curvature of the Fermi surface. Despite this formal irrelevance, the effects of interactions like large-angle scattering

---

[4]Strictly speaking, inter-patch scattering processes should be included for the RG analysis of BCS couplings developed in Ref. [65]. However, as explained in Section 2.1, there is no superconducting instability at the loop current QCP when the bare BCS coupling is sufficiently small. Thus, we will turn off BCS couplings altogether and focus on the order parameter fluctuations. See Ref. [19] for a discussion of the effects of finite BCS couplings on the types of non-perturbative arguments used in this work.

on physical observables may not be small because the number of patches connected by the scattering processes diverge in the IR limit. Our working assumption is that contributions from these interactions are not more singular than the contributions from tree-level marginal terms that are already included in the mid-IR action. Certainly, a more careful treatment of large-angle scattering is an important future direction.

## 3.2  Introducing the anomaly: Why does the boson affect the conductivity?

We now describe how the critical boson fluctuations can affect transport and motivate some of the non-perturbative relations that will ultimately become the basis for the "anomaly-assisted large $N$ expansion" we will develop in Section 4. The results derived in the remainder of this Section were also obtained in Ref. [19], but in the present work we will arrive at them from a somewhat different perspective that can be more straightforwardly generalized to the random-flavor large $N$ theory studied in Section 4.

We begin by reviewing a naïve argument for why one might *not* have expected the critical fluctuations of the boson to be important for the conductivity. At low energies, the boson has only very long-wavelength fluctuations, so we do not expect that the boson can scatter electrons from one point on the Fermi surface to another distant point (i.e. there is only "forward scattering"). If we assume that the current operator can be expressed, as it can in Fermi liquid theory, in terms of the occupation numbers of fermions near the Fermi surface, it would follow that the boson cannot affect the current.

The problem with this argument is that the current operator in the model we are considering actually involves *both* the conserved fermion occupation numbers *and* the order parameter. This can be transparently seen in the microscopic model, Eq. Eq. (2.1), by minimally coupling to a background gauge electric vector potential, $\boldsymbol{A}(t)$, which couples to the current operator,

$$\boldsymbol{J}(t) = \frac{1}{V} \int_{\boldsymbol{k}} \boldsymbol{v}(\boldsymbol{k})\, \psi^\dagger(\boldsymbol{k})\, \psi(\boldsymbol{k}) + \frac{1}{V} \int_{\boldsymbol{k},\boldsymbol{q}} \partial_{\boldsymbol{k}} \left[ \vec{g}(\boldsymbol{k}) \cdot \vec{\phi}(\boldsymbol{q}) \right] \psi^\dagger(\boldsymbol{k}+\boldsymbol{q}/2)\, \psi(\boldsymbol{k}-\boldsymbol{q}/2), \qquad (3.3)$$

where $V$ is the total volume of the system, $\boldsymbol{v}(\boldsymbol{k}) = \partial_{\boldsymbol{k}}\epsilon(\boldsymbol{k})$, and we have suppressed the time dependence of the fields on the right-hand side. This expression is obtained by replacing $\vec{g}(\boldsymbol{k}) \rightarrow \vec{g}(\boldsymbol{k}+\boldsymbol{A})$, $\epsilon(\boldsymbol{k}) \rightarrow \epsilon(\boldsymbol{k}+\boldsymbol{A})$ and expanding to linear order in $\boldsymbol{A}$. Hence, we see that although forward scattering should not affect the "non-interacting" part of the current given by the first term of Eq. (3.3), the current operator is modified by a term that explicitly involves the boson. Therefore, in understanding the results of this paper, one should not think about transport purely in terms of the fermions "scattering" off of the bosons, as such

a picture fails to capture the effect of the second term in Eq. (3.3).

This result admits a simple interpretation in the mid-IR theory, Eqs. (3.1) – (3.2). Integrating the second term in Eq. (3.3) by parts and approximating $\partial_{\boldsymbol{k}}[\psi^\dagger(\boldsymbol{k}+\boldsymbol{q}/2)\psi(\boldsymbol{k}-\boldsymbol{q}/2)]$ by $\partial_{\boldsymbol{k}}\rho(k_F)$, the current density in patch $\theta$ is given by

$$j_\theta^i = v_F^i(\theta)\, n_\theta = v_F^i(\theta)\, \tilde{n}_\theta - \frac{\Lambda(\theta)}{(2\pi)^2}\, w^i(\theta)\, g^a(\theta)\, \phi_a\,, \tag{3.4}$$

where $\tilde{n}_\theta$ and $v_F^i(\theta) \equiv v_F(\theta)w^i(\theta)$ are the conserved charge density and the Fermi velocity vector in the patch. We have also defined what we will sometimes refer to as the "chiral" patch density, $n_\theta$, which unlike $\tilde{n}_\theta$ is invariant under independent U(1) gauge transformations in each patch, $\psi_\theta \to e^{i\alpha_\theta(w^i x_i)}\psi_\theta, g^a(\theta)\phi_a \to g^a(\theta)\phi_a + w^i(\theta)\partial_i\alpha_\theta$ [19].

Conservation of $\tilde{n}_\theta$ in the absence of external fields thus implies that conservation of $n_\theta$ is broken by an *anomaly* proportional to the "electric field" generated by $\phi$,

$$\partial_t \tilde{n}_\theta(t, \boldsymbol{q}=0) = 0\,, \tag{3.5}$$

$$\partial_t n_\theta(t, \boldsymbol{q}=0) = -\frac{\Lambda(\theta)}{(2\pi)^2}\frac{1}{v_F(\theta)}\, g^a(\theta)\, \partial_t\phi_a(t, \boldsymbol{q}=0)\,. \tag{3.6}$$

The details of this anomaly, a cousin of the 1+1-D chiral anomaly, are developed in Ref. [19]. Although the above equations are simply a way of expressing charge conservation on each Fermi surface patch, they define a family of exact operator relationships (up to contact terms) relating the fermion current operator and the boson field that will prove crucial in our analysis in this work[5]. From these exact relations, Eqs. (3.4) – (3.6), it immediately follows (after taking into account contact terms in the external field) that the conductivity can be related to the retarded Green's function of the boson, $G_{\phi_a\phi_b}^R(\boldsymbol{q}=0, \omega) \equiv D_{ab}(\omega)$, according to

$$\sigma^{ij}(\omega) = \frac{i}{\omega}\left[\frac{\mathcal{D}_0^{ij}}{\pi} - V^{ia}\, V^{jb}\, D_{ab}(\omega)\right]\,, \tag{3.7}$$

where we have defined constant matrices

$$\mathcal{D}_0^{ij} = \pi \operatorname{Tr}_\theta\left[v_F^i v_F^j\right]\,, \quad V^{ia} = \operatorname{Tr}_\theta\left[v_F^i g^a\right]\,, \tag{3.8}$$

---

[5]As a side note, we remark that in the mid-IR theory, the particular expressions for $n_\theta$ and $\tilde{n}_\theta$ in terms of the fields, $\psi_\theta$ and $\phi$, depend on a choice of UV regularization [19, 30]. For example, using the microscopic model in Eq. (2.1) as a short distance regulator leads to $\tilde{n}_\theta = \psi_\theta^\dagger\psi_\theta$. A different regularization that is often more convenient in perturbative calculations instead leads to $n_\theta = \psi_\theta^\dagger\psi_\theta$.

in terms of an inner product, $\mathrm{Tr}_\theta$, which denotes

$$\mathrm{Tr}_\theta [fg] = \sum_\theta \frac{\Lambda(\theta)}{(2\pi)^2 \, v_F(\theta)} \, f(\theta)g(\theta) \,. \tag{3.9}$$

Here $\mathcal{D}_0^{ij}$ is the free Fermi gas Drude weight, since we have turned off Landau parameters and consider only interactions with the boson. The vertex factor, $V^{ia}$, is roughly the overlap of the Fermi velocity with the form factor. Note finally that, in arriving at Eq. (3.7), we have ignored a contribution to the current operator from the dispersion of the fermions parallel to the Fermi surface. As argued in Ref. [19], this parallel current is expected not to contribute to the conductivity in the IR fixed-point theory due to emergent momentum conservation in each patch.

In the case of an order parameter that is even under inversion and time-reversal symmetry (such as Ising-nematic), the analogue of the second term in Eq. (3.7) would vanish for symmetry reasons. This means that neglecting the explicit contribution to the current from the boson in Eq. (3.4) would actually give the right answer. However, in this work we are of course interested in order parameters that are *odd* under inversion and time-reversal, in which case retaining the second term in Eq. (3.7) will be essential. For instance, in the problem of a Fermi surface coupled to a fluctuating $U(1)$ gauge field $a$, the second term cancels the free fermion Drude weight and leads to a vanishing conductivity, consistent with the fact that an external gauge field $A$ can be absorbed by a change of variables $a \to a + A$ in the path integral in the IR limit.

## 3.3 Drude weight from susceptibilities

The model we are considering, Eq. (2.1), has a very large emergent symmetry group. This is most transparently seen in the mid-IR theory, Eqs. (3.1) – (3.2), when the number of patches is taken to infinity. Therefore, in the IR limit there is roughly speaking, a conserved charge associated with each point on the Fermi surface. The more precise statement, alluded to above, is that, if we parameterize the Fermi surface by a continuous variable, $\theta$, then we can introduce the conserved charge distribution operators $\tilde{n}(\theta)$, such that the total charge is given by $\int \tilde{n}(\theta)d\theta$. Nevertheless, in the rest of this section, we will instead work with discrete patches, and continue to refer to $n_\theta$ as the conserved charge of a given patch. To recover the full emergent symmetry group, one should imagine taking $N_{\mathrm{patch}} \to \infty$, $\Lambda(\theta) \to |\partial_\theta \boldsymbol{k}_F(\theta)| \, d\theta$ and replacing sums over patches with integrals over $\theta$ at the end of the calculation.

Assuming that there are no other emergent conserved quantities, the general arguments of Section 2.2 allow us to determine the Drude weight of the system, provided that we know

the susceptibility function $\chi_{\tilde{n}_\theta, \tilde{n}_{\theta'}}$. We showed in Ref. [19] that this susceptibility can be fixed by general arguments to be

$$\chi_{\tilde{n}_\theta, \tilde{n}_{\theta'}} = \frac{\Lambda(\theta)}{(2\pi)^2 v_F(\theta)} \delta_{\theta\theta'} + \frac{1}{(2\pi)^4} \frac{\Lambda(\theta) g^i(\theta)}{v_F(\theta)} \frac{\Lambda(\theta') g_i(\theta')}{v_F(\theta')} \frac{1}{m^2 - m_c^2}, \qquad (3.10)$$

where $m^2 - m_c^2$ is the tuning of the boson off of criticality into the disordered phase. From the results of Ref. [19] one can also show that[6]

$$\chi_{J^i \tilde{n}_\theta} = \frac{1}{(2\pi)^2} w^i(\theta) \Lambda(\theta). \qquad (3.11)$$

Substituting Eq. (3.10) and Eq. (3.11) into the general formula Eq. (2.8), one obtains the Drude weight (see Appendix A),

$$\frac{\mathcal{D}^{ij}}{\pi} = \frac{\mathcal{D}_0^{ij}}{\pi} - V^{ia} V^{jb} (\Pi_0^{-1})_{ab}, \qquad (3.12)$$

where $\mathcal{D}_0^{ij}$, $V^{ij}$ are defined as before by Eqs. (3.8), and

$$\Pi_0^{ab} = \text{Tr}_\theta \left[ g^a g^b \right]. \qquad (3.13)$$

We are now prepared to determine the circumstances under which the Drude weight is actually zero. In Appendix A, we prove that $\mathcal{D}^{ij}$ given by Eq. (3.12) is zero if and only if there exists some matrix $S^i{}_j$ (independent of $\theta$) such that

$$v_F^i(\theta) = S^i{}_j \, g^j(\theta), \qquad (3.14)$$

for all $\theta$, where we now explicitly denote boson indices as spatial indices $i, j, \ldots$

To interpret this result, let us assume that the Fermi surface is generic, such that any vector in $\mathbb{R}^2$ can be obtained as a linear combination of the unit Fermi surface normals, $\boldsymbol{w}(\theta)$, at different values of $\theta$ (the only way for this not to occur would be if the Fermi surface is a perfectly straight line). It follows that $S$ is full-rank and therefore invertible. If we define a vector, $\boldsymbol{a}$, in terms of the boson fields according to

$$a_j = (S^{-1})^i{}_j \, \phi_i, \qquad (3.15)$$

then writing the boson-fermion coupling in terms of the $\boldsymbol{a}$ field, we find

$$g^i(\theta) \phi_i \, \psi_\theta^\dagger \psi_\theta = g^i(\theta) \, S^j{}_i a_j \, \psi_\theta^\dagger \psi_\theta^\dagger = [\boldsymbol{a} \cdot \boldsymbol{v}_F(\theta)] \, \psi_\theta^\dagger \psi_\theta \qquad (3.16)$$

---

[6] Actually, the $N_{\text{patch}} \to \infty$ version of this result holds not just in the Hertz-Millis theories we are studying but in any "ersatz Fermi liquid", i.e. any system with that has an emergent loop group with an anomaly. For example, see Ref. [15].

Thus, we see that $\boldsymbol{a}$ couples to the fermions exactly as a U(1) gauge field would. This makes sense physically: when the order parameter couples like an internal gauge field, any external probe gauge field can be "shifted away" by a change of variables. As a result, in the absence of boson self-interactions the optical conductivity is only affected by the frequency-dependent terms in the quadratic boson action, which will only correct the conductivity at higher orders in frequency. In the low energy limit as these terms are taken to zero (they are irrelevant compared to Landau damping), there is an emergent gauge symmetry: the boson fluctuations elimate the dependence of the partition function on the probe completely, and the conductivity vanishes.

One can now ask why the Drude weight is not *always* zero at the critical point. This reflects the considerations discussed at the end of Section 2.2 (see also Ref. [19]). Because there are *multiple* (in fact, infinitely many) conserved quantities associated with the Fermi surface that overlap with the current, it follows that even at a quantum critical point with critical fluctuations where the susceptibility of the order parameter diverges, the Drude weight is not inevitably zero.

## 3.4    Full optical conductivity: Coherent and incoherent contributions

In Ref. [19], we examined mid-IR effective theories of the type in Eqs. (3.1) – (3.2) using general arguments leveraging their emergent symmetry and anomaly structure. In that work, we obtained an exact result for the boson Green's function and hence, via Eq. (3.7), the conductivity, in the mid-IR theory. We found that the conductivity at the ultimate IR fixed point (assuming a second-order transition) is simply

$$\mathrm{Re}\,\sigma^{ij}(\omega) = \mathcal{D}^{ij}\,\delta(\omega), \tag{3.17}$$

with $\mathcal{D}^{ij}$ given by Eq. (3.12). Thus, the result of Ref. [19] agrees with the value of the Drude weight predicted from the general susceptibility arguments above.

Crucially, Eq. (3.17) shows that *there is no incoherent conductivity* generated by critical fluctuations. Thus, the IR fixed point of the model in Eqs. (3.1) – (3.2) does not fulfill our goal of generating a nontrivial incoherent conductivity at a metallic quantum critical point. This leaves two possible routes to incoherent conductivity in a clean theory in the Hertz-Millis framework:

1. Introducing irrelevant operators. Importantly, irrelevant operators will give a non-trivial $\omega$-dependent conductivity but cannot lead to $\omega/T$ scaling behavior at finite temperature.

2. Finding models in which the anomaly constraints leading to Eq. (3.17) are relaxed. In such models it should be possible to obtain nontrivial frequency scaling intrinsic to the fixed point without adding any irrelevant operators.

Most transport calculations in the literature that report a nontrivial frequency scaling in the conductivity require the presence of irrelevant operators and thus fall into the first category. Unfortunately, irrelevant operators cannot lead to incoherent conductivity at the IR fixed point, which is the central goal of this work. Therefore, in the next few sections, we choose to focus on the second route. As foreshadowed in Ref. [19], the random-flavor large $N$ model developed in Refs. [59–61] manages to evade some of the anomaly constraints valid in the $N = 1$ theory, and it is not possible to determine the conductivity exactly at large but finite $N$. Nevertheless, we will develop an "anomaly-assisted large N expansion" that allows us to access the most singular contributions to the conductivity at the IR fixed point.

Readers who are primarily interested in fixed point transport can now skip to Section 4. In the remainder of this section, we critically reexamine the corrections to scaling in $\sigma(\omega > 0)$ induced by irrelevant operators and find an important discrepancy with results in the literature. The starting point of our analysis is a prominent, albeit uncontrolled, calculation of these effects done by Kim, Furusaki, Wen, and Lee in Ref. [64]. There, they modify the model in Section 3.1 to include $N_f$ fermion flavors coupled to a $U(1)$ gauge field and compute correlation functions order-by-order in powers of $1/N_f$. The inclusion of leading irrelevant operators results in a boson self energy $\Pi(\omega)$ and an incoherent conductivity $\sigma(\omega > 0)$ that both scale as $\sim \omega^{-2(z-2)/z}$, where $z$ is the dynamical critical exponent of the boson[7] (the same scaling has been found for other order parameters in Refs. [64, 73–76]).

In Appendix F, we revisit these calculations in detail. For the boson self energy $\Pi(\omega)$, we find surprising cancellations between various diagrams that lead to a *vanishing* coefficient of $\omega^{-2(z-2)/z}$ for all inversion-odd order parameters. Any further frequency dependence must then come from subleading irrelevant operators that give rise to a higher power of $\omega$. For inversion-even and non-constant order parameters, no such cancellation happens and the $\omega^{-2(z-2)/z}$ scaling is robust. Taking these results as a starting point, we then evaluate the current-current correlator and find that the coefficient of $\omega^{-2(z-2)/z}$ in the conductivity *vanishes* for all order parameters. We note that a similar claim of "vanishing" $\omega^{-2(z-2)/z}$ coefficient has been emphasized in a recent work [63]. However, their results were obtained for the special case of a constant scalar form factor $f_a(\boldsymbol{k}) = 1$ in the random-flavor large

---

[7]If the boson kinetic term is local, then $z = 3$ at one loop. A correction to $z = 3$ was calculated at four loops [72], but whether this correction is ultimately canceled against others has not been settled.

$N$ expansion and for the Galilean invariant form factor in the RPA expansion. It would be interesting to understand whether the dichotomy between inversion-even and inversion-odd order parameters that we found can be reproduced in a generalization of their calculation that includes arbitrary form factors in the random flavor large N model.

Finally, we emphasize that the $1/N_f$ (i.e. RPA) expansion is only valid in some intermediate range of $\omega$ lower than the energy scale of Landau damping but much higher than the energy scale where fermionic quasiparticles are destroyed (this latter energy scale is proportional to $N_f^{1/(2/z-1)}$). Therefore, our analysis does not extend down to the IR fixed point, where the effects of irrelevant operators could potentially lead to a different frequency scaling in $\sigma(\omega)$. Nailing down the critical exponents of the IR fixed point and the scaling dimensions of leading irrelevant operators remains a challenging task that we hope to return to in the future.

## 4   Random-flavor large $N$ model

### 4.1   Motivation and summary of results

In Ref. [19], we found using general symmetry and anomaly arguments that the boson propagator at $\boldsymbol{q} = 0$ in the mid-IR theory, Eqs. $(3.1) - (3.2)$, is exactly fixed to be a frequency-independent constant in the IR limit. Related arguments also provide the relation between the optical conductivity and the boson propagator, Eq. $(3.7)$, thereby fixing the Drude form of the conductivity. In order to obtain a non-trivial incoherent conductivity at the IR fixed point, it would be especially convenient to find a model which retains the anomaly relation in Eq. $(3.7)$ without fixing the frequency dependence of the boson propagator.

As explained in Ref. [19], a model which achieves this is the "random-flavor large $N$" theory introduced in Refs. [59–61]. In the mid-IR formulation of this model, $N$ species of bosons, $\phi_I$, and $N$ species of fermions, $\psi_{\theta,I}$, are coupled through (spatially uniform) Gaussian random variables, $g_{IJK}$,

$$S_{\phi\psi} = \int_{\boldsymbol{x},\tau} \frac{g_{IJK}}{N} f_a(\theta) \, \psi_{\theta,I}^{\dagger} \, \psi_{\theta,J} \, \phi_K^a \, , \tag{4.1}$$

$$\overline{g_{IJK}^* g_{I'J'K'}} = g^2 \, \delta_{II'} \delta_{JJ'} \delta_{KK'} \, , \quad \overline{g_{IJK}} = 0 \, , \tag{4.2}$$

where $f_a(\theta)$ is a form factor depending on the Fermi surface angle (patch index), $\theta$, and $a$ is again an internal index. Because each $\phi_I$ generally couples to a complicated linear combination of flavor density operators, $\psi_I^{\dagger}\psi_J$, rather than the conserved (up to anomaly)

total patch charge, the anomaly structure of this random-flavor large $N$ model no longer fixes the boson propagator, allowing for the possibility of a nonvanishing incoherent conductivity at the fixed point.

If one considers the conductivity taking the $N \to \infty$ prior to the low frequency (IR) limit, one finds a vanishing incoherent conductivity [62]. However, the physically correct approach takes the IR limit first, and the two limits need not commute, as famously occurs in the large $N_f$, or RPA, expansion [28]. To perform this calculation, we leverage the non-perturbative anomaly relations explained in Section 3 and developed in detail in Ref. [19]. The anomaly constraints furnish exact relations between different correlation functions, which are valid in any order of limits, facilitating the extraction the fixed point conductivity in a procedure we dub the *anomaly-assisted large N expansion*.

The relations used in the anomaly-assisted approach are simple to outline. For example, the nonperturbative susceptibility arguments in Section 3 carry over in a similar way to this mid-IR theory, leading to a Drude weight,

$$N \, \mathcal{D}^{ij} = N \left[ \mathcal{D}_0^{ij} - \frac{\pi}{N} V^{ia} V^{jb} \left( \Pi_0^{-1} \right)_{ab} \right] \tag{4.3}$$

with $\mathcal{D}_0^{ij}, V^{ia}, \Pi_0^{ab}$ respectively having the same expressions as in Eqs. (3.8) and (3.13), except that the form factor $\frac{1}{N} \sum_{IJ} g_{IIJ} f^a(\theta)$ replaces $g^a(\theta)$. Furthermore, a generalized version of Eq. (3.7) persists for the full optical conductivity,

$$\sigma^{ij}(\boldsymbol{q} = 0, \omega) = N \frac{i}{\omega} \left[ \frac{\mathcal{D}_0^{ij}}{\pi} - \frac{1}{N} \sum_{I,J,K,L} \frac{g_{IIJ}}{N} \frac{g_{KKL}}{N} \operatorname{Tr}_\theta \left[ v_F^i f^a \right] \operatorname{Tr}_\theta \left[ v_F^j f^b \right] D_{ab}^{JL}(\boldsymbol{q} = 0, \omega) \right] , \tag{4.4}$$

where $D_{ab}^{JL} = \langle \phi_{a,J} \phi_{b,L} \rangle$ is the boson propagator. This relation relates the large-$N$ and small frequency expansions of the boson propagator to those of the conductivity.

However, the calculation of the conductivity indeed depends on the order of the large $N$ and IR limits. As mentioned above, the physical order of limits is to take the IR limit *before* the large-$N$ limit. Treating this order of limits improperly by naïvely plugging the $N \to \infty$ form of the boson propagator into Eq. (4.4) would thus lead to incorrect results. This can be seen immediately from the fact that the second term in Eq. (4.4) vanishes as $N \to \infty$, contradicting the exact Drude weight reduction in Eq. (4.3). Physically, the failure of these limits to commute reflects the existence of "slow modes" in the system; that is, modes whose relaxation times become large as $N \to \infty$.

The memory matrix formalism (for a review, see Ref. [71]) allows for a transparent treatment of these slow modes. Combining this technique with non-perturbative susceptibility

formulae determined by the anomaly, we obtain a final result for the zero temperature optical conductivity with both coherent and incoherent contributions,

$$\sigma^{ij}(\omega) = N \frac{i}{\omega} \left[ \frac{\mathcal{D}_0^{ij}}{\pi} - \mathrm{Tr}_\theta \left[ v_F^i f^k \right] \mathrm{Tr}_\theta \left[ v_F^j f^l \right] \left( \Pi_0 - i C_z \, N \, \omega^{1-2/z} \right)_{kl}^{-1} + \dots \right] \tag{4.5}$$

$$= \frac{N \mathcal{D}^{ij}}{\pi} \frac{i}{\omega} + N^2 \, \mathcal{C}_z^{ij} \, \omega^{-2/z} + \dots \, , \quad \text{for } \omega^{1-2/z} \ll \frac{1}{N} \ll 1 \, , \tag{4.6}$$

where $\mathcal{C}_z^{ij}$ and $C_z^{ij}$ are constant matrices depending on $z$. The ellipses contain terms that are less singular in $\omega$ as $\omega \to 0$. Interestingly, the incoherent conductivity in this expression differs in an essential way from the $\omega^{-2(z-2)/z}$ scaling generated by the leading irrelevant operators (see Appendix F). Being intrinsic to the IR fixed point theory, this result also indicates a problem with continuing $N$ to unity: as we found in Ref. [19], the incoherent conductivity of the theory at $N = 1$ *must vanish*. Note that we also compute the conductivity at finite temperature, but we find that thermal effects dominate over quantum critical effects to the order at which we calculate. It is possible that there is an IR scale at which this is no longer the case, although it is beyond the scope of the large $N$ analysis performed here [77]. We will discuss the finite temperature calculation in detail in Section 4.4 and Appendix C.

It is clear from Eq. (4.5) that our final result depends sensitively on the order of the large-$N$ and low frequency limits. If the large-$N$ limit is taken first, then the resulting conductivity contains the free fermion Drude weight, contradicting the exact result in Eq. (4.3). In this order of limits, there are also frequency-dependent incoherent terms at sub-leading orders in $1/N$. In contrast, taking the low frequency limit first yields a correct result for the Drude weight along with a frequency-dependent incoherent conductivity that is *enhanced* by a power of $N$. This signals a breakdown of the large $N$ expansion in evaluating the conductivity.

The remainder of this work will be devoted to the derivation and interpretation of the result in Eq. (4.5). In Section 4.2, we write down the model and set up important notations before briefly reviewing the formalism developed by Ref. [61]. In Section 4.3, we then compute the conductivity at fixed frequency $\omega$ in the $N \to \infty$ limit and explain why the perturbative $1/N$ expansion of Ref. [61] cannot access the true fixed point conductivity. To go beyond the $1/N$ expansion, we leverage non-perturbative anomaly arguments to improve the conductivity calculation, first in the memory matrix formalism (Section 4.4), and then in the standard Kubo formalism (Section 4.5). These complementary approaches ultimately yield the same IR scaling quoted in Eq. (4.5), providing a nontrivial consistency check of our calculations.

## 4.2 Model and large $N$ expansion

Following Refs. [59–61], we consider $N$ flavors of bosons $\phi_I$ and $N$ flavors of fermions $\psi_I$ described by the action,

$$S = S_\phi + S_\psi + S_{\phi\psi}\,, \tag{4.7}$$

$$S_\phi = \frac{1}{2}\int_{\boldsymbol{q},\tau} \vec{\phi}_I(\boldsymbol{q},\tau)\left[(-\lambda\,\partial_\tau^2 + |\boldsymbol{q}|^2)\,\delta_{IJ} + m_{IJ}^2\right]\vec{\phi}_J(\boldsymbol{q},\tau) \tag{4.8}$$

$$S_\psi = \int_{\boldsymbol{k},\tau} \psi_I^\dagger(\boldsymbol{k},\tau)\left[\partial_\tau + \epsilon(\boldsymbol{k})\right]\psi_I(\boldsymbol{k},\tau)\,, \tag{4.9}$$

$$S_{\phi\psi} = \frac{g_{IJK}}{N}\int_{\boldsymbol{k},\boldsymbol{q},\tau} \vec{f}(\boldsymbol{k})\cdot\vec{\phi}_K(\boldsymbol{q},\tau)\,\psi_I^\dagger(\boldsymbol{k}+\boldsymbol{q}/2,\tau)\,\psi_J(\boldsymbol{k}-\boldsymbol{q}/2,\tau)\,. \tag{4.10}$$

Here we have introduced a form factor $\vec{f}(\boldsymbol{k})$ that is independent of the flavor indices, as well as the flavor-dependent Yukawa couplings $g_{IJK}$. The mass matrix, $m_{IJ}^2$, tunes the theory to a multicritical point where all of the boson species are gapless. As briefly mentioned in Section 4.1, we will sample these couplings from a complex Gaussian distribution with $\overline{g_{IJK}} = 0$ and $\overline{g_{IJK}^* g_{I'J'K'}} = g^2 \delta_{II'}\delta_{JJ'}\delta_{KK'}$.

In the low energy limit, we can consider the "mid-IR" theory analogous to the one described in Section 3.1, with action

$$S = S_\phi + \sum_\theta S_{\text{patch}}(\theta)\,, \tag{4.11}$$

where

$$S_{\text{patch}}(\theta) = \int_{\boldsymbol{x},\tau} \psi_{I,\theta}^\dagger\left[\partial_\tau + \epsilon_\theta(\boldsymbol{\nabla})\right]\psi_{I,\theta} + \frac{g_{IJK}}{N}\int_{\boldsymbol{x},\tau} f^a(\theta)\,\phi_{a,K}\,\psi_{I,\theta}^\dagger\,\psi_{J,\theta}\,, \tag{4.12}$$

where $\epsilon_\theta(\boldsymbol{\nabla}) = iv_F(\theta)\boldsymbol{w}(\theta)\cdot\boldsymbol{\nabla} + \kappa_{ij}(\theta)\partial_i\partial_j$ is the fermion dispersion expanded to quadratic order inside the patch $\theta$. We again drop the curvature of the dispersion but retain the curvature of the Fermi surface.

We now adapt the large $N$ technology developed by [61] to study the mid-IR theory of loop current order defined in Eq. (4.11). A crucial simplifying feature of the model is that for physical quantities that are $\mathcal{O}(1)$ in the $N \to \infty$ limit, annealed averaging agrees with quenched averaging up to $\mathcal{O}(1/N^2)$ corrections[8]. More explicitly, self-averaging is treated as the statement,

$$\overline{\log Z} = \log\overline{Z}\,, \tag{4.13}$$

---

[8]This *self-averaging* property was assumed in Ref. [61]. In Appendix B, we provide a concrete diagrammatic derivation. To our knowledge, ours is the first such derivation in the literature on these models.

As we now review, this leads to a bilocal collective field description of the original theory that is valid to leading two orders in the $1/N$ expansion. While this is a powerful calculational tool for various correlation functions at finite $\omega, \boldsymbol{q}$ in the large $N$ limit, there are interesting non-perturbative effects that cannot be captured by this formalism if one keeps $N$ finite and take the limit as $\omega, \boldsymbol{q} \to 0$. These non-perturbative subtleties will be explained in the later sections.

To arrive at the bilocal collective field description, we disorder-average the partition function over complex Gaussian couplings $g_{IJK}$

$$\overline{Z} = \int \mathcal{D}\phi \, e^{-S_{\text{boson}}} \, \overline{Z}_{\text{patch}}[\phi] \,, \tag{4.14}$$

$$\overline{Z}_{\text{patch}}[\phi] = \int \left[ \prod_\theta \mathcal{D}\psi_\theta^\dagger \mathcal{D}\psi_\theta \right] \exp\left\{ -\sum_\theta \int_{\boldsymbol{x},\tau} \psi_{I,\theta}^\dagger [\partial_\tau + \epsilon_\theta(\boldsymbol{\nabla})]\psi_{I,\theta} - |\mathcal{O}_{IJK}|^2 \right\}, \tag{4.15}$$

$$\mathcal{O}_{IJK} = \sum_\theta \int_{\boldsymbol{x},\tau} f^a(\theta) \, \phi_K^a \, \psi_{I,\theta}^\dagger \, \psi_{J,\theta} \,. \tag{4.16}$$

Generalizing the notation of [61] to multiple patches, we introduce bilocal fields,

$$D^{ab}(x,y) = \frac{1}{N} \sum_K \phi_K^a(x) \, \phi_K^b(y) \,, \qquad G_{\theta\theta'}(x,y) = \frac{1}{N} \sum_I \psi_{I,\theta}^\dagger(x) \, \psi_{I,\theta'}(y) \,. \tag{4.17}$$

Here we use notation where $x, y$ are coordinates in space and imaginary time, i.e. $x = (\tau, \boldsymbol{x})$, and we will denote $\int_x \equiv \int d^3x$. The interacting part of the effective action can now be recast as

$$S_{\text{int}} = \frac{g^2 N}{2} \sum_{\theta,\theta'} \int_{x,y} f^a(\theta) \, f^b(\theta') \, D^{ab}(x,y) \, G_{\theta\theta'}(x,y) \, G_{\theta'\theta}(y,x) \,. \tag{4.18}$$

The identification of $D^{ab}, G_{\theta\theta'}$ with boson/fermion bilinears is enforced by Lagrange multipliers $\Pi^{ab}$ and $\Sigma_{\theta\theta'}$ coupling as

$$S_{\Sigma G} = -N \int_{x,y} \Sigma_{\theta'\theta}(y,x) \left[ G_{\theta\theta'}(x,y) - \frac{1}{N} \sum_I \psi_{I,\theta}^\dagger(x) \, \psi_{I,\theta'}(y) \right], \tag{4.19}$$

$$S_{\Pi D} = N \int \int_{x,y} \Pi_{\theta'\theta}^{ab}(y,x) \left[ D^{ab}(x,y) - \frac{1}{N} \sum_K \phi_K^a(x) \, \phi_K^b(y) \right]. \tag{4.20}$$

After integrating out the original fields $\psi_{\theta,I}$ and $\phi_I$, one obtains a path integral over the bilocal fields $G_{\theta\theta'}, \Sigma_{\theta\theta'}, D^{ab}, \Pi^{ab}$,

$$\bar{Z} = \int_{G,D,\Sigma,\Pi} \exp\left\{ -N S_{\text{eff}}[G, D, \Sigma, \Pi] \right\}, \tag{4.21}$$

where the effective action $S_{\text{eff}}$ takes the form

$$S_{\text{eff}}[G, D, \Sigma, \Pi] = \text{Tr} \ln \left[ i\omega_n - \epsilon(\boldsymbol{k}) - \Sigma_{\theta\theta'} \right] + \frac{1}{2} \text{Tr} \ln \left[ |\boldsymbol{q}|^2 - \Pi^{ab} \right] + S_{\text{int}}$$

$$- \sum_{\theta,\theta'} \int_{x,y} \Sigma_{\theta'\theta}(x, y) \, G_{\theta\theta'}(y, x) + \frac{1}{2} \int_{x,y} \Pi^{ab}(x, y) \, D^{ba}(y, x) \, . \tag{4.22}$$

Here we use $\boldsymbol{k}$ and $\boldsymbol{q}$ to respectively denote the fermion and boson momenta, and we use $\omega_n$ and $\Omega_n$ to denote the fermion and boson Matsubara frequencies.

The path integral in Eq. (4.21) is amenable to a standard large $N$ saddle point expansion. For simplicity, we will restrict below to models where only a single boson component couples strongly to the FS, so the indices $a, b$ will be dropped below. The spatial symmetries of the Yukawa coupling would then be encoded completely in the symmetries of a single function $f(\theta)$. To leading order in $N$, classical minimization of $S_{\text{eff}}$ yields saddle point equations,

$$G_{\theta\theta}(\boldsymbol{k}, i\omega_n) = \frac{1}{i\omega_n - \epsilon(\boldsymbol{k}) - \Sigma_{\theta\theta}(\boldsymbol{k}, i\omega_n)} \, , \quad D(\boldsymbol{q}, i\Omega_m) = \frac{1}{|\boldsymbol{q}|^2 - \Pi(\boldsymbol{q}, i\Omega_m)} \, , \tag{4.23}$$

$$\Sigma_{\theta\theta}(x) \approx g^2 \, [f(\theta)]^2 \, D(x) \, G_{\theta\theta}(x) \, , \quad \Pi(x) = -g^2 \sum_{\theta} [f(\theta)]^2 \, G_{\theta\theta}(x) \, G_{\theta\theta}(-x) \, . \tag{4.24}$$

Solving these equations for a rotationally invariant Fermi surface is a standard computation in Migdal-Eliashberg theory. But since we are considering generic Fermi surfaces, we state slightly more general formulae for the saddle point solutions, leaving a self-contained derivation to Appendix C. For the boson self energy, we find standard Landau damping in the regime $|\Omega_m| \ll |\boldsymbol{q}|$,

$$\Pi(\boldsymbol{q}, i\Omega_m) = -\gamma_{\hat{\boldsymbol{q}}} \frac{|\Omega_m|}{|\boldsymbol{q}|} \, , \tag{4.25}$$

where the coefficient $\gamma_{\hat{\boldsymbol{q}}}$ is determined by the choice of form factor $f(\theta)$ and the Fermi surface parameters $k_F(\theta), v_F(\theta)$, and it is a function of the orientation of the boson momentum, $\hat{\boldsymbol{q}}$. The fermion self energy decomposes into terms generated by quantum and thermal fluctuations,

$$\Sigma_{\theta\theta}(\boldsymbol{k}, i\omega_n) = \Sigma_{T,\theta\theta}(\boldsymbol{k}, i\omega_n) + \Sigma_{Q,\theta\theta}(\boldsymbol{k}, i\omega_n) \, , \tag{4.26}$$

where the quantum term, $\Sigma_Q$, obeys quantum critical scaling and the thermal term, $\Sigma_T$, is generated by dangerously irrelevant boson self-interactions at finite temperature (more comments on this point will be given in Section 4.4.4). At low $\omega$ and $T$, neither of these terms depend on the spatial momentum $\boldsymbol{k}$,

$$\Sigma_{T,\theta\theta}(\boldsymbol{k}, i\omega_n) = -i \, \text{sgn}(\omega_n) \, h_\theta \sqrt{\frac{T}{\ln(1/T)}} \tag{4.27}$$

$$\Sigma_{Q,\theta\theta}(\boldsymbol{k}, i\omega_n) = -i \, \text{sgn}(\omega_n) \, \lambda_\theta \, T^{2/z} \, H_z \left( |\omega_n|/T \right) \, . \tag{4.28}$$

Here $\lambda_\theta, h_\theta$ are positive functions of $\theta$ while $H_z(\xi)$ approaches a constant as $\xi \to 0$ and scales as $\xi^{2/z}$ for $\xi \gg 1$. These scaling forms will play an important role in the calculation of the conductivity.

Going beyond the saddle point, one can expand the action in Eq. (4.22) to higher order in the fluctuations of bilocal fields and obtain corrections to various correlation functions in powers of $1/N$. However, we emphasize again that the random-flavor large $N$ theory *fails to be self-averaging* beyond order $1/N$. This means that the utility of the bilocal field description is restricted to the level of quadratic fluctuations and higher order corrections are not relevant to the physical model (see Appendix B). This behavior can be contrasted with the $q = 4$ SYK model, where self-averaging only begins to fail at $\mathcal{O}(1/N^3)$ [78].

### 4.3   Calculation of the conductivity in the naïve large $N$ limit

To compute the conductivity, we start with the standard Kubo formula

$$\sigma^{ij}(\boldsymbol{q} = 0, \omega) = \frac{i}{\omega} \left[ \frac{N \mathcal{D}_0^{ij}}{\pi} - G_{J^i J^j}(\boldsymbol{q} = 0, \omega) \right] . \tag{4.29}$$

The first diamagnetic term is dictated by electromagnetic gauge invariance and provides the free-fermion Drude weight. The second term is the current-current correlator. Within the mid-IR theory, the current operator can be decomposed into contributions from different patches,

$$J^i = \sum_\theta j_\theta^i \quad j_\theta^i = v_F(\theta)\, w^i(\theta)\, n_\theta\,, \tag{4.30}$$

where $n_\theta = \sum_I \psi_{\theta,I}^\dagger \psi_{\theta,I}$ is the chiral charge density[9] within patch $\theta$. In the presence of Yukawa interactions, the $U(1)_{\text{patch}}$ symmetry corresponding to the conservation of $n_\theta$ becomes anomalous and $n_\theta$ acquires nontrivial dynamics

$$\frac{d}{dt}\, n_\theta(\boldsymbol{q} = 0, t) = -\frac{\Lambda(\theta)}{(2\pi)^2} \frac{f(\theta)}{v_F(\theta)} \sum_{I,J} \frac{g_{IIJ}}{N} \frac{d}{dt}\, \phi(\boldsymbol{q} = 0, t)\,. \tag{4.31}$$

After substituting the above expression in Eq. (4.29), we obtain an exact formula for the conductivity in terms of the connected boson correlator, $D_{JL} = \langle \phi_J \phi_L \rangle$,

$$\sigma^{ij}(\boldsymbol{q} = 0, \omega) = N \frac{i}{\omega} \left[ \frac{\mathcal{D}_0^{ij}}{\pi} - \frac{1}{N} \sum_{I,J,K,L} \frac{g_{IIJ}}{N} \frac{g_{KKL}}{N} V^i V^j D^{JL}(\boldsymbol{q} = 0, \omega) \right], \tag{4.32}$$

---

[9]Note that this operator identification depends on a choice of UV regularization [19, 30].

where $V^i = \mathrm{Tr}_\theta \, [v_F^i f]$. We now import some results from the bilocal field formalism in Section 4.2. For fixed $\omega$, taking the large $N$ limit gives $D^{JL}(\boldsymbol{q} = 0, \omega) \approx \delta_{JL} D_\star(\boldsymbol{q} = 0, \omega)$ up to subleading in $N$ corrections, where $D_\star$ is the saddle point solution for the boson propagator. Under this approximation,

$$\sum_{I,J,K,L} \frac{g_{IIJ}}{N} \frac{g_{KKL}}{N} D^{JL}(\boldsymbol{q} = 0, \omega) = g^2 D_\star(\boldsymbol{q} = 0, \omega) + \mathcal{O}(1/N), \qquad (4.33)$$

and the conductivity simplifies to

$$\sigma^{ij}(\boldsymbol{q} = 0, \omega) = \frac{i}{\omega} \left[ \frac{N \mathcal{D}_0^{ij}}{\pi} - g^2 \, V^i \, V^j \, D_\star(\boldsymbol{q} = 0, \omega) + \mathcal{O}(1/N) \right]. \qquad (4.34)$$

Importantly, the saddle point solution $D_\star(\boldsymbol{q} = 0, \omega)$ is independent of $N$ and the second term is subleading in $N$ relative to the first term. Therefore we recover the free-fermion Drude weight $\mathcal{D}_0^{ij}$ independent of $f(\theta)$. In the bilocal field formalism, this simple result corresponds to an infinite sum of ladder diagrams generated by quadratic fluctuations of the bilocal fields $\delta G, \delta \Sigma, \delta D, \delta \Pi$ (we resum these diagrams within the two-patch model in Appendix D. Ref. [63] contains a more involved calculation that accounts for the full Fermi surface and obtains the same answer). The matching of these two calculations showcases the power of anomaly arguments: what seems like a complicated diagrammatic exercise ultimately reduces to a one-line derivation.

Importantly, this result is only correct for fixed $\omega$ in the $N \to \infty$ limit. This explains why the correct Drude weight in Eq. (4.3) is invisible in the perturbative $1/N$ expansion. More generally, since properties of the IR fixed point always require keeping $N$ finite and taking $\omega \to 0$, the perturbative $1/N$ expansion would also miss any putative frequency scaling in the incoherent conductivity. Accessing the ultimate IR conductivity requires more nonperturbative input, which will be the focus of the next subsection.

## 4.4  Anomaly-assisted large $N$ expansion: Memory matrix approach

The subtle order of limits between $\omega \to 0$ and $N \to \infty$ originates from the existence of slow modes whose relaxation rates vanish as $N \to \infty$. The method of choice for studying the conductivity in the presence of slow modes is the memory matrix formalism, which we review and apply in the rest of this section.

The physical intuition behind the memory matrix formalism is a separation of timescales. In a strongly interacting quantum many-body system with slow modes (i.e. approximately conserved quantities), the decay rates of slow modes are generally much smaller than the

decay rates of generic short-lived excitations. When this is the case, IR properties of the system are governed by fluctuations of the slow modes alone, which live in a low-dimensional subspace $\mathcal{S}$ of the Hilbert space of operators. In the context of generalized transport, the IR property of interest is the correlation function $\mathcal{C}_{\mathcal{O}_1\mathcal{O}_2}$ which captures the linear response of an operator $\mathcal{O}_1$ to a source conjugate to $\mathcal{O}_2$. This correlation function is simply related to the retarded Green's function $G^R_{\mathcal{O}_1\mathcal{O}_2}$ via

$$\mathcal{C}_{\mathcal{O}_1\mathcal{O}_2}(\boldsymbol{q}=0,\omega,T) \equiv G^R_{\mathcal{O}_1\mathcal{O}_2}(\boldsymbol{q}=0,\omega,T) - \chi_{\mathcal{O}_1\mathcal{O}_2}. \tag{4.35}$$

For a time-reversal symmetric system where slow modes dominate in the IR, we therefore expect the correlation function to take a "generalized Drude form"

$$\mathcal{C}_{\mathcal{O}_1\mathcal{O}_2}(\omega,T) = -\sum_{A,B\in\mathcal{S}} \mathcal{D}^{\mathcal{O}_1\mathcal{O}_2}_{AB} \left[1 + i\omega^{-1}\tau^{-1}(\omega,T)\right]^{-1}_{AB}, \tag{4.36}$$

where the matrix $\tau^{-1}_{AB}$ encodes the decay rate of an operator $A$ in response to a source conjugate to $B$ and the matrix $\mathcal{D}^{\mathcal{O}_1\mathcal{O}_2}_{AB}$ is a set of generalized Drude weights that measure the overlap of $\mathcal{O}_1, \mathcal{O}_2 \in \mathcal{S}$ with $A, B \in \mathcal{S}$.

The crux of the memory matrix formalism is an explicit recipe for calculating the phenomenological parameters $\mathcal{D}, \tau^{-1}$ from microscopic correlation functions

$$\mathcal{D}^{\mathcal{O}_1\mathcal{O}_2}_{AB} = \sum_{C\in\mathcal{S}} \chi_{\mathcal{O}_1 A}(\chi^{-1})_{BC}\chi_{\mathcal{O}_2 C}, \qquad \tau^{-1}_{AB} = \sum_{C\in\mathcal{S}} \chi^{-1}_{AC} M_{CB}(\omega,T), \tag{4.37}$$

$$M_{AB}(\omega,T) = i\beta(A|L\mathsf{q}(\omega - \mathsf{q}L\mathsf{q})^{-1}\mathsf{q}L|B), \tag{4.38}$$

where $M$ is the eponymous memory matrix. Here we have defined the inner product, $(A|B) = \beta^{-1}\chi_{AB}$, the Liouvillian operator, $L = [H, \cdot]$, and the projector $\mathsf{q}$ onto the subspace $\mathcal{S}^{\perp}$ orthogonal to $\mathcal{S}$. A particularly transparent derivation of these results can be found in Ref. [71]. The electrical conductivity is obtained by noting $\mathcal{C}_{J^iJ^j}(\omega,T) \equiv i\omega\,\sigma^{ij}(\omega,T)$, so we recover the generalized Drude form,

$$\sigma^{ij}(\omega,T) = \sum_{A,B,C\in\mathcal{S}} \chi_{J^iA}\left[-i\omega + \tau^{-1}(\omega,T)\right]^{-1}_{AB}(\chi^{-1})_{BC}\chi_{J^jC}. \tag{4.39}$$

In the context of the random-flavor model of interest, Eqs. (4.11) – (4.12), we identify the slow subspace as the $N_{\text{patch}} + N$ dimensional space spanned by the conserved patch densities $\tilde{n}_\theta$ as well as the boson fields themselves $\phi_I$,

$$\mathcal{S} = \{\tilde{n}_\theta, \phi_I\}. \tag{4.40}$$

We will see below that the inclusion of the boson modes in the slow subspace is consistent at large $N$. A key assumption of our calculation is the absence of additional slow operators in the model. One possible effect of additional slow operators could be to generate a more singular frequency scaling of the conductivity.

By applying anomaly arguments, we can determine the static susceptibilities $\chi$ in this subspace non-perturbatively. After plugging the exact $\chi$ into Eq. (4.39) and evaluating the memory matrix $M(\omega, T)$ perturbatively in the $1/N$ expansion, we can simplify the conductivity to a sum of coherent and incoherent terms. The exactly conserved operators contribute to the coherent conductivity and reproduce the nonperturbative Drude weights in Eq. (4.3); the almost conserved slow operators contribute to the incoherent conductivity and reproduce the frequency scaling in Eq. (4.5). We now proceed to go through these steps.

### 4.4.1 Static susceptibilities in the slow subspace

As discussed in Section 3, in the mid-IR theory, the perpendicular current operator $J^i$ can be written as a sum over chiral densities, $n_\theta$, on the Fermi surface patches,

$$J^i = \sum_\theta j^i_\theta \,, \quad j^i_\theta = v_F(\theta)\, w^i(\theta)\, n_\theta \,. \tag{4.41}$$

Due to the large emergent symmetry, the current $J^i$ overlaps with a large number of conserved charge densities, $\tilde{n}_\theta$. As for the $N = 1$ model discussed in Section 3.2, these $\tilde{n}_\theta$'s are further related to $n_\theta$ by a term involving a linear combination of the boson fields

$$\tilde{n}_\theta = n_\theta + \frac{\Lambda(\theta)}{(2\pi)^2} \frac{f(\theta)}{v_F(\theta)} \sum_{I,J} \frac{g_{IIJ}}{N} \phi_J \tag{4.42}$$

$$= n_\theta + F(\theta)L(\theta) \sum_J G_J \, \phi_J \,, \tag{4.43}$$

where we absorbed miscellaneous constants into $L(\theta) = \sqrt{\frac{\Lambda(\theta)}{(2\pi)^2 v_F(\theta)}}$, $F(\theta) = f(\theta)L(\theta)$, and $G_J = \sum_I g_{IIJ}/N$ (these choices of constants will significantly simplify the later calculations). Since $\tilde{n}_\theta$'s are exactly conserved, conservation of the chiral densities, $n_\theta$, is broken by the emergent "electric field" generated by the bosons. Thus, the relationship, Eq. (4.42) may be thought of as a rewriting of the *anomaly* of the random-flavor large $N$ model.

The static susceptibilities follow from the same anomaly arguments that we leveraged to study the $N = 1$ model in Ref. [19]. Using the anomaly, Eq. (4.42), and the $n_\theta$ susceptibility,

$$\chi_{n_\theta n_{\theta'}} = NL(\theta)^2 \, \delta_{\theta\theta'} \,, \tag{4.44}$$

we can immediately fix the susceptibility for the conserved densities, $\tilde{n}_\theta$,

$$\chi_{\tilde{n}_\theta \tilde{n}_{\theta'}} = \chi_{n_\theta n_{\theta'}} + F(\theta)L(\theta)\,F(\theta')L(\theta')\sum_{I,J} G_I G_J\, \chi_{\phi_I \phi_J} \tag{4.45}$$

$$= NL(\theta)^2 \delta_{\theta\theta'} + F(\theta)L(\theta)\,F(\theta')L(\theta')\,\bar{G}^2\,\chi_{\phi\phi}\,, \tag{4.46}$$

as well as the cross susceptibilities,

$$\chi_{\tilde{n}_\theta \phi_K} = F(\theta)L(\theta)\sum_I G_I\,\chi_{\phi_I \phi_J} = F(\theta)L(\theta)\,G_J\,\chi_{\phi\phi}\,. \tag{4.47}$$

Note that we have tuned the UV mass matrix such that $\chi_{\phi_I \phi_J} = \chi_{\phi\phi}\delta_{IJ} \to \infty$ at criticality. When $N$ is large, $\bar{G}^2$ should be thought of as an $\mathcal{O}(1)$ constant,

$$\bar{G}^2 = \sum_K G_K^2 = \frac{1}{N^2} \sum_{I,J,K} g_{IIK} g_{JJK} \approx g^2\,, \tag{4.48}$$

where $g^2$ is the variance of the disorder distribution of $g_{IJK}$.

With these non-perturbative susceptibilities at hand, we can immediately work out the structure of the decay rate matrix, $\tau^{-1} = \chi^{-1}M$, at the quantum critical point. We first invert the block matrix $\chi$ to get

$$(\chi^{-1})_{\theta,I;\theta',J} = \begin{pmatrix} \frac{1}{NL(\theta)^2}\delta_{\theta\theta'} & -\frac{F(\theta)}{NL(\theta)}G_J \\ -\frac{F(\theta')}{NL(\theta')}G_I & \chi_{\phi\phi}^{-1}\delta_{IJ} + \frac{1}{N}\bar{F}^2 G_I G_J \end{pmatrix}, \tag{4.49}$$

where $\bar{F}^2 = \int d\theta F(\theta)^2 = \mathrm{Tr}_\theta\,[f^2]$. Since the operators $\tilde{n}_\theta$ are exactly conserved, $M_{\tilde{n}_\theta \mathcal{O}} = 0$ for all operators $\mathcal{O}$. This means that

$$\left(\chi^{-1}M\right)_{\theta,I;\theta',J} = \begin{pmatrix} 0 & -\frac{F(\theta)}{NL(\theta)}G_K M_{KJ} \\ 0 & \chi_{\phi\phi}^{-1}M_{IJ} + \frac{1}{N}\bar{F}^2 G_I G_K M_{KJ} \end{pmatrix}. \tag{4.50}$$

Although $\chi^{-1}M$ is not diagonalizable, we can still define its spectrum as solutions of the equation $\det(\lambda - \chi^{-1}M) = 0$ (we abuse notations a little bit and continue to call these solutions eigenvalues). Since $\lambda - \chi^{-1}M$ is upper block triangular, this amounts to the condition

$$\det\left(\lambda I_{\theta\theta'}\right)\det\left(\lambda\delta_{IJ} - \chi_{\phi\phi}^{-1}M_{IJ} - \frac{1}{N}\bar{F}^2 G_I G_K M_{KJ}\right) = 0\,. \tag{4.51}$$

The $\lambda = 0$ subspace corresponds exactly to the subspace spanned by the $N_{\mathrm{patch}}$ exactly conserved quantities $\tilde{n}_\theta$. The rest of the eigenvalues can be found by diagonalizing $\chi_{\phi\phi}^{-1}M_{IJ} + \frac{1}{N}\bar{F}^2 G_I G_K M_{KJ}$ in the boson subspace. After working out the algebraic details (see Appendix E), we find one eigenvalue $\lambda = \mathcal{O}(1/N)$ and $N-1$ eigenvalues $\lambda = \mathcal{O}(\chi_{\phi\phi}^{-1})$. At sufficiently small temperatures, the boson susceptibility is controlled by the thermal mass and diverges as $\chi_{\phi\phi}^{-1} \sim T\ln(1/T)$. Therefore, for sufficiently low temperatures, all modes in the subspace $\mathcal{S}$ are indeed slow and our choice of slow subspace is self-consistent.

### 4.4.2 Decomposition into coherent and incoherent contributions

We now plug the nonperturbative susceptibities $\chi$ into the generalized Drude formula Eq. (4.39)

$$\sigma^{ij}(\omega) = \frac{i}{\omega} \sum_{A,B \in \mathcal{S}} \chi_{J^i A} \left[\chi + \mathcal{M}\right]^{-1}_{AB} \chi_{J^j B}, \tag{4.52}$$

where $\mathcal{M}$ is related to the memory matrix $M$ by $M = -i\omega\mathcal{M}$. Since the only nontrivial block of $\mathcal{M}$ in the slow subspace $\mathcal{S}$ is the boson block $\mathcal{M}_{IJ} = \mathcal{M}_{\phi_I \phi_J}$, we have

$$\chi + \mathcal{M} = \mathcal{L} \begin{pmatrix} A & B \\ B^T & C \end{pmatrix} \mathcal{L}, \tag{4.53}$$

where the subblocks take the simple form

$$\mathcal{L}_{\theta,I;\theta',J} = \begin{pmatrix} L(\theta)\delta_{\theta\theta'} & 0 \\ 0 & \delta_{IJ} \end{pmatrix}, \tag{4.54}$$

$$A_{\theta\theta'} = N\delta_{\theta\theta'} + F(\theta)F(\theta')\bar{G}^2\chi_{\phi\phi} \quad B_{\theta,L} = F(\theta)G_L\chi_{\phi\phi} \quad C_{KL} = \chi_{\phi\phi}\delta_{KL} + \mathcal{M}_{KL}, \tag{4.55}$$

$$(A^{-1})_{\theta\theta'} = N^{-1} \left[\delta_{\theta\theta'} - \frac{\chi_{\phi\phi}\bar{G}^2\bar{F}^2}{N + \chi_{\phi\phi}\bar{G}^2\bar{F}^2} \frac{F(\theta)F(\theta')}{\bar{F}^2}\right]. \tag{4.56}$$

Since the current operator doesn't overlap with the boson fields (i.e. $\chi_{J^i\phi} = 0$), we only need the projection of the inverse $(\chi + \mathcal{M})^{-1}$ onto the subspace spanned by $\{\tilde{n}_\theta\}$ to compute the conductivity. After carrying out this tedious computation in Appendix E, we obtain a compact formula for the conductivity with both coherent and incoherent contributions

$$\begin{aligned}
\sigma^{ij}(\omega, T) = {} & \frac{i}{\omega}N \left(\mathrm{Tr}_\theta\left[v_F^i v_F^j\right] - \frac{\mathrm{Tr}_\theta\left[v_F^i f\right]\mathrm{Tr}_\theta\left[v_F^j f\right]}{\mathrm{Tr}_\theta\left[f^2\right]}\right) \\
& + \frac{i}{\omega}N^2 \frac{\mathrm{Tr}_\theta\left[v_F^i f\right]\mathrm{Tr}_\theta\left[v_F^j f\right]}{\mathrm{Tr}_\theta\left[f^2\right]\left(N + \mathrm{Tr}_\theta\left[f^2\right]G_K\left[\mathcal{M}(\mathbb{I} + \chi_{\phi\phi}^{-1}\mathcal{M})^{-1}\right]_{KL}G_L\right)}.
\end{aligned} \tag{4.57}$$

So far, this expression is exact. The first term reproduces the Drude weight $\mathcal{D}^{ij}$ derived in Eq. (4.3). The second term simplifies upon taking the $\chi_{\phi\phi} \to \infty$ limit. The end result is

$$\sigma^{ij}(\omega, T) = \frac{iN\mathcal{D}^{ij}}{\pi\omega} + \frac{iN^2}{\omega} \frac{\mathrm{Tr}_\theta\left[v_F^i f\right]\mathrm{Tr}_\theta\left[v_F^j f\right]}{\mathrm{Tr}_\theta\left[f^2\right]\left(N + \mathrm{Tr}_\theta\left[f^2\right]G_K\mathcal{M}_{KL}(\omega, T)G_L\right)}. \tag{4.58}$$

Before moving on, we comment on an alternative derivation of the same result that is conceptually more transparent but technically more cumbersome. Instead of directly computing the matrix inverse $(\chi + \mathcal{M})^{-1}$, one could compute the eigenvectors and eigenvalues of the

decay rate matrix $\tau^{-1}$ and identify how each of the $N_{\text{patch}} + N$ eigenmodes contribute to the conductivity. When this diagonalization is carried out, we find $N_{\text{patch}}$ modes in the subspace spanned by $\{\tilde{n}_\theta\}$ with $\lambda_i = \mathcal{O}(1/N)$ independent of $\omega, T$. These *coherent contributions* reduce the Drude weight from the free fermion value $\mathcal{D}_0$ to $\mathcal{D}$. On the other hand, due to quantum criticality, there are $N-1$ modes for which $\lambda_i \sim \chi_{\phi\phi}^{-1} \to 0$. These modes are infinitely slow but do not overlap with the current operator. Thus they do not contribute to the conductivity. The only remaining mode mixes the $\{\phi_I\}$ and $\{\tilde{n}_\theta\}$ sectors and has an eigenvalue $\lambda_i$ that depends nontrivially on $\omega, T$, thereby giving the *incoherent contribution* in Eq. (4.58).

### 4.4.3 Evaluating the memory matrix

The final step is to evaluate the memory matrix $M(\omega, T) = -i\omega\mathcal{M}(\omega, T)$. To do that, we compute the correlation function $\mathcal{C}_{\phi_I\phi_J}(\omega, T)$ defined via Eq. (4.35) in two different ways. In the memory matrix formalism, this correlation function is obtained by plugging $\mathcal{O}_1 = \phi_I$ and $\mathcal{O}_2 = \phi_J$ into Eq. (4.36)

$$\mathcal{C}_{\phi_I\phi_J}(\omega, T) = -\sum_{A,B,C\in\mathcal{S}} \chi_{\phi_I A} \left[1 + i\omega^{-1}\tau^{-1}\right]_{AB}^{-1} (\chi^{-1})_{BC}\chi_{\phi_J C}. \tag{4.59}$$

On the other hand, if one directly plugs $\chi_{\phi_I\phi_J} = \chi_{\phi\phi}\delta_{IJ}$ into Eq. (4.35), one finds

$$\mathcal{C}_{\phi_I\phi_J}(\omega, T) = G_{\phi_I\phi_J}^R(\boldsymbol{q} = 0, \omega, T) - \chi_{\phi\phi}\delta_{IJ}. \tag{4.60}$$

Since all susceptibilities are fixed by anomaly arguments, the only unknown quantity in Eq. (4.59) is the memory matrix $M$ (recall that $\tau^{-1} = \chi^{-1}M$) while the only unknown quantity in Eq. (4.60) is the boson Green's function. Equating these two expressions, we thus obtain a direct relationship between the memory matrix $M$ and the boson Green's function. After some simple algebra (see Appendix E), we can reduce this relationship to a transparent form

$$NG_I(\Pi\mathcal{M})_{IJ}G_J - \text{Tr}_\theta\, f^2 G_I \mathcal{M}_{IJ} G_J = -N\bar{G}^2\,, \tag{4.61}$$

where $\Pi_{IJ} = -(D^{-1})_{IJ}$ is the boson self energy matrix (the bare kinetic term in the boson propagator can be neglected in the deep IR limit).

Despite the simplicity of the above equation, it is difficult to solve for $\mathcal{M}$ at finite $N$ since we have no analytic control over the $\mathcal{O}(N^2)$ off-diagonal components of $\Pi_{IJ}$ in the IR limit. However, assuming that $\mathcal{S} = \{n_\theta, \phi_I\}$ indeed captures all the slow modes, the memory matrix, which plays the role of an effective "resistivity", should be insensitive to the order

in which we take the IR limit ($\omega \to 0$) and the slow decay rate limit ($h \sim 1/N \to 0$). Hence, we are free to take the $N \to \infty$ limit at finite $\omega$ and use the emergent $O(N)$ invariance in this limit to deduce

$$\Pi_{IJ}(\boldsymbol{q} = 0, \omega, T) = \Pi_\star(\boldsymbol{q} = 0, \omega, T)\delta_{IJ} + \mathcal{O}(1/N) \,. \tag{4.62}$$

Since $\text{Tr}_\theta\left[f^2\right] = \mathcal{O}(1)$, the above equation immediately implies $G_I \mathcal{M}_{IJ} G_J \approx -\bar{G}^2 \Pi_\star^{-1}$ in the large $N$ limit. We now plug this relation into Eq. (4.58). In the denominator of the incoherent conductivity, the term proportional to $\mathcal{M}$ dominates over the constant term $N$ deep in the IR limit since $\Pi_\star(\boldsymbol{q} = 0, \omega, T) \to 0$ as $\omega, T \to 0$. Therefore

$$\sigma^{ij}(\omega, T) \approx \frac{iN\mathcal{D}^{ij}}{\pi\omega} - \frac{iN^2}{\omega} \frac{\text{Tr}_\theta\left[v_F^i f\right] \text{Tr}_\theta\left[v_F^j f\right]}{\bar{G}^2 \left(\text{Tr}_\theta\left[f^2\right]\right)^2} \Pi_\star(\boldsymbol{q} = 0, \omega, T) \,. \tag{4.63}$$

### 4.4.4  Frequency and temperature scaling

We are now ready to derive the $\omega, T$ scaling of the incoherent conductivity. Recall the saddle point solution for $\Pi_\star(\boldsymbol{q} = 0, \omega, T)$ derived in Appendix C

$$\Pi_\star(\boldsymbol{q} = 0, \omega, T) = \begin{cases} \tilde{C}\frac{\omega}{h(T)}\pi\left(\frac{\omega}{T}\right) & \omega \ll T \\ \tilde{C}_z(-i\omega)^{1-2/z} & \omega \gg T \end{cases} \,, \tag{4.64}$$

where $h(T) \sim \sqrt{T/\ln(1/T)}$ and $\pi(\xi)$ is a scaling function that has a finite limit as $\xi \to 0$. Using the above form of $\Pi_\star$ in Eq. (4.63), we find

$$\sigma^{ij}(\omega, T) = \frac{iN\mathcal{D}^{ij}}{\pi\omega} + N^2 \begin{cases} \sim \frac{1}{\sqrt{T/\ln(1/T)}} & \omega \ll T \\ \sim \omega^{-2/z} & \omega \gg T \end{cases} \,. \tag{4.65}$$

The most striking feature of this result is that the incoherent conductivity doesn't fit into a scaling function of $\omega$ and $T$. The basic reason is that although the quantum critical point is well-defined at zero temperature, there are dangerously irrelevant operators that turn on at finite temperature and overwhelm the effects of critical fluctuations. These dangerously irrelevant operators generate a thermal mass $M^2(T) \sim T\ln(1/T)$ for the boson (see e.g. Refs. [61, 73, 79, 80]), which ultimately leads to the $[T/\ln(1/T)]^{-1/2}$ conductivity scaling.

If one insists on dropping all dangerously irrelevant operators and focuses strictly on the fixed point theory at finite temperature, perturbative IR-divergences appear in the fermion self energy and obscure the effect of critical fluctuations[10]. One possibility is that the divergence is merely an artifact of perturbation theory and physical correlation functions at the

---

[10]The same divergence is found across a variety of different perturbative expansion schemes [30, 61, 64, 80].

fixed point do fit into a scaling function of $\omega$ and $T$. For the special case of fermions coupled to a $U(1)$ gauge field, there are some ideas towards curing this IR divergence to leading order in the RPA expansion [77]. But since the RPA expansion is known to break down at the lowest energy scales, a fully satisfactory resolution still eludes us.

The other possibility is that the divergence is physical and the fixed point theory fails to satisfy quantum critical scaling. An interesting example featuring this phenomenon is the quantum Lifshitz model in 2+1 dimensions considered in Ref. [81]. Since the quantum Lifshitz model is Gaussian, one can exactly solve the theory and find power law correlations at zero temperature but strictly short-ranged correlations at any finite temperature. Determining whether this sharp distinction between zero and finite temperatures exists in models of metallic quantum criticality is an open problem that we hope to understand better in future work.

## 4.5  Alternate perspective: Kubo formula and anomaly substitution

While physically intuitive, the memory matrix derivation of the conductivity is rather technically involved. Here we present a simpler but more heuristic derivation of the memory matrix result by augmenting the standard Kubo formula with anomaly constraints, albeit with one uncontrolled assumption.

The starting point of the derivation is the exact Drude weight derived from susceptibility arguments in Section 3,

$$\frac{N\mathcal{D}^{ij}}{\pi} = \lim_{\omega \to 0} -i\omega\, \sigma^{ij}(\boldsymbol{q}=0,\omega) = N\left(\frac{\mathcal{D}_0^{ij}}{\pi} - \frac{\mathrm{Tr}_\theta\left[v_F^i f\right]\mathrm{Tr}_\theta\left[v_F^j f\right]}{\mathrm{Tr}_\theta\left[f^2\right]}\right). \tag{4.66}$$

As shown in Eq. (4.32) of Section 4.3, the current-current correlation function can be expressed in terms of the boson propagator

$$\sigma^{ij}(\omega) = \frac{iN}{\omega}\left[\frac{\mathcal{D}_0^{ij}}{\pi} - \frac{1}{N}\sum_{I,J,K,L}\frac{g_{IIJ}}{N}\frac{g_{KKL}}{N}\mathrm{Tr}_\theta\left[v_F^i f\right]\mathrm{Tr}_\theta\left[v_F^j f\right]D^{JL}(\boldsymbol{q}=0,\omega)\right]. \tag{4.67}$$

For this relation to be consistent with the non-perturbative result for the Drude weight in Eq. (4.66), the boson propagator, $D^{JL}(\boldsymbol{q}=0,\omega\to 0)$, must satisfy

$$\frac{N}{\mathrm{Tr}_\theta\left[f^2\right]} = \frac{g_{IIJ}\,g_{KKL}}{N^2}D^{JL}(\boldsymbol{q}=0,\omega\to 0). \tag{4.68}$$

The above relation is exact. But this single constraint does not fix all components of the matrix $D^{JL}$. To make more progress, we need to make one uncontrolled assumption: Based

on the approximate $O(N)$ symmetry of the disorder averaged theory, we assume that the boson propagator is approximately an $O(N)$ singlet,

$$D^{JL}(\boldsymbol{q}=0,\omega,T) \approx \delta^{JL} D(\boldsymbol{q}=0,\omega,T)\,. \tag{4.69}$$

This assumption, combined with the independent randomness of $g_{IJK}$, implies that different terms in the sum over $I,J,K,L$ destructively interfere unless $I=K$ and $J=L$. The constraint in Eq. (4.68) then simplifies to

$$\frac{N}{\mathrm{Tr}_\theta\left[f^2\right]} \approx -\bar{G}^2\,\Pi^{-1}(\boldsymbol{q}=0,\omega\to0) = \bar{G}^2 D(\boldsymbol{q}=0,\omega\to0)\,, \tag{4.70}$$

where $\Pi(\boldsymbol{q},\omega)$ is the boson self energy. Plugging this simplified constraint back into the Kubo formula, Eq. (4.68), and expressing $\Pi$ as a sum of its $N\to\infty$ saddle point solution, $\Pi_\star$, and a sub-leading constant term dictated by Eq. (4.70) gives

$$\begin{aligned}
\sigma^{ij}(\omega) &= \frac{iN}{\omega}\left[\frac{\mathcal{D}_0^{ij}}{\pi} - \frac{1}{N}\bar{G}^2\,\mathrm{Tr}_\theta\left[v_F^i f\right]\mathrm{Tr}_\theta\left[v_F^j f\right] D(\boldsymbol{q}=0,\omega\to0)\right]\\
&\quad + \frac{i\bar{G}^2}{\omega}\mathrm{Tr}_\theta\left[v_F^i f\right]\mathrm{Tr}_\theta\left[v_F^j f\right]\left[D(\boldsymbol{q}=0,\omega\to0) - D(\boldsymbol{q}=0,\omega)\right]\\
&= \frac{iN\mathcal{D}^{ij}}{\omega\pi} + \frac{i\bar{G}^2}{\omega}\mathrm{Tr}_\theta\left[v_F^i f\right]\mathrm{Tr}_\theta\left[v_F^j f\right]\frac{N}{\bar{G}^2\,\mathrm{Tr}_\theta\left[f^2\right]}\left[1 - \left(1 - \frac{N\Pi_\star(\boldsymbol{q}=0,\omega,T)}{\bar{G}^2\,\mathrm{Tr}_\theta\left[f^2\right]}\right)^{-1}\right]\\
&\approx \frac{iN\mathcal{D}^{ij}}{\omega\pi} - \frac{iN^2}{\omega}\frac{\mathrm{Tr}_\theta\left[v_F^i f\right]\mathrm{Tr}_\theta\left[v_F^j f\right]}{\bar{G}^2\left(\mathrm{Tr}_\theta\left[f^2\right]\right)^2}\Pi_\star(\boldsymbol{q}=0,\omega,T)\,,\\
&= \frac{iN\mathcal{D}^{ij}}{\omega\pi} + N^2\frac{\mathrm{Tr}_\theta\left[v_F^i f\right]\mathrm{Tr}_\theta\left[v_F^j f\right]}{\bar{G}^2\left(\mathrm{Tr}_\theta\left[f^2\right]\right)^2}C_z\,\omega^{-2/z}\,, \qquad T\ll\omega\,.
\end{aligned} \tag{4.71}$$

Note that we neglected corrections with higher powers of $\Pi_\star(\boldsymbol{q}=0,\omega,T)$. The last line exactly matches the incoherent conductivity Eq. (4.63) that we found in the memory matrix formalism.

# 5 Discussion

The central theme that underlies this work and its close companion [19] is the quest for quantum critical incoherent conductivity in clean models of non-Fermi liquids. Specifically, our search is focused on the important class of non-Fermi liquids described by the Hertz-Millis framework, where gapless bosonic order parameter fields with zero crystal momentum couple

strongly to a Fermi surface. Although we have not yet considered finite-momentum order parameters, there has been much recent progress in studying that class of theories [82, 83].

In Ref. [19], leveraging nonperturbative anomaly constraints, we showed that the incoherent conductivity always vanishes at the IR fixed points of conventional Hertz-Millis models. In the present work, we demonstrated how one can evade these strict constraints in a random-flavor large $N$ deformation of the conventional model that possesses a distinct anomaly structure. A reliable transport analysis at the IR fixed point of this deformed model necessitates the development of a novel calculational tool dubbed *anomaly-assisted perturbation theory* that goes beyond the naïve large $N$ expansion. Using this approach, we found that optical transport in the IR limit depends strongly on the symmetries of the gapless order parameter: while the incoherent conductivity continues to vanish for inversion-even order parameters (e.g. Ising-nematic), it exhibits a nontrivial frequency and temperature scaling for inversion-odd order parameters including the titular loop current order.

Our results for the random-flavor model provide a positive answer to the guiding questions posed in Section 1. However, there are a number of undesirable features intrinsic to the random-flavor model that undermine its physical significance. First of all, since $O(N)$ symmetry is explicitly broken by random-flavor interactions, the model at any finite $N$ is a multicritical point with $N^2$ relevant couplings tuned to zero. A priori, one may hope that this multicritical point merges with the standard Hertz-Millis QCP as $N \to 1$. But our transport results demonstrate that this is impossible, since the random-flavor model features a nontrivial incoherent conductivity at any finite $N$ that is absent in the $N = 1$ model. Thus the random-flavor model cannot be viewed as the starting point of a controlled expansion that accesses the physical Hertz-Millis QCP. Moreover, even if we view the random-flavor model as an interesting QCP on its own, the bilocal field formalism used to compute its properties at $N = \infty$ relies on a crucial self-averaging property that only holds to leading two orders in the $1/N$ expansion. Extrapolating to small $N$ thus appears intractable. Finally, even at the $N = \infty$ QCP, the random-flavor model suffers from IR divergences at finite temperature which can only be cured by dangerously irrelevant operators. Thermal effects induced by these operators overwhelm the quantum critical fluctuations and ultimately destroy the putative $\omega/T$ scaling in the optical conductivity.

The Hertz-Millis models studied here generically have nonzero Drude weight: thus, in addition to the incoherent conductivity, the full optical conductivity $\text{Re}\,\sigma(\omega)$ also contains a narrow "coherent" peak (which becomes a delta function in the IR fixed-point theory). This Drude weight only goes to zero in certain fine-tuned limits, though for loop current order parameters it is more generally diminished by the coupling to the boson. This is

despite the diverging susceptibility of an order parameter that is odd under time-reversal and inversion symmetry, and is due to the presence of an infinite number of emergent conserved quantities that overlap with the electrical current most of which have finite susceptibilities. The criterion for vanishing Drude weight discussed in Ref. [15] is thus *necessary* but not sufficient. For the future it will be interesting to explore models of clean compressible metals beyond the Hertz-Millis paradigm where the Drude weight goes to zero. In thinking about experiments, one could also consider the possibility that disorder has the effect of broadening the coherent peak to an extent that it cannot be observed, without necessarily substantially affecting the incoherent conductivity.

The absence of quantum critical transport in generic Hertz-Millis models describing zero momentum ordering transitions, along with the presence of unphysical features in the random-flavor deformation, should not subvert the significance of our results in this paper. Indeed it is very likely that some of the prominent examples of strange metals observed in experiments (e.g. in the cuprates or in several quantum critical heavy fermion metals) are not described within the Hertz-Millis paradigm. Thus, the purpose of studying these Hertz-Millis models is not to directly explain experiments, but rather to "build muscle" in preparation for analyzing more realistic theories of strange metals. Such theories may involve, for instance, emergent gauge fields and multiple emergent matter fields, some of which may form gapless Fermi surfaces of their own. For such future studies, what lessons can we learn from our exploration of transport in the Hertz-Millis models? A crucial lesson is the benefits of focusing on the emergent symmetries and anomalies of the low energy theory. In previous papers, we discussed how such a focus leads to some general conceptual statements about the IR fixed point. In the present work, we showed how this focus provides calculational benefits as well. It is our hope that the anomaly-assisted large-$N$ expansion introduced in the context of the random-flavor Hertz-Millis models will facilitate controlled transport calculations in more complex models of metallic QCPs to be explored in the future.

## Acknowledgements

We thank Ehud Altman, Andrey Chubukov, Ilya Esterlis, Eduardo Fradkin, Haoyu Guo, Sean Hartnoll, Steve Kivelson, Sung-Sik Lee, Aavishkar Patel, Sri Raghu, Subir Sachdev, and Cenke Xu for discussions. HG was supported by the Gordon and Betty Moore Foundation EPiQS Initiative through Grant No. GBMF8684 at the Massachusetts Institute of Technology. DVE was supported by the Gordon and Betty Moore Foundation EPiQS Initiative through Grant No. GBMF8683 at Harvard University. TS was supported by US

Department of Energy grant DE- SC0008739, and partially through a Simons Investigator Award from the Simons Foundation. This work was also partly supported by the Simons Collaboration on Ultra-Quantum Matter, which is a grant from the Simons Foundation (651446, TS).

## A   Computing the Drude weight from susceptibilities

In this appendix, we compute the Drude weight, given the susceptibilities described in Section 3. Let us define $\chi(\theta, \theta') := \chi_{\tilde{n}_\theta \tilde{n}_{\theta'}}$. We start from Eq. (3.10), which we write as

$$\chi_{\tilde{n}_\theta, \tilde{n}_{\theta'}} = \frac{\Lambda(\theta)}{(2\pi)^2 v_F(\theta)}\delta_{\theta\theta'} + \frac{1}{(2\pi)^4}\frac{g^i(\theta)g^j(\theta')}{v_F(\theta)v_F(\theta')}(M^{-1})_{ij}\Lambda(\theta)\Lambda(\theta')\,. \tag{A.1}$$

Here $\Lambda(\theta)$ is the momentum cutoff in the direction parallel to the Fermi surface, which approaches $\left|\frac{d}{d\theta}\mathbf{k}_F(\theta)\right|$ as $N_{\text{patch}} \to \infty$. For the sake of greater generality we introduced the boson mass matrix $M^{ij}$, such that setting $M^{ij} = (m^2 - m_c^2)\delta^{ij}$ recovers Eq. (3.10).

We need to find the inverse function $\chi^{-1}(\theta, \theta')$, which by definition satifies

$$\sum_{\theta'}\chi(\theta, \theta')\chi^{-1}(\theta', \theta'') = \delta_{\theta\theta''}\,. \tag{A.2}$$

We make the ansatz that

$$\chi^{-1}(\theta, \theta') = \frac{(2\pi)^2 v_F(\theta)}{\Lambda(\theta)}\delta_{\theta\theta'} - g^i(\theta)G_{ij}g^j(\theta')\,, \tag{A.3}$$

for some matrix $G$ to be determined. Then we compute

$$\sum_{\theta'}\chi(\theta, \theta')\chi^{-1}(\theta', \theta'') = \delta_{\theta\theta''} + \frac{\Lambda(\theta)}{(2\pi)^2 v_F(\theta)}g^i(\theta)g^j(\theta'')(M^{-1} - G - M^{-1}\Pi_0 G)_{ij}\,. \tag{A.4}$$

where we defined the matrix $\Pi_0^{ij} = \text{Tr}_\theta\, g^i g^j$. Hence we can ensure that $\chi$ and $\chi^{-1}$ satisfy Eq. (A.2) provided that

$$M^{-1} - G - M^{-1}\Pi_0 G = 0\,, \tag{A.5}$$

or in other words,

$$G = (M + \Pi_0)^{-1}\,. \tag{A.6}$$

Now, the generalization of Eq. (2.8) is

$$\mathcal{D}^{ij} = \sum_{\theta,\theta'}\chi_{J^i\tilde{n}_\theta}\chi_{J^j\tilde{n}_{\theta'}}\chi^{-1}(\theta, \theta')\,. \tag{A.7}$$

Substituting Eq. (A.3) and Eq. (3.11), we obtain Eq. (3.12).

Finally, let us give the proof of the result claimed in Section 3.3, that the Drude weight at the critical point vanishes if and only if there exists some matrix $S^i{}_j$ (independent of $\theta$) such that

$$v^i_F(\theta) \equiv w^i(\theta)v_F(\theta) = S^i{}_j g^j(\theta) \tag{A.8}$$

for all $\theta$. Towards this end, we will rewrite Eq. (3.12) in a different way. Let us define

$$\widetilde{w}^i(\theta) = w^i(\theta) - \frac{1}{v_F(\theta)} V^{ij} G_{jk} g^k(\theta) \,, \tag{A.9}$$

where $V^{ij} = \mathrm{Tr}_\theta \left[ v^i_F g^j \right]$. Then we find that

$$\frac{1}{(2\pi)^2} \sum_\theta v_F(\theta) \widetilde{w}^i(\theta) \widetilde{w}^j(\theta) \Lambda(\theta) \tag{A.10}$$

$$= \frac{1}{\pi} \mathcal{D}^{ij}_0 + \frac{1}{(2\pi)^2} \left( -2V^{ik} V^{jl} G_{kl} + V^{ik} G_{kl} V^{jr} G_{rs} \Pi^{ls}_0 \right) . \tag{A.11}$$

$$= \frac{\mathcal{D}^{ij}}{\pi} \,, \tag{A.12}$$

where $\mathcal{D}^{ij}$ is given by Eq. (3.12), and we have used the fact that at criticality, $M = 0$ and hence $G = \Pi^{-1}_0$. It follows that $\mathcal{D}^{ij} = 0$ if and only if $\widetilde{w}^i(\theta) = 0$ for all $\theta$. Now, if $\widetilde{w}^i(\theta) = 0$ then we find that $w^i(\theta)v_F(\theta) = S^i{}_j g^j(\theta)$ with $S^i{}_j = V^{ik} G_{kj}$. Conversely, if there exists $S^i{}_j$ satisfying Eq. (A.8) for all $\theta$, then from the definition of $V$, we find that

$$V^{ij} = S^i{}_k \Pi^{jk}_0 . \tag{A.13}$$

Substituting the above form of $V^{ij}$ into Eq. (A.9) gives $\widetilde{w}^i(\theta) = 0$ for all $\theta$.

# B  Extent of self-averaging in the random-flavor model

The goal of this section is to show that $\mathcal{O}(1)$ thermodynamic observables and dynamical correlation functions within the random-flavor large N model are self averaging up to $\mathcal{O}(\frac{1}{N^2})$ corrections.

To avoid notational clutter, we will work in the one-patch model with a scalar form factor (the same arguments apply to the general case because the nature of the arguments is completely combinatorial). The action for a fixed set of couplings $g_{IJK}$ is given by

$$S[g] = \int d\tau d^2 x \psi^\dagger_I \left[ \partial_\tau - \epsilon(\boldsymbol{k}) \right] \psi_I + \phi_I(-\partial^2_\tau - \nabla^2)\phi_I + \frac{g_{IJK}}{N} \psi^\dagger_I \psi_J \phi_K$$
$$= S_0 + \frac{g_{IJK}}{N} \int d\tau d^2 x \psi^\dagger_I \psi_J \phi_K \,. \tag{B.1}$$

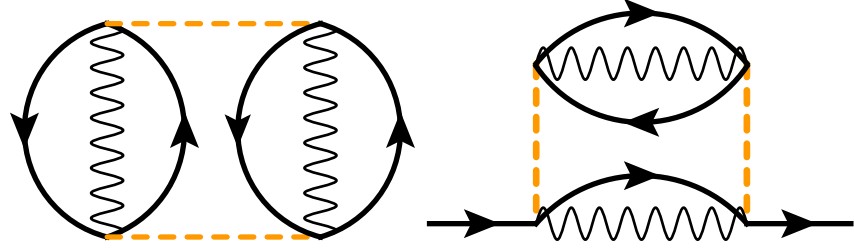

**Figure 3:** The lowest order Feynman diagram that contributes to the annealed average but not the quenched average of free energy (left) and correlation functions (right).

The associated partition function is

$$Z[g] = \int D\psi D\phi \exp\left\{-S_0 - \frac{g_{IJK}}{N} \int d\tau d^2 x \psi_I^\dagger \psi_J \phi_K\right\}. \tag{B.2}$$

In the annealed average, we perform a disorder average of the partition function to get

$$\overline{Z[g]} = \int Dg D\psi D\phi \exp\left\{-\sum_{IJK} \frac{g_{IJK}^2}{2g^2} - S_0 - \frac{g_{IJK}}{N} \int d\tau d^2 x \psi_I^\dagger \psi_J \phi_K\right\}. \tag{B.3}$$

Therefore, within the linked cluster expansion $\log \overline{Z[g]}$ will be a sum over connected diagrams with $g$ treated as a fluctuating field. On the other hand, $\overline{\log Z[g]}$ will not contain diagrams that would be disconnected in the absence of $g$ propagators. In exact analogy with SYK [78], the lowest order term that contributes to the annealed but not the quenched average is shown on the left half of Figure 3. Here solid lines denote the fermionic propagators, wiggly lines denote bosonic propagators, and dotted orange lines denote disorder averages over pairs of vertices. The four interaction vertices bring down a factor of $N^{-4}$. But the summation over three internal propagators give $N^3$. Therefore, the overall scaling of this diagram is $\mathcal{O}(1/N)$ which is $N^{-2}$ suppressed relative to the dominant $\mathcal{O}(N)$ contribution to the free energy. Now for the replicated average $\overline{Z[g]^M}$, the Gaussian part of the action that we expand around is replica diagonal. Therefore, the linked cluster expansion only captures the replica diagonal (RD) saddle. What we have established is then

$$\log \overline{Z[g]} = \overline{\log Z_{\mathrm{RD}}[g]} + \mathcal{O}(\frac{1}{N}). \tag{B.4}$$

Since exact diagonalization or quantum Monte Carlo numerics on this model are not available, we cannot check that $I_\gamma > I_{RD}$ for all off-diagonal saddles $\gamma$. However, there is some quantum Monte Carlo evidence up to $N \sim 40$ that the 0+1 dimensional version of the problem ($g_{IJK}\psi_I^\dagger \psi_J \phi_K$ coupling without spatial dependence) is well described by the replica-diagonal saddle [84]. Therefore, *assuming the off-diagonal saddles are suppressed also in the*

*2+1D problem*, we have

$$\log \overline{Z[g]} = \overline{\log Z[g]} + \mathcal{O}(\frac{1}{N}) \,. \tag{B.5}$$

Since all thermodynamic observables can be obtained from derivatives of the free energy, we conclude that the non self-averaging corrections are always $\mathcal{O}(N^{-2})$ suppressed relative to the leading $\mathcal{O}(N)$ contribution.

The preceding analysis can be easily generalized to correlation functions. Let us consider singlet operators (i.e. an operator $f(\psi, \phi)$ built out of $N^{-1}\sum_I \psi_I^\dagger(\boldsymbol{x}, \tau)\psi_I(\boldsymbol{x'}, \tau')$ and $N^{-1}\sum_I \phi_I(\boldsymbol{x}, \tau)\phi_I(\boldsymbol{x'}, \tau')$) whose leading correlation functions are $\mathcal{O}(1)$. The quenched averages for these operators are defined as

$$\langle f(\psi, \phi)\rangle_{\text{quenched}} = \int Dg\, e^{-\sum_{IJK} g_{IJK}^2/2g^2} \left( \frac{\int D\psi D\phi f(\psi, \phi) \exp\{-S[g]\}}{\int D\psi D\phi \exp\{-S[g]\}} \right)$$
$$= \int Dg\, e^{-\sum_{IJK} g_{IJK}^2/2g^2} \lim_{M\to 0} \int D\psi^a D\phi^a \exp\{-S[g, \psi^a, \phi^a]\} f(\psi^M, \phi^M) \,. \tag{B.6}$$

where $a = 1, \ldots, M$ label the replica indices and $S[g, \psi^a, \phi^a]$ is the M-fold replicated action of the fermions. On the other hand, the annealed averages are defined as

$$\langle f(\psi, \phi)\rangle_{\text{annealed}} = \frac{\int Dg\, e^{-\sum_{IJK} g_{IJK}^2/2g^2} \int D\psi D\phi f(\psi, \phi) \exp\{-S[g]\}}{\int Dg\, e^{-\sum_{IJK} g_{IJK}^2/2g^2} \int D\psi D\phi \exp\{-S[g]\}} \,. \tag{B.7}$$

Within the linked cluster expansion, we again see that the quenched average $\langle f(\psi, \phi)\rangle_{\text{quenched}}$ is a disorder average of all the connected fermion-boson diagrams, while $\langle f(\psi, \phi)\rangle_{\text{annealed}}$ contains in addition disconnected fermion-boson diagrams that become connected by $g$ propagators. If we take $f(\psi, \phi) = N^{-1}\sum_I \psi_I^\dagger(\boldsymbol{x}, \tau)\psi_I(\boldsymbol{x'}, \tau')$, then the first diagram of this kind is shown on the right half of Figure 3.

By O(N) invariance, we can fix the external vertex to be $i$. The two $g$ propagators give $N^{-4}$ and the internal index sums give $N^2$. Therefore this diagram contributes at $\mathcal{O}(N^{-2})$. In conclusion,

$$\left\langle \frac{1}{N}\sum_I \psi_I^\dagger(\boldsymbol{x}, \tau)\psi_I(\boldsymbol{x'}, \tau') \right\rangle_{\text{annealed}} = \left\langle \frac{1}{N}\sum_I \psi_I^\dagger(\boldsymbol{x}, \tau)\psi_I(\boldsymbol{x'}, \tau') \right\rangle_{\text{quenched}} + \mathcal{O}(\frac{1}{N^2}), \tag{B.8}$$

which means that in the $1/N$ expansion of the annealed average, we can only trust the leading $1/N$ correction and no higher. A similar story holds for higher-point correlation functions. The general intuition is that diagrams not shared between the annealed and quenched averages must have $n \geq 4$ interaction vertices and the contractions by g-propagators must get rid of at least two internal loops. This means we always have a $1/N^2$ suppression relative to the leading order diagrams.

# C   Saddle point solutions for boson and fermion self energies in the random-flavor model

## C.1   Computing the boson self energy $\Pi(|q| \gg |\Omega_m|, T)$

We first evaluate the boson self energy $\Pi(\boldsymbol{q}, i\Omega_m, T)$ in the Landau damping regime, where it doesn't depend on the precise form of the fermion self energy $\Sigma_{\theta\theta}$, so long as we make the self-consistent assumption that $\Sigma_{\theta\theta}$ doesn't vary with the spatial momentum $\boldsymbol{k}$. We start with the last equation in Eq. (4.24),

$$\Pi(\boldsymbol{q}, i\Omega_m) = -g^2 T \sum_\theta \int_\theta \frac{d^2k}{(2\pi)^2} \sum_{\omega_n} |f(\theta)|^2 G_{\theta\theta}(\boldsymbol{k}, i\omega_n) G_{\theta\theta}(\boldsymbol{k} + \boldsymbol{q}, i\omega_n + i\Omega_m), \qquad (C.1)$$

where $\int_\theta \frac{d^2k}{(2\pi)^2}$ denotes a momentum integral inside the patch $\theta$ and $\phi_{v_1 v_2}$ is the angle measured from vector $v_2$ to $v_1$ in the counterclockwise direction. To simplify this integral, we use the simple identity

$$G_{\theta\theta}(\boldsymbol{k}, i\omega_n) G_{\theta\theta}(\boldsymbol{k}+\boldsymbol{q}, i\omega_n + i\Omega_m) \approx \frac{G_{\theta\theta}(\boldsymbol{k}, i\omega_n) - G_{\theta\theta}(\boldsymbol{k} + \boldsymbol{q}, i\omega_n + i\Omega_m)}{i\Omega_m - |\boldsymbol{q}| v_F(\theta) \cos \phi_{\boldsymbol{v}_F(\theta)\boldsymbol{q}} - \Sigma_{\theta\theta}(i\omega_n + i\Omega_m) + \Sigma_{\theta\theta}(i\omega_n)}.$$
$$(C.2)$$

Within the patch $\theta$, we can perform a change of variables from $k_x, k_y$ to $\epsilon, \bar{\theta}$ where $\epsilon$ is an energy coordinate and $\bar{\theta} \in [\theta, \theta + \frac{2\pi}{N_{\text{patch}}}]$ is a continuous angular coordinate. After accounting for the appropriate Jacobian factor

$$d^2k = d\epsilon d\bar{\theta} \frac{k_F(\theta)}{v_F(\theta) \cos \phi_{\boldsymbol{k}_F(\theta)\boldsymbol{v}_F(\theta)}}, \qquad (C.3)$$

the integral over $\bar{\theta}$ simply gives a factor of $\Lambda(\theta)$ (the momentum cutoff in the patch) while the integral over $\epsilon$ can be done by residue since the only $\epsilon$-dependence in the integrand comes from the factor $G(\boldsymbol{k}, i\omega_n) - G(\boldsymbol{k} + \boldsymbol{q}, i\omega_n + i\Omega_m)$, in which $\epsilon$ appears linearly in the denominator

$$\int d\epsilon \left[ G_{\theta\theta}(\boldsymbol{k}, i\omega_n) - G_{\theta\theta}(\boldsymbol{k} + \boldsymbol{q}, i\omega_n + i\Omega_m) \right] = \pi i \left[ \text{sgn}(\omega_n + \Omega_m) - \text{sgn}(\omega_n) \right]. \qquad (C.4)$$

Using Eq. (C.4) and Eq. (C.3) in Eq. (C.1), we are left with

$$\Pi(\boldsymbol{q}, i\Omega_m)$$
$$\approx -\frac{ig^2 T}{4\pi|\boldsymbol{q}|} \sum_{\omega_n} \int d\theta \frac{k_F(\theta)|f(\theta)|^2}{v_F(\theta) \cos \phi_{\boldsymbol{k}_F(\theta)\boldsymbol{v}_F(\theta)}} \frac{\text{sgn}(\omega_n + \Omega_m) - \text{sgn}(\omega_n)}{i\frac{\Omega_m}{|\boldsymbol{q}|} - v_F(\theta) \cos \phi_{\boldsymbol{v}_F(\theta)\boldsymbol{q}} - \frac{\Sigma_{\theta\theta}(i\omega_n+i\Omega_m)}{|\boldsymbol{q}|} + \frac{\Sigma_{\theta\theta}(i\omega_n)}{|\boldsymbol{q}|}}.$$
$$(C.5)$$

To do the last integral over $\theta$, we make the assumption (which we later check to be self-consistent) that the internal boson fluctuations are dominated by the kinematic regime where $|\Omega_m|, |\Sigma_{\theta\theta}(i\omega_n + i\Omega_m) - \Sigma_{\theta\theta}(i\omega_n)| \ll |\boldsymbol{q}|$. This means that in the IR limit, we can use the identity $\text{Im} \frac{1}{x - i\epsilon} \approx i\pi \, \text{sgn}(\epsilon)\delta(x)$ with $\epsilon = [\Omega_m + i\Sigma_{\theta\theta}(i\omega_n + i\Omega_m) - i\Sigma_{\theta\theta}(i\omega_n)] |\boldsymbol{q}|^{-1}$ so that for every orientation $\hat{\boldsymbol{q}}$, the angular integral localizes to patches $\theta_{\hat{\boldsymbol{q}}}, \theta_{\hat{\boldsymbol{q}}} + \pi$ where $\cos \phi_{\boldsymbol{v}_F(\theta_{\hat{\boldsymbol{q}}})\boldsymbol{q}} = 0$ (i.e. $\vec{q}$ is tangent to the patches). Finally, summing the contributions from these two patches and doing the Matsubara sum over $\omega_n$ gives

$$
\Pi(\boldsymbol{q}, i\Omega_m) \approx -\frac{ig^2}{4\pi|\boldsymbol{q}|} \frac{k_F(\theta_{\hat{\boldsymbol{q}}})|f(\theta_{\hat{\boldsymbol{q}}})|^2}{v_F(\theta_{\hat{\boldsymbol{q}}}) \cos \phi_{\boldsymbol{k}_F(\theta_{\hat{\boldsymbol{q}}})\boldsymbol{v}_F(\theta_{\hat{\boldsymbol{q}}})}} \frac{\Omega_m}{\pi} (-i\pi) \, \text{sgn}(\Omega_m) \int d\theta \delta \left[ v_F(\theta) \cos \phi_{\boldsymbol{v}_F(\theta)\boldsymbol{q}} \right]
$$

$$
= -\frac{g^2}{2\pi} \frac{k_F(\theta_{\hat{\boldsymbol{q}}})|f(\theta_{\hat{\boldsymbol{q}}})|^2}{v_F(\theta_{\hat{\boldsymbol{q}}})^2 \cos \phi_{\boldsymbol{k}_F(\theta_{\hat{\boldsymbol{q}}})\boldsymbol{v}_F(\theta_{\hat{\boldsymbol{q}}})}|C'(\theta_{\hat{\boldsymbol{q}}})|} \frac{|\Omega_m|}{|\boldsymbol{q}|} = -\gamma(\theta_{\hat{\boldsymbol{q}}}) \frac{|\Omega_m|}{|\boldsymbol{q}|},
$$

(C.6)

where we defined $C(\theta) = \cos \phi_{\boldsymbol{v}_F(\theta)\hat{\boldsymbol{q}}}$ and then absorbed all the angular dependence into $\gamma(\theta_{\hat{\boldsymbol{q}}})$ in the last line[11]. The final formula has the expected structure: the Landau damping coefficient depends on data localized to the two anti-podal patches tangent to $\hat{\boldsymbol{q}}$.

## C.2 Computing the fermion self energy

We now turn to the saddle point solution of the electron self-energy $\Sigma_{\theta\theta}(\boldsymbol{k}, i\omega_n)$ in an arbitrary patch $\theta$, which is given by a single boson-fermion loop Eq. (4.24). Since the loop integral is dominated by small transverse fluctuations of the boson, we can always take $|q_\perp| \ll |q_\parallel| \ll k_F$ where $q_\perp, q_\parallel$ are the virtual boson momentum components perpendicular/parallel to the patch $\theta$. Within this approximation, the loop integral in $\Sigma_{\theta\theta}(\boldsymbol{k}, i\omega_n)$ simplifies to

$$
\Sigma_{\theta\theta}(i\omega_n) \approx \frac{g^2 T}{4\pi^2} \int d^2q \sum_{i\Omega_m} \frac{1}{q_\parallel^2 + \gamma(\theta_{\hat{\boldsymbol{q}}}) \frac{|\Omega_m|}{|q_\parallel|}} \frac{|f(\theta)|^2}{i\omega_n + i\Omega_m - \epsilon(\boldsymbol{k}) - q_\perp v_F(\theta) - \Sigma_{\theta\theta}(i\omega_n + i\Omega_m)}.
$$

(C.7)

Now we encounter a problem: at finite temperature the above equation suffers from an IR divergence. This can be seen by decomposing the fermion self energy into a thermal part $\Sigma_T$ and a quantum part $\Sigma_Q$

$$
\Sigma_{\theta\theta}(\boldsymbol{k}, i\omega_n) = \Sigma_{\theta\theta,T}(\boldsymbol{k}, i\omega_n) + \Sigma_{\theta\theta,Q}(\boldsymbol{k}, i\omega_n),
$$

(C.8)

with

$$
\Sigma_{\theta\theta,T}(\boldsymbol{k}, i\omega_n) = g^2 T \int \frac{d^2q}{(2\pi)^2} D(\boldsymbol{q}, i\Omega_m = 0) G_{\theta\theta}(\boldsymbol{k} - \boldsymbol{q}, i\omega_n),
$$

(C.9)

---

[11]As a sanity check, note that for a rotationally invariant Fermi surface, $k_F(\theta) = m v_F(\theta)$ is independent of $\theta$, $\vec{f}(\theta) = v_F(\cos\theta, \sin\theta)$, and all other angular factors evaluate to 1. Therefore we recover the more familiar Landau damping coefficient $\frac{g^2 k_F}{2\pi}$.

$$\Sigma_{\theta\theta,Q}(\boldsymbol{k}, i\omega_n) = g^2 T \int \frac{d^2q}{(2\pi)^2} \sum_{\Omega_m \neq 0} D(\boldsymbol{q}, i\Omega_m) G_{\theta\theta}(\boldsymbol{k} - \boldsymbol{q}, i\omega_n - i\Omega_m) \,. \tag{C.10}$$

At the multi-critical point, the quantum part is IR-convergent and gives a scaling form

$$\Sigma_{\theta\theta,Q}(i\omega_n) = -i\lambda_\theta \operatorname{sgn}(\omega_n) T^{2/z} H_{1-2/z}\left(\frac{|\omega_n| - \pi T}{2\pi T}\right), \tag{C.11}$$

where $\lambda_\theta = \frac{g^2 2^{2/3}}{3\sqrt{3} v_F(\theta)} |f(\theta)|^2 \gamma(\theta)^{-1/3}$. On the other hand, the thermal part contains an IR-divergent momentum integral (which is in fact observed in other perturbative treatments of Hertz-Millis QCPs [61, 64, 73, 79, 80]). The origin of this IR-divergence is the emergent $U(1)_{\text{patch}}$ gauge invariance in the mid-IR theory which forces the boson to be massless at all $T$. While a general resolution of this divergence is not known, Ref. [77] has shown that, at least within the RPA expansion, the boson self energy and the conductivity are completely insensitive to the IR-divergence. We interpret this as partial evidence that the divergent fermion self energy might be a pathology of the perturbative expansion that is cured in a fully non-perturbative treatment. If this conjecture were true, then the self energies $\Sigma(\boldsymbol{k} = 0, i\omega_n), \Pi(\boldsymbol{q} = 0, i\Omega_m)$ must obey quantum critical scaling with boson dynamical exponent $z$. However, for the microscopic system at finite $T$, this putative scaling form would be overwhelmed by corrections due to dangerously irrelevant terms in the action that we neglected at $T = 0$. The most important term of this kind is a boson self-interaction, which generates a boson thermal mass $M^2(T) \sim T \ln(1/T)$ and regulates the momentum integral in $\Sigma_{\theta\theta,T}$ [61, 73, 79, 80], leading to

$$\Sigma_{\theta\theta,T}(i\omega_n) = -i \operatorname{sgn}(\omega_n) h(T) \quad \Sigma_{\theta\theta,T}(\omega) = -i \operatorname{sgn}(\omega) h(T) \quad h(T) \sim \sqrt{T/\ln(1/T)} \,. \tag{C.12}$$

As we take the IR limit $\omega, T \to 0$ while holding $\omega/T$ fixed, $\Sigma_{\theta\theta,T}$ always dominates over $\Sigma_{\theta\theta,Q}$. In other words, thermal effects hide the quantum critical scaling at finite temperature.

## C.3 Computing the boson self energy $\Pi(q = 0, \omega, T)$

Finally, we compute the leading $\omega, T$-dependence of the boson self energy $\Pi(\boldsymbol{q} = 0, \omega, T)$ quoted in Section 4.4. Note that since we set $\boldsymbol{q} = 0$, this is very different from the Landau damping regime where $|\omega| \ll |\boldsymbol{q}|$.

### C.3.1 Computation at $T = 0$

At $T = 0$, it is convenient to work directly in the Matsubara formalism and only perform the analytic continuation back to real frequency at the very end. Since the singular scalings

in $\omega$ are independent of regularization, we will choose regulators $\Lambda_\perp, \Lambda_\omega$ on $k_x, \Omega_m$ and take $\Lambda_\perp \to \infty$ before $\Lambda_\omega \to \infty$. From the saddle point equations, we know that

$$\Pi(\boldsymbol{q} = 0, i\Omega_m) = -g^2 \sum_\theta f(\theta)^2 T \sum_{\omega_n} \int \frac{d^2k}{(2\pi)^2} G_{\theta\theta}(\boldsymbol{k}, i\omega_n) G_{\theta\theta}(\boldsymbol{k}, i\omega_n + i\Omega_m). \tag{C.13}$$

Within each patch $\theta$, we make a change variables $d^2k = d\epsilon_\theta d\bar{\theta} J(\theta)$ where $\epsilon_\theta, \bar{\theta}$ are energy and angle coordinates in the vicinity of the patch and $J(\theta)$ is a Jacobian factor. To obtain the scaling answer, we take the number of patches to infinity and replace the sum over patches with an angular integral. After introducing a notation $\{\omega_n\} = \text{sgn}(\omega_n)|\omega_n|^{1-2/z}$ and recalling the zero temperature fermion self energy $\Sigma_{\theta\theta}(k, i\omega_n) = -i\lambda_\theta\{\omega_n\}$, we find

$$\Pi(0, i\Omega_m \gg T) \tag{C.14}$$

$$= -g^2 T \sum_{\omega_n} \int d\theta \frac{f(\theta)^2 J(\theta)}{(2\pi)^2} \int d\epsilon_\theta \frac{1}{i\omega_n + i\lambda_\theta\{\omega_n\} - \epsilon_\theta} \frac{1}{i(\omega_n + \Omega_m) + i\lambda_\theta\{\omega_n + \Omega_m\} - \epsilon_\theta} \tag{C.15}$$

$$= -g^2 T \sum_{\omega_n} \int d\theta \frac{f(\theta)^2 J(\theta)}{(2\pi)^2} \pi i \left[ \frac{\text{sgn}(\omega_n + \Omega_m) - \text{sgn}(\omega_n)}{i\Omega_m - i\lambda_\theta\{\omega_n\} + i\lambda_\theta\{\omega_n + \Omega_m\}} \right] \tag{C.16}$$

$$\approx -g^2 \int d\theta \frac{f(\theta)^2 J(\theta)}{(2\pi)^2 \lambda_\theta} \int_{-|\Omega_m|/2}^{|\Omega_m|/2} d\omega \frac{1}{(\omega + |\Omega_m|/2)^{2/z} + (|\Omega_m|/2 - \omega)^{2/z}} \tag{C.17}$$

$$= -\tilde{C}_z |\Omega_m|^{1-2/z}. \tag{C.18}$$

In the last integral over $\omega$, we work in the low $\Omega_m$ limit where the self energy term dominates over the $i\Omega_m$ term for all $\omega \in [-|\Omega_m|/2, |\Omega_m|/2]$. After dropping $i\Omega_m$ in the denominator, the prefactor $\tilde{C}_z(\theta)$ is a $z$-dependent constant multiplied by $\int d\theta \frac{f(\theta)^2 J(\theta)}{(2\pi)^2 \lambda_\theta}$. We will not be interested in the precise value of this constant, noting only that it is real and finite.

Now let us perform the analytic continuation $i\Omega_m \to \omega + i\epsilon$ with $\Omega_m > 0$ with the branch cut placed on the negative real axis

$$|\Omega_m|^{1-2/z} \to (-i \cdot i\Omega_m)^{1-2/z} = (-i\omega)^{1-2/z} \quad \text{for } \Omega_m, \omega > 0. \tag{C.19}$$

We can extend the above function to an analytic function in the upper half $\omega$-plane. Thus the self energy comes out to be

$$\Pi(\boldsymbol{q} = 0, \omega \gg T) \approx -\tilde{C}_z(-i\omega)^{1-2/z}. \tag{C.20}$$

### C.3.2   Computation at $T \neq 0$

At finite temperature, it is difficult to directly compute the boson self energy $\Pi(\boldsymbol{q}, \omega, T)$ in the limit $\omega \ll T$, because the Matsubara frequencies are always larger than $T$. To get

around this, we use the spectral function representation of the fermion Green's function $G_{\theta\theta}(\boldsymbol{k}, i\omega_n) = \int \frac{A_{\theta\theta}(\boldsymbol{k},\omega)}{\omega - i\omega_n}$ inside $\Pi$. This allows us to do the Matsubara sums explicitly and obtain the following expression

$$\Pi(0, i\Omega_m, T) = -g^2 \sum_\theta f(\theta)^2 \int \frac{d^2k}{(2\pi)^2} \int d\omega' d\omega'' A_{\theta\theta}(\boldsymbol{k}, \omega') A_{\theta\theta}(\boldsymbol{k}, \omega'') \frac{n_F(\omega'/T) - n_F(\omega''/T)}{\omega'' - \omega' - i\Omega_m}.$$
(C.21)

Now we can analytically continue $i\Omega_m \to \omega + i\epsilon$ and then safely take the $\omega \ll T$ limit. In terms of the fermion spectral function

$$A_{\theta\theta}(\boldsymbol{k}, \omega') = -\frac{1}{\pi} \frac{\text{Im}\,\Sigma_{\theta\theta}(\omega')}{[\omega' - \epsilon_\theta(\boldsymbol{k}) - \text{Re}\,\Sigma_{\theta\theta}(\omega')]^2 + [\text{Im}\,\Sigma_{\theta\theta}(\omega')]^2}.$$
(C.22)

The boson self energy can be written as the following integral

$$\Pi(0, \omega, T) = -\frac{g^2}{\pi^2} \int d\theta \frac{f(\theta)^2 J(\theta)}{(2\pi)^2} d\omega' d\omega'' \frac{n_F(\omega'/T) - n_F(\omega''/T)}{\omega'' - \omega' - \omega - i\epsilon} \mathcal{I}_\theta(\omega', \omega''),$$
(C.23)

where

$$\mathcal{I}_\theta(\omega', \omega'') = \int d\epsilon_\theta \left( \frac{\text{Im}\,\Sigma_{\theta\theta}(\omega')}{[\omega' - \epsilon_\theta - \text{Re}\,\Sigma_{\theta\theta}(\omega')]^2 + [\text{Im}\,\Sigma_{\theta\theta}(\omega')]^2} \cdot (\omega' \to \omega'') \right).$$
(C.24)

The $\epsilon_\theta$ integral can be done by residue. There are four poles located at

$$\epsilon_\theta = \omega' - \text{Re}\,\Sigma_{\theta\theta}(\omega') \pm i\,\text{Im}\,\Sigma_{\theta\theta}(\omega') \quad \epsilon_\theta = \omega'' - \text{Re}\,\Sigma_{\theta\theta}(\omega'') \pm i\,\text{Im}\,\Sigma_{\theta\theta}(\omega'').$$
(C.25)

Let us take two copies of the integral over $\epsilon_\theta$ and close the contour in the UHP for one copy and in the LHP for the other copy. This gives us a sum over four poles in the full complex plane. In the low frequency limit, $\text{Re}\,\Sigma_{\theta\theta}(\omega) \sim \omega^{2/3}$ while $\text{Im}\,\Sigma_{\theta\theta}(\omega) \sim T^{1/2}$. Thus the dominant contributions to the integral come from $\omega' \ll |\text{Re}\,\Sigma_{\theta\theta}(\omega')| \ll |\text{Im}\,\Sigma_{\theta\theta}(\omega')|, \omega'' \ll |\text{Re}\,\Sigma_{\theta\theta}(\omega'')| \ll |\text{Im}\,\Sigma_{\theta\theta}(\omega'')|$ and we have the clean expression:

$$\begin{aligned}
\mathcal{I}_\theta(\omega', \omega'') &\approx \pi \frac{\text{sgn}\,[\text{Im}\,\Sigma_{\theta\theta}(\omega')]\,\text{sgn}\,[\text{Im}\,\Sigma_{\theta\theta}(\omega'')]\,[|\,\text{Im}\,\Sigma_{\theta\theta}(\omega')| + |\,\text{Im}\,\Sigma_{\theta\theta}(\omega'')|]}{(\text{Re}\,\Sigma_{\theta\theta}(\omega') - \text{Re}\,\Sigma_{\theta\theta}(\omega''))^2 + (|\,\text{Im}\,\Sigma_{\theta\theta}(\omega')| + |\,\text{Im}\,\Sigma_{\theta\theta}(\omega'')|)^2} \\
&\approx \pi \frac{\text{sgn}\,[\text{Im}\,\Sigma_{\theta\theta}(\omega')]\,\text{sgn}\,[\text{Im}\,\Sigma_{\theta\theta}(\omega'')]}{|\,\text{Im}\,\Sigma_{\theta\theta}(\omega')| + |\,\text{Im}\,\Sigma_{\theta\theta}(\omega'')|}.
\end{aligned}$$
(C.26)

Now we plug this approximate expression into the boson self energy integral

$$\Pi(0, \omega, T) \approx -\frac{g^2}{\pi} \int d\theta \frac{f(\theta)^2 J(\theta)}{(2\pi)^2} d\omega' d\omega'' \frac{n_F(\omega'/T) - n_F(\omega''/T)}{\omega'' - \omega' - \omega - i\epsilon} \frac{\text{sgn}\,[\text{Im}\,\Sigma_{\theta\theta}(\omega')]\,\text{sgn}\,[\text{Im}\,\Sigma_{\theta\theta}(\omega'')]}{|\,\text{Im}\,\Sigma_{\theta\theta}(\omega')| + |\,\text{Im}\,\Sigma_{\theta\theta}(\omega'')|}.$$
(C.27)

Now we recall that $|\text{Im}\,\Sigma_{\theta\theta}(\omega, T)| \approx h(T) \sim \sqrt{T/\ln(1/T)}$. Rewriting everything in terms of $T$ and the scaling variables $x = \omega/T, x' = \omega'/T, x'' = \omega''/T$, we find

$$
\begin{aligned}
&\Pi(0, x, T) \\
&\approx -\frac{g^2}{\pi} \int d\theta \frac{f(\theta)^2 J(\theta)}{(2\pi)^2} T \int dx' dx'' \frac{n_F(x') - n_F(x'')}{x'' - x' - x - i\epsilon} \frac{\text{sgn}[x']\,\text{sgn}[x'']}{2h(T)}\,.
\end{aligned}
\tag{C.28}
$$

The singular IR contributions come from the integration domain where $x', x'' \sim \mathcal{O}(x)$. Thus we can expand $n_F(x') - n_F(x'') \approx n_F'(\frac{x'+x''}{2})(x' - x'') + \mathcal{O}(x^2)$ and keep only the leading order term. This gives us

$$
\begin{aligned}
&\Pi(0, x, T) \\
&\approx \frac{g^2}{\pi} \int d\theta \frac{f(\theta)^2 J(\theta)}{(2\pi)^2} T \int dx' dx'' \frac{n_F(x') - n_F(x'')}{x'' - x' - x - i\epsilon} \frac{\text{sgn}[x']\,\text{sgn}[x'']}{2h(T)} \\
&= -\frac{g^2}{\pi} \int d\theta \frac{f(\theta)^2 J(\theta)}{(2\pi)^2} T \int dx' dx'' \frac{e^{\frac{x'+x''}{2}}}{[e^{\frac{x'+x''}{2}} + 1]^2} \left[ 1 + \frac{x^+}{x'' - x' - x^+} \right] \frac{\text{sgn}[x']\,\text{sgn}[x'']}{2h(T)}\,.
\end{aligned}
\tag{C.29}
$$

The first term in the bracket is completely independent of $x$ and should be viewed as a renormalization of the bare mass in this regularization, which can be tuned to zero. The leading singular frequency dependence of $\Pi$ will come from the second term. Taking a factor of $x$ outside the integral, we find a scaling form

$$
\Pi(0, \omega, T) = -\tilde{C} \frac{\omega}{h(T)} \pi \left( \frac{\omega}{T} \right)\,,
\tag{C.30}
$$

where

$$
\pi(x) = \int dx' dx'' \frac{e^{\frac{x'+x''}{2}}}{[e^{\frac{x'+x''}{2}} + 1]^2} \frac{\text{sgn}[x']\,\text{sgn}[x'']}{x'' - x' - x^+}\,.
\tag{C.31}
$$

Numerically we have verified that $\pi(x)$ approaches a nonzero constant as $x \to 0$. On the other hand, for $x \gg 1$, the Taylor expansion of Fermi functions that we performed no long works and we would resort to the earlier zero temperature computation to find $\Pi(0, \omega, T) \sim \text{sgn}(\omega)|\omega|^{1/3}$ to leading order at large $\omega$. To summarize, the leading $\omega, T$-dependent part of the boson self energy takes the form

$$
\Pi(0, \omega, T) = \begin{cases} -\tilde{C} \frac{\omega}{h(T)} \pi \left( \frac{\omega}{T} \right) & \omega \ll T \\ -\tilde{C}_z (-i\omega)^{1-2/z} & \omega \gg T \end{cases}\,.
\tag{C.32}
$$

Note that these two limiting cases cannot be connected by a scaling function of $\omega/T$. This is to be expected because the calculation is controlled by $\Sigma_{\theta\theta, T}$ in the $\omega \ll T$ limit and by $\Sigma_{\theta\theta, Q}$ in the $\omega \gg T$ limit.

# D Contributions to the conductivity from quadratic fluctuations of collective fields

The goal of this appendix is to show, via an explicit diagrammatic calculation in the bilocal field formalism, that in the unphysical order of limits where $N \to \infty$ before $\omega \to 0$, the conductivity is simply the free Fermi gas Drude peak, independent of order parameter symmetries. We first review the structure of $1/N$ corrections in the bilocal field formalism, and then perform the conductivity calculation in the two-patch model. A more involved calculation in Ref. [63] that accounts for the entire Fermi surface gives the same answer.

## D.1 Review of the structure of $1/N$ corrections in the bilocal field formalism

To access the $\mathcal{O}(1/N)$ corrections, one can expand the action Eq. (4.22) to quadratic order in the fluctuations of bilocal collective fields $\delta G_{\theta\theta'}, \delta \Sigma_{\theta\theta'}, \delta D, \delta \Pi$ and read off the propagators that emerge. Instead of going through this procedure for the mid-IR theory (which is conceptually straightforward but notationally heavy), we restrict our attention to two antipodal patches and rescale $g$ so that $\gamma(\theta) = \frac{g^2}{4\pi}$. As explained in Section 4.3, the conductivity $\sigma(\boldsymbol{q} = 0, \omega)$ (which is the primary physical quantity of interest in this paper) can be written as a sum over contributions from different anti-podal patches. Therefore, the two-patch effective action already contains all the important ingredients for transport.

Now let $s = \pm 1$ label the two patches and let $G^\star_{ss}, \Sigma^\star_{ss}, D^\star, \Pi^\star$ denote the saddle point solutions with $S^\star$ the action evaluated at the saddle point. By a simple calculation, one can organize the Gaussian effective action for $\delta G_{ss'}, \delta \Sigma_{ss'}, \delta D, \delta \Pi$ into a matrix multiplication

$$S = S^\star + \frac{1}{2} \begin{pmatrix} \delta\Pi^T & \delta\Sigma^T & \delta D^T & \delta G^T \end{pmatrix} \begin{pmatrix} -\frac{1}{2}P^{(\Pi\Pi)} & 0 & \frac{1}{2} & 0 \\ 0 & P^{(\Sigma\Sigma)} & 0 & -1 \\ \frac{1}{2} & 0 & 0 & -\frac{1}{2}P^{(DG)} \\ 0 & -1 & P^{(GD)} & P^{(GG)} \end{pmatrix} \begin{pmatrix} \delta\Pi \\ \delta\Sigma \\ \delta D \\ \delta G \end{pmatrix}, \quad \text{(D.1)}$$

where we defined various kernels

$$
\begin{aligned}
P^{(\Sigma\Sigma)}_{s_1s_2,s_3s_4}(1234) &= G^\star_{s_1s_3}(13)G^\star_{s_4s_2}(42) \quad P^{(\Pi\Pi)}(1234) = D^\star(13)D^\star(42), \\
P^{(DG)}_{ss'}(1234) &= -g^2 ss' \left[ G^\star_{s's}(21)\delta_{13}\delta_{24} + G^\star_{s's}(12)\delta_{23}\delta_{14} \right], \\
P^{(GD)}_{ss'}(1234) &= \frac{g^2}{2} ss' \left[ G^\star_{ss'}(12)\delta_{23}\delta_{14} + G^\star_{ss'}(12)\delta_{13}\delta_{24} \right], \\
P^{(GG)}_{s_1s_2,s_3s_4}(1234) &= g^2 s_1 s_2 D^\star(21)\delta_{s_1s_3}\delta_{s_2s_4}\delta_{13}\delta_{24}.
\end{aligned}
\quad \text{(D.2)}
$$

In these kernels, 1234 is a short hand for a quadruple of spacetime coordinates and $s_1s_2s_3s_4$ is a quadruple of patch indices. Matrix multiplication involves expressions like $\delta G^T P^{(GG)} \delta G$

which mean $\delta G_{s_2 s_1}(21) P^{(GG)}_{s_1 s_2 s_3 s_4}(1234) \delta G_{s_3 s_4}(34)$ where repeated patch/spacetime indices are summed/integrated over.

By introducing a constant matrix $\Lambda = \mathrm{diag}(-1/2, 1, 1, 1, 1)$ where $-1/2$ acts on the bosonic subspace and $(1, 1, 1, 1)$ acts on the fermionic subspaces with four different choices of $s, s' = \pm 1$, we can remove the spurious factors of $\frac{1}{2}$ in Eq. (D.1) so that

$$S = S^\star + \frac{1}{2} \begin{pmatrix} \delta\Pi^T & \delta\Sigma^T & \delta D^T & \delta G^T \end{pmatrix} (\Lambda \oplus \Lambda) \begin{pmatrix} P^{(\Pi\Pi)} & 0 & -1 & 0 \\ 0 & P^{(\Sigma\Sigma)} & 0 & -1 \\ -1 & 0 & 0 & P^{(DG)} \\ 0 & -1 & P^{(GD)} & P^{(GG)} \end{pmatrix} \begin{pmatrix} \delta\Pi \\ \delta\Sigma \\ \delta D \\ \delta G \end{pmatrix} . \quad (D.3)$$

Here the direct sum $\oplus$ is a sum between the "self-energy" subspace spanned by $\delta\Pi, \delta\Sigma$ and the "Green's function' subspace spanned by $\delta D, \delta G$. Adapting the notation of [61], we bundle together the boson and fermion self energies as $\delta\Xi = (\delta\Pi, \delta\Sigma)$ and the boson + fermion Green's functions as $\delta\mathcal{G} = (\delta D, \delta G)$. We can now write Eq. (D.3) in a cleaner block notation

$$S = S^\star + \frac{1}{2} \begin{pmatrix} \delta\Xi^T & \delta\mathcal{G}^T \end{pmatrix} (\Lambda \oplus \Lambda) \begin{pmatrix} W_\Xi & -I \\ -I & W_\mathcal{G} \end{pmatrix} \begin{pmatrix} \delta\Xi \\ \delta\mathcal{G} \end{pmatrix} , \quad (D.4)$$

where

$$W_\Xi = \begin{pmatrix} P^{(\Pi\Pi)} & 0 \\ 0 & P^{(\Sigma\Sigma)} \end{pmatrix} \qquad W_\mathcal{G} = \begin{pmatrix} 0 & P^{(DG)} \\ P^{(GD)} & P^{(GG)} \end{pmatrix} . \quad (D.5)$$

If one was only interested in the correlation function between different components of $\mathcal{G}$ (e.g. computing the $GG$ correlation function that appears in the Kubo formula for conductivity), one can integrate out the self-energy variables $\Xi$ to get an effective action for $\delta\mathcal{G}$

$$S = S^\star + \frac{1}{2} \delta\mathcal{G}^T \Lambda W_\Xi^{-1} (W_\Xi W_\mathcal{G} - 1) \delta\mathcal{G} = S^\star + \frac{1}{2} \delta\mathcal{G}^T \Lambda W_\Xi^{-1} (K - 1) \delta\mathcal{G} \quad (D.6)$$

where the important kernel $K$ that determines the soft modes is given by

$$K = \begin{pmatrix} P^{(\Pi\Pi)} & 0 \\ 0 & P^{(\Sigma\Sigma)} \end{pmatrix} \begin{pmatrix} 0 & P^{(DG)} \\ P^{(GD)} & P^{(GG)} \end{pmatrix} = \begin{pmatrix} 0 & P^{(\Pi\Pi)} P^{(DG)} \\ P^{(\Sigma\Sigma)} P^{(GD)} & P^{(\Sigma\Sigma)} P^{(GG)} \end{pmatrix} . \quad (D.7)$$

## D.2 Relating the conductivity to fluctuations of collective fields

We now use the formalism described in Appendix D.1 to directly compute the conductivity to leading order in the $1/N$ expansion without leveraging the anomaly. We find that the conductivity is equal to the free-fermion result $\frac{1}{N} \sigma(\boldsymbol{q} = 0, \omega) = \frac{i\mathcal{D}^{(0)}}{\pi\omega}$ independent of the

choice of order parameter form factor $f(\theta)$, in agreement with the anomaly-based arguments in the $N \to \infty$ limit.

For the purpose of this calculation, it is convenient to work with a regularization where the cutoff $\Lambda_\perp$ on $k_x$ is sent to $\infty$ first. In this regularization, the Kubo formula relates the conductivity to the current-current correlator

$$\sigma^{ij}(\boldsymbol{q}, \omega, T) = \frac{1}{i\omega} G^R_{J^i J^j}(\boldsymbol{q}, \omega) = \frac{1}{i\omega} G^E_{J^i J^j}(\boldsymbol{q}, i\omega_n)\big|_{i\omega_n \to \omega + i\epsilon}. \tag{D.8}$$

As explained in Section 4.3, the current two-point function is a sum over contributions from all pairs of anti-podal patches

$$G_{J^i J^j}(\boldsymbol{q} = 0, i\Omega_m) = \sum_{\theta, \theta' \in [0, \pi)} G_{J^i(\theta) J^j(\theta')}(\boldsymbol{q} = 0, i\Omega_m). \tag{D.9}$$

Within the large $N$ expansion, dominant contributions to the low frequency conductivity will come from terms with $\theta = \theta'$ (i.e. the external fermion currents have to live in a pair of anti-podal patches). This observation allows us to drop all the off-diagonal terms and write the full current-current correlation function as

$$G_{J^i J^j}(\boldsymbol{q} = 0, i\Omega_m) = \sum_{\theta \in [0, \pi)} G_{J^i(\theta) J^j(\theta)}(\boldsymbol{q} = 0, i\Omega_m). \tag{D.10}$$

Due to this simplification, we can focus on a particular pair of patches labeled by $s = \pm 1$ corresponding to $\theta, \theta + \pi$ and import the two-patch Gaussian effective action Eq. (D.6).

Within these two patches, we go to a coordinate system where $x, y$ are the directions perpendicular/parallel to the Fermi surface. In terms of the bilocal fields, $J^x(\boldsymbol{x}, \tau) = N \sum_s s G_{ss}(\boldsymbol{x}, \tau, \boldsymbol{x}, \tau)$ and the annealed-average of the current-current correlator is

$$\frac{G^E_{J^x J^x}(\boldsymbol{x}, \tau)}{N^2} = \frac{\int D\{G, \Sigma, D, \Pi\} \sum_{ss'} ss' G_{ss}(\boldsymbol{x}, \tau, \boldsymbol{x}, \tau) G_{s's'}(0, 0, 0, 0) \exp\{-S[G, \Sigma, D, \Pi]\}}{\int D\{G, \Sigma, D, \Pi\} \exp\{-S[G, \Sigma, D, \Pi]\}}. \tag{D.11}$$

The rest of the calculation is a strenuous but conceptually straightforward evaluation of the above functional integral, accounting for the structure of the saddle point as well as the quadratic fluctuations.

To leading order in $N$, $G_{ss}$ can be approximated by the saddle point solution. Therefore

$$G^\star_{ss}(\boldsymbol{x}, \tau, \boldsymbol{x}, \tau) = T \sum_{\omega_n} \int \frac{d^2 k}{(2\pi)^2} G^\star_{ss}(\boldsymbol{k}, i\omega_n) = T \sum_{\omega_n} \int \frac{d^2 k}{(2\pi)^2} \frac{1}{i\omega_n - sk_x - k_y^2 - \Sigma^\star_{ss}(\boldsymbol{k}, i\omega_n)}. \tag{D.12}$$

Since $\Sigma^\star_{ss}$ is independent of $s$, the integral over $k_x$ is independent of $s$ and $G^\star_{++}(\boldsymbol{x},\tau,\boldsymbol{x},\tau) = G^\star_{--}(\boldsymbol{x},\tau,\boldsymbol{x},\tau)$. This immediately shows that the leading $\mathcal{O}(1)$ contribution to $\frac{G^E_{J^x J^x}(\boldsymbol{x},\tau)}{N^2}$ vanishes.

To compute the $\mathcal{O}(1/N)$ corrections[12], it is convenient to first work in Fourier space so that

$$J^x(q)J^x(-q) = \sum_{s_1,s_2} s_1 s_2 T \sum_{\Omega_m} \int \frac{d^2 q}{(2\pi)^2} G_{s_1 s_1}(q_1, q - q_1) G_{s_2 s_2}(q_2, q - q_2), \qquad (D.13)$$

with $q = (\boldsymbol{q}, i\Omega_m)$. The expectation value of this operator follows from the correlators $\langle \delta G_{ss} \delta G_{s's'} \rangle$. To evaluate this correlator, we recall the Gaussian effective action $S_{\text{eff}} = \frac{1}{2} \delta \mathcal{G}^T \Lambda W_\Xi^{-1} (K-1) \delta \mathcal{G}$ which implies a propagator $\langle \delta \mathcal{G}^T \delta \mathcal{G} \rangle = (1-K)^{-1} W_\Xi \Lambda^{-1}$. Since $W_\Xi, \Lambda$ are both diagonal in boson/fermion space, the correlator for $\langle \delta G \delta G \rangle$ is the projection of $\langle \delta \mathcal{G}^T \delta \mathcal{G} \rangle$ onto the fermionic sector

$$\langle \delta G_{s_2 s_1}(21) \delta G_{s_5 s_6}(56) \rangle = \left[ (1-K)^{-1} \right]_{s_1 s_2 s_3 s_4} (1234) P^{(\Sigma\Sigma)}_{s_3 s_4 s_5 s_6}(3456), \qquad (D.14)$$

where summations over intermediate patch indices $s_3, s_4$ and integrations over coordinates $x_3, x_4$ are implied. Using the form of $P^{\Sigma\Sigma}$ in Eq. (D.2) and the fact that $G^\star$ is translation invariant and diagonal in patch space, we have

$$P^{\Sigma\Sigma}_{s_3 s_4 s_5 s_6}(q_3, q_4, q_5, q_6) = \delta_{s_4 s_6} \delta_{s_3 s_5} G^\star_{s_6 s_6}(q_6) G^\star_{s_5 s_5}(-q_5) \delta(q_3 + q_5) \delta(q_4 + q_6). \qquad (D.15)$$

This formula yields a compact momentum space representation of the $\delta G \delta G$ correlator

$$\langle \delta G_{s_2 s_1}(q_2, q_1) \delta G_{s_5 s_6}(q_5, q_6) \rangle = \left[ (1-K)^{-1} \right]_{s_1 s_2 s_5 s_6} (q_1, q_2, -q_5, -q_6) G^\star_{s_5 s_5}(-q_5) G^\star_{s_6 s_6}(q_6), \qquad (D.16)$$

as well as the current-current correlator

$$G^E_{J^x J^x}(\boldsymbol{q} = 0, i\omega) = \langle J^x(\boldsymbol{q} = 0, i\omega) J^x(\boldsymbol{q} = 0, -i\omega) \rangle$$

$$= \sum_{s_1,s_2} s_1 s_2 \int_{\boldsymbol{q_1},\boldsymbol{q_2},\omega_1,\omega_2} \langle \delta G_{s_1 s_1}(-\boldsymbol{q_1}, i\omega - i\omega_1, \boldsymbol{q_1}, i\omega_1) \delta G_{s_2 s_2}(\boldsymbol{q_2}, i\omega_2, -\boldsymbol{q_2}, -i\omega - i\omega_2) \rangle$$

$$= \sum_{s_1,s_2} s_1 s_2 \int_{\boldsymbol{q_1},\boldsymbol{q_2},\omega_1,\omega_2} \sum_{\omega_1,\omega_2} \left[ (1-K)^{-1} \right]_{s_1 s_1 s_2 s_2} (\boldsymbol{q_1}, i\omega_1, -\boldsymbol{q_1}, i\omega - i\omega_1, -\boldsymbol{q_2}, -i\omega_2, \boldsymbol{q_2}, i\omega + i\omega_2)$$

$$\cdot G^\star_{s_2 s_2}(-\boldsymbol{q_2}, -i\omega_2) G^\star_{s_2 s_2}(-\boldsymbol{q_2}, -i\omega - i\omega_2). \qquad (D.17)$$

---

[12]Recall that corrections higher order in $1/N$ are not accessible in the bilocal field description.

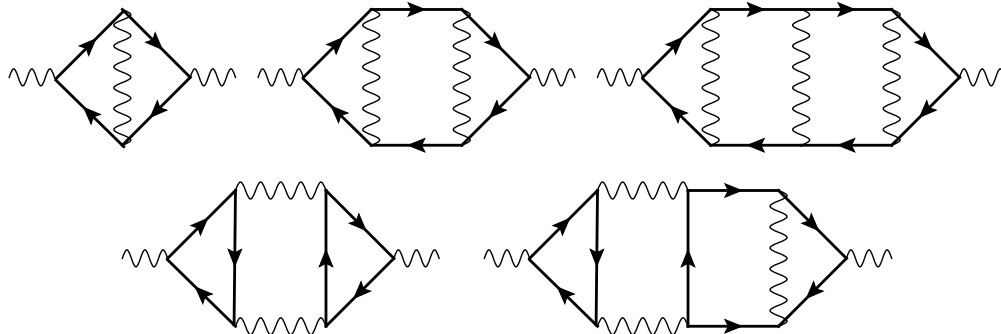

**Figure 4:** A subset of diagrams that are selected by the Gaussian effective action Eq. (D.6) for bilocal field fluctuations to leading order in $1/N$. The top three diagrams are the first three terms in the an infinite series of virtual fermion pair interactions mediated by the boson, while the bottom two diagrams are more exotic diagrams involving two boson to two fermion scattering mediated by a fermion.

Above and throughout the appendix, we use $\int_{\boldsymbol{k},\omega}$ to denote the integration measure $\int \frac{d^2k\,d\omega}{(2\pi)^3}$. To compute the conductivity is to extract the most singular (in $\omega$) contributions to the above integral.

This integral in fact has a simple diagrammatic interpretation. If we expand $(1-K)^{-1}$ as a geometric series, then the $\mathcal{O}(K^0)$ term corresponds to the one-loop bubble diagram with $G^\star$ as the internal fermion propagators. Since $K$ is an operator acting on bilocal fields, all higher order diagrams in this series can be drawn horizontally as a ladder, where each rung is a vertical propagator that connects two incoming particles of the same type with two outgoing particles of the same type. A selection of diagrams that are included/excluded by this geometric series are shown in Figure 4 and Figure 5 respectively. Remarkably, although fermions with different patch indices $s = \pm 1$ can appear in virtual loops, the structure of the kernel $K$ dictates that each factor of $s$ is always raised to an even power. This means that to this order, the correlator $G^E_{J^x J^x}(\boldsymbol{q} = 0, i\omega)$ *does not distinguish between Ising-nematic and loop current order cases.* Therefore, we anticipate that the conductivity in both the loop current order model and the Ising-nematic model should be a Drude peak $\sigma^{ij}(\boldsymbol{q} = 0, \omega) = \frac{iN\mathcal{D}^0_{ij}}{\pi(\omega+i\epsilon)}$ where $\mathcal{D}^0_{ij}$ is the Drude weight of the non-interacting model with $g = 0$.

## D.3 Eigenmodes of the kernel $K$ in the two-patch model

To carry out the ladder resummation we diagonalize the kernel $K$. If the spectrum of $K$ is gapped away from 1, the conductivity scaling would be controlled by the one-loop bubble with zero rung. However, it turns out that there are two eigenmodes of $K$ with eigenvalues

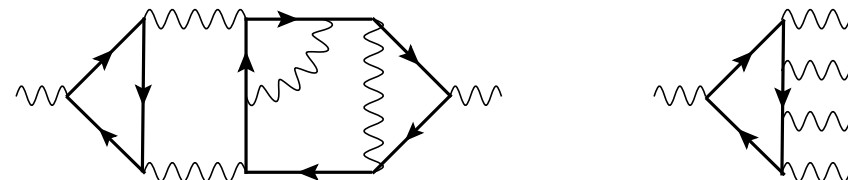

**Figure 5:** Examples of diagrams not captured by the Gaussian effective action Eq. (D.6). Viewed horizontally, virtual processes in these diagrams do not obey the two-in-two-out structure. These diagrams will be suppressed by additional factors of $1/N$ and will *not* be captured by the bilocal fields due to failure of self-averaging at this order.

$1 + \mathcal{O}(\Omega^{1/3})$. As $\Omega \to 0$, these modes give an anomalously large contribution to $(1 - K)^{-1}$ and dominate the conductivity. In what follows, we first find these eigenmodes using the methods of Ref. [61] and then show that the contribution from these modes precisely gives rise to the Drude peak $G^E_{J^x J^x}(0, i\Omega) = -N\mathcal{D}^0_{xx}/\pi$.

Instead of the spacetime coordinates $x_1, x_2$ for the bilocal fields, we work with the 3-momentum $k = (\omega, \boldsymbol{k})$ and the center-of-mass 3-momentum $p = (\Omega, \boldsymbol{p})$. The Fourier transform is defined so that

$$f(k, p) = \int d^3x_1 d^3x_2 e^{i(k+p/2)\cdot x_1 + i(p/2-k)\cdot x_2} f(x_1, x_2). \tag{D.18}$$

From here, one can easily work out the action of $K$ on a general vector $(B(k, p), F_{ss'}(k, p))^T$

$$\begin{pmatrix} \tilde{B}(k, p) \\ \tilde{F}_{ss'}(k, p) \end{pmatrix} = K \begin{pmatrix} B(k, p) \\ F_{ss'}(k, p) \end{pmatrix}. \tag{D.19}$$

For the purpose of the conductivity calculation, since the operator insertions are in the fermionic subspace, it is natural to look for eigenvectors of $K$ with $B = \tilde{B} = 0^{13}$. The full action of the kernel on this subspace takes the simple form

$$\boxed{\tilde{F}_{ss}(k, p) = g^2 G_{ss}(k + \frac{p}{2})G_{ss}(k - \frac{p}{2}) \int \frac{d^3k'}{(2\pi)^3} D(k - k')F_{ss}(k', p).} \tag{D.20}$$

Now recall that under sliding symmetry, the relative momentum $k$ transforms in the fermionic representation $(k_x, k_y) \to (k_x - \theta k_y - \frac{s\theta^2}{4}, k_y + \frac{s\theta}{2})$ while the center-of-mass momentum $p$ transforms in the bosonic representation $(p_x, p_y) \to (p_x - \theta p_y, p_y)$. Since $\tilde{F}_{ss}$ is taken to be sliding-symmetric, it must depend only on the three invariants $p_y, sk_x + k_y^2, p_x + 2sp_y k_y$ and the frequencies. From here, we will separate the spatial momentum and frequency components

---

[13] This is in fact justified since [61] demonstrated numerically that eigenvalues of $K$ outside of this subspace are gapped away from 1.

$k = (\omega, \boldsymbol{k}), p = (\Omega, \boldsymbol{p})$ and consider only the case where $p_x = p_y = 0$, which is relevant for the conductivity subject to a spatially uniform probe

$$F_{ss}(k,p) = F_{ss}(\omega, \Omega, sk_x + k_y^2, p_y, p_x + 2sp_yk_y) = F_{ss}(\omega, \Omega, sk_x + k_y^2)\delta^2(\boldsymbol{p}). \qquad \text{(D.21)}$$

Using the above ansatz, and working in the k-first regularization, we can make a change of variables $\tilde{k}_x = sk_x + k_y^2$ and simplify the action of $K$ as

$$\tilde{F}_{ss}(k,p) = g^2G_{ss}(\omega + \frac{\Omega}{2}, \boldsymbol{k})G_{ss}(\omega - \frac{\Omega}{2}, \boldsymbol{k})\int \frac{d\omega' d\tilde{k}_x}{(2\pi)^2}\frac{dk_y'}{2\pi}\frac{F_{ss}(\tilde{k}_x, \omega', \Omega)\delta^2(\boldsymbol{p})}{(k_y - k_y')^2 + \frac{g^2|\omega-\omega'|}{4\pi|k_y-k_y'|}}$$

$$= g^2G_{ss}(\omega + \frac{\Omega}{2}, \boldsymbol{k})G_{ss}(\omega - \frac{\Omega}{2}, \boldsymbol{k})\int \frac{d\omega' d\tilde{k}_x}{(2\pi)^2}\frac{2^{5/3}\pi^{1/3}}{3\sqrt{3}g^{2/3}|\omega-\omega'|^{1/3}}F_{ss}(\tilde{k}_x, \omega', \Omega)\delta^2(\boldsymbol{p}).$$
$$\text{(D.22)}$$

The LHS depends on $\boldsymbol{k}$ only through the product of two $G$'s. Thus, if $F_{ss}$ were a unit eigenvector, it must also depend on the product of two $G$'s with some additional frequency-dependent factors to compensate for the integral over the internal boson propagator. The correct structure turns out to be

$$F_{ss}(k,p) = if_s[G_{ss}(\omega - \frac{\Omega}{2}, \boldsymbol{k}) - G_{ss}(\omega + \frac{\Omega}{2}, \boldsymbol{k})]\delta^2(\boldsymbol{p}). \qquad \text{(D.23)}$$

It is not difficult to check by doing the residue integral over $\tilde{k}_x = sk_x + k_y^2$ that

$$\tilde{F}_{ss}(k,p) = g^2G_{ss}(k + p/2)G_{ss}(k - p/2)(-f_s)\frac{3}{2}[(\omega + \frac{\Omega}{2})^{2/3} + (-\omega + \frac{\Omega}{2})^{2/3}]\frac{2^{5/3}}{6\sqrt{3}g^{2/3}\pi^{2/3}}$$

$$= G_{ss}(k + p/2)G_{ss}(k - p/2)(-f_s)g^{4/3}\frac{(\omega + \frac{\Omega}{2})^{2/3} + (-\omega + \frac{\Omega}{2})^{2/3}}{2^{1/3}\sqrt{3}\pi^{2/3}}$$

$$= G_{ss}(k + p/2)G_{ss}(k - p/2)if_s[\Sigma_{ss}(\omega - \frac{\Omega}{2}) - \Sigma_{ss}(\omega + \frac{\Omega}{2})],$$
$$\text{(D.24)}$$

where in the last step we used $\Sigma_{ss}(\omega) = -i\text{sgn}(\omega)\frac{g^{4/3}|\omega|^{2/3}}{\pi^{2/3}2^{1/3}\sqrt{3}}$. Now comparing $F_{ss}(k,p)$ with $\tilde{F}_{ss}(k,p)$, we see that

$$\tilde{F}_{ss}(k,p) = \frac{\Sigma(\omega - \frac{\Omega}{2}) - \Sigma(\omega + \frac{\Omega}{2})}{i\Omega + \Sigma(\omega - \frac{\Omega}{2}) - \Sigma(\omega + \frac{\Omega}{2})}F_{ss}(k,p). \qquad \text{(D.25)}$$

Therefore, in the limit $|\Omega| \to 0$, $F_{ss}(k,p)$ is indeed a unit eigenvalue.

## D.4   Contributions to the conductivity from the near-unit eigenvalue of $K$

In the limit of small external frequency $|\Omega| \ll g$, the ladder sum $(1 - K)^{-1}$ is dominated by the eigenmodes of $K$ with eigenvalues closest to 1. In the previous section, we have identified

these eigenmodes when $\Omega \to 0$. In this section, we first do a simple 1st order perturbation theory to compute the shift of these eigenvalues away from 1 in the limit $\Omega \ll g$. Then we demonstrate that taking only these modes into account precisely recover the Drude weight.

To leading order in $\Omega$, the eigenvectors do not shift and the eigenvalue shift can be computed by taking the expectation value

$$\langle F_{ss}|F_{ss}\rangle = \int_{\boldsymbol{k},\omega}[\frac{1}{i(\omega - \frac{\Omega}{2}) - k_x - k_y^2 - \Sigma(\omega - \frac{\Omega}{2})} - \frac{1}{i(\omega + \frac{\Omega}{2}) - k_x - k_y^2 - \Sigma(\omega + \frac{\Omega}{2})}]$$
$$\cdot [\frac{1}{-i(\omega - \frac{\Omega}{2}) - k_x - k_y^2 + \Sigma(\omega - \frac{\Omega}{2})} - \frac{1}{-i(\omega + \frac{\Omega}{2}) - k_x - k_y^2 + \Sigma(\omega + \frac{\Omega}{2})}] .$$
$$(\text{D.26})$$

There are a total of four terms. Two of them consists of products of propagators with matching frequencies. The total contribution is

$$4\pi\Lambda_y \int_0^\infty \frac{d\omega}{\omega + C\omega^{2/3}} , \tag{D.27}$$

where $\Lambda_y$ is the momentum cutoff within a single patch in the direction paralell to the Fermi surface. The other two consists of products of propagators with frequencies that differ by $\Omega$. The total contribution is

$$- 4\pi\Lambda_y \int_0^\infty \frac{\text{sgn}(\omega + \frac{\Omega}{2}) + \text{sgn}(\omega - \frac{\Omega}{2})}{2\omega + C\text{sgn}(\omega + \frac{\Omega}{2})|\omega + \frac{\Omega}{2}|^{2/3} + C\text{sgn}(\omega - \frac{\Omega}{2})|\omega - \frac{\Omega}{2}|^{2/3}} . \tag{D.28}$$

Each of these terms is individually UV divergent. But when we add them together, the UV divergences cancel out and we are left with a finite answer dominated by the small $\omega$ part of the integration domain. This means we can drop linear in $\omega$ terms in the denominator (for safety, we have checked that this works numerically). Therefore, to leading order in $\Omega$, we find

$$\langle F_{ss}|F_{ss}\rangle = \frac{4\pi\Lambda_y}{C}(\frac{\Omega}{2})^{1/3}\left(3 - \int_1^\infty (\frac{2}{(x-1)^{2/3} + (x+1)^{2/3}} - \frac{2}{2x^{2/3}})dx\right)$$
$$= \frac{4\pi\Lambda_y}{C}(\frac{\Omega}{2})^{1/3}I_0 .$$
$$(\text{D.29})$$

Similarly, we can compute the eigenvalue correction by recalling that

$$\langle \boldsymbol{k},\omega,\boldsymbol{p}=0,\Omega|\,(K-1)\,|F_{ss}\rangle = \frac{\Omega\,\langle \boldsymbol{k},\omega,\boldsymbol{p}=0,\Omega|F_{ss}\rangle}{C[\text{sgn}(\omega - \frac{\Omega}{2})|\omega - \frac{\Omega}{2}|^{2/3} - \text{sgn}(\omega + \frac{\Omega}{2})|\omega + \frac{\Omega}{2}|^{2/3}]} . \tag{D.30}$$

This means that

$$\langle F_{ss}| (K-1) |F_{ss}\rangle = \frac{4\pi\Lambda_y\Omega}{C}\int_0^\infty d\omega\frac{1}{\text{sgn}(\omega-\frac{\Omega}{2})|\omega-\frac{\Omega}{2}|^{2/3}-\text{sgn}(\omega+\frac{\Omega}{2})|\omega+\frac{\Omega}{2}|^{2/3}}$$
$$\left[\frac{1}{\omega+C\omega^{2/3}}-\frac{\text{sgn}(\omega+\frac{\Omega}{2})+\text{sgn}(\omega-\frac{\Omega}{2})}{2\omega+C\text{sgn}(\omega+\frac{\Omega}{2})|\omega+\frac{\Omega}{2}|^{2/3}+C\text{sgn}(\omega-\frac{\Omega}{2})|\omega-\frac{\Omega}{2}|^{2/3}}\right].$$
(D.31)

Again the UV divergences of various terms cancel and the dominant contribution at small $\Omega$ comes from the integration region $\omega \sim \Omega$. This justifies dropping linear in $\omega$ terms in the denominator. Thus we find

$$\langle F_{ss}| (K-1) |F_{ss}\rangle = \frac{4\pi\Lambda_y\Omega}{C^2}(\frac{\Omega}{2})^{-1/3}\left[\int_0^\infty dx\frac{1}{x^{2/3}[\text{sgn}(x-1)|x-1|^{2/3}-\text{sgn}(x+1)|x+1|^{2/3}]}\right.$$
$$\left.-\int_1^\infty \frac{2}{(x+1)^{2/3}+(x-1)^{2/3}}\frac{1}{-(x+1)^{2/3}+(x-1)^{2/3}}\right]$$
$$=\frac{4\pi\Lambda_y\Omega}{C^2}(\frac{\Omega}{2})^{-1/3}I_K.$$
(D.32)

Therefore, we find that the normalized eigenvalue shift is

$$\delta k = \frac{4\pi\Lambda_y\Omega}{C^2}(\frac{\Omega}{2})^{-1/3}I_K \cdot \frac{C}{4\pi\Lambda_y I_0(\frac{\Omega}{2})^{1/3}} = \frac{\Omega^{1/3}}{C}\frac{2^{2/3}I_K}{I_0}.$$
(D.33)

Now let us compute the leading in $\Omega$ contribution to the ladder sum. Recall that

$$\frac{G^E(0,i\Omega)}{-v_F N}=\sum_s\int_{\boldsymbol{q_1},\boldsymbol{q_2},\omega_1,\omega_2}\frac{\langle\boldsymbol{q_1},\omega_1,0,\Omega|F_{ss}\rangle\,\langle F_{ss}|\boldsymbol{q_2},\omega_2,0,\Omega\rangle\,G^\star_{ss}(\boldsymbol{q_2},\omega_2+\frac{\Omega}{2})G^\star_{ss}(\boldsymbol{q_2},\omega_2-\frac{\Omega}{2})}{(-\delta k)\,\langle F_{ss}|F_{ss}\rangle}.$$
(D.34)

Since the integral factorizes, we can evaluate the integral over $\boldsymbol{q_1},\omega_1$ first:

$$\int_{\boldsymbol{q_1},\omega_1}\langle\boldsymbol{q_1},\omega_1,\boldsymbol{p}=0,\Omega|F_{ss}\rangle=\int_{\boldsymbol{q_1},\omega_1}[G^\star_{ss}(\boldsymbol{q_1},\omega_1-\frac{\Omega}{2})-G^\star_{ss}(\boldsymbol{q_1},\omega_1+\frac{\Omega}{2})]$$
$$=\frac{\Lambda_y}{2\pi}\int\frac{d\omega_1}{2\pi}\frac{2\pi i}{2\pi}[\frac{\text{sgn}(\omega_1+\frac{\Omega}{2})-\text{sgn}(\omega_1-\frac{\Omega}{2})}{2}]$$
$$=\frac{i\Lambda_y\Omega}{4\pi^2}.$$
(D.35)

As for the integral over $\boldsymbol{q_2},\omega_2$, we first make the observation that

$$G^\star_{ss}(\boldsymbol{q_2},\omega_2+\frac{\Omega}{2})G^\star_{ss}(\boldsymbol{q_2},\omega_2-\frac{\Omega}{2})[i\Omega-\Sigma_{ss}(\omega_2+\frac{\Omega}{2}+\Sigma_{ss}(\omega-\frac{\Omega}{2})]$$
$$=G^\star_{ss}(\boldsymbol{q_2},\omega_2-\frac{\Omega}{2})-G^\star_{ss}(\boldsymbol{q_2},\omega_2+\frac{\Omega}{2}).\quad\text{(D.36)}$$

Hence, the integral over $\boldsymbol{q_2}, \omega_2$ simplifies to

$$I = \int_{\boldsymbol{q_2},\omega_2} \frac{\langle F_{ss}|\boldsymbol{q_2},\omega_2,0,\Omega\rangle \left[i\Omega - \Sigma_{ss}(\omega_2 + \frac{\Omega}{2}) + \Sigma_{ss}(\omega - \frac{\Omega}{2})\right]^{-1} \langle \boldsymbol{q_2},\omega_2,0,\Omega|F_{ss}\rangle}{\langle F_{ss}|F_{ss}\rangle} \, . \qquad (D.37)$$

We know that the $\omega_2$ integral is dominated by $\omega_2 \sim \Omega$. Thus, for $\Omega$ sufficiently small, the $i\Omega$ term can be dropped. After dropping $i\Omega$, the remaining integral is directly related to the computation of $\delta k$ as $\delta k \approx -i\Omega I$. Using this relationship, we immediately see that

$$G^E(\boldsymbol{q} = 0, i\Omega) = -v_F N \sum_s \frac{i\Lambda_y \Omega}{4\pi^2} \cdot \frac{1}{-\delta k} \cdot \frac{\delta k}{-i\Omega} = -\frac{N v_F \Lambda_y}{2\pi^2} \, . \qquad (D.38)$$

Now we simply have to sum over all pairs of patches. Within each patch, $J^x$ should be identified with $v_F(\theta)w^i(\theta)n_\theta + v_F(\theta + \pi)w^i(\theta + \pi)n_{\theta+\pi}$ and $\Lambda_y$ should be replaced with a patch-dependent cutoff $\Lambda(\theta)$. After summing over all patches and using inversion symmetry, we recover the Drude weight of the non-interacting model

$$G^E_{J^i J^j}(\boldsymbol{q} = \boldsymbol{0}, i\Omega) = -N \sum_\theta \frac{\Lambda(\theta)}{(2\pi)^2} v_F(\theta)w^i(\theta)w^j(\theta) = -\frac{N\mathcal{D}_0^{ij}}{\pi} \, . \qquad (D.39)$$

# E   Technical aspects of the memory matrix approach

## E.1   Eigenvalues of the decay rate matrix $\tau^{-1} = \chi^{-1}M$

From Section 4.4, we learned that the spectrum of $\chi^{-1}M$ contains $N$ nonzero eigenvalues $\lambda$ determined by the condition

$$\det\left(\lambda\delta_{IJ} - \chi_{\phi\phi}^{-1}M_{IJ} - \frac{1}{N}\bar{F}^2 G_I G_K M_{KJ}\right) = 0 \, . \qquad (E.1)$$

To solve for the eigenvalues, it is convenient to introduce a bra-ket notation in the boson subspace where $|\mathcal{G}\rangle$ represents the normalized vector with coefficients

$$\langle K|\mathcal{G}\rangle = \frac{G_K}{\sum_K G_K^2} \, . \qquad (E.2)$$

In this notation, the eigenvalue condition can be rewritten as

$$\det\left(\lambda - \chi_{\phi\phi}^{-1}M - \frac{1}{N}\bar{F}^2\bar{G}^2 |\mathcal{G}\rangle\langle\mathcal{G}| M\right) = 0 \, . \qquad (E.3)$$

As $\chi_{\phi\phi} \to \infty$, $\left(\chi_{\phi\phi}^{-1} + \frac{\bar{G}^2}{N}\bar{F}^2 |\mathcal{G}\rangle\langle\mathcal{G}|\right)$ approaches a rank-one matrix whose only nontrivial eigenvector is $|\mathcal{G}\rangle$. Since $M$ is full rank, $\left(\chi_{\phi\phi}^{-1} + \frac{\bar{G}^2}{N}\bar{F}^2 |\mathcal{G}\rangle\langle\mathcal{G}|\right) M$ is also rank-one as $\chi \to \infty$.

Therefore, for $\chi_{\phi\phi}$ finite but large, we expect $\left(\chi_{\phi\phi}^{-1} + \frac{\bar{G}^2}{N}\bar{F}^2 \left|\mathcal{G}\right\rangle\left\langle\mathcal{G}\right|\right)M$ to have one nontrivial eigenvector almost parallel to $\left|\mathcal{G}\right\rangle$ and $N-1$ eigenvectors almost orthogonal to $\left|\mathcal{G}\right\rangle$. To check this more explicitly, we can decompose an arbitrary vector in the boson subspace as $\alpha\left|\mathcal{G}\right\rangle + \beta\left|\mathcal{W}\right\rangle$ with $\langle\mathcal{G}|\mathcal{W}\rangle = 0$ and rewrite the eigenvalue equation as

$$\left(\alpha\chi_{\phi\phi}^{-1} + \frac{\alpha\bar{G}^2}{N}\bar{F}^2\right)\left|\mathcal{G}\right\rangle + \beta\chi_{\phi\phi}^{-1}\left|\mathcal{W}\right\rangle = \lambda M^{-1}\left(\alpha\left|\mathcal{G}\right\rangle + \beta\left|\mathcal{W}\right\rangle\right). \tag{E.4}$$

Assuming the invertibility of $(1 - \lambda\chi_{\phi\phi}M^{-1})$, this is equivalent to

$$\left|\mathcal{W}\right\rangle = -\alpha\beta^{-1}\left(1 - \lambda\chi_{\phi\phi}M^{-1}\right)^{-1}\left(1 - \lambda\chi_{\phi\phi}M^{-1} + \frac{\bar{G}^2\chi_{\phi\phi}}{N}\bar{F}^2\right)\left|\mathcal{G}\right\rangle \tag{E.5}$$

$$= -\alpha\beta^{-1}\left|\mathcal{G}\right\rangle - \alpha\beta^{-1}\left(1 - \lambda\chi_{\phi\phi}M^{-1}\right)^{-1}\frac{\bar{G}^2\chi_{\phi\phi}}{N}\bar{F}^2\left|\mathcal{G}\right\rangle. \tag{E.6}$$

1. If $\lambda$ approaches a nonzero finite value as $\chi \to \infty$, then

$$\left|\mathcal{W}\right\rangle \to \frac{1}{\beta}\left(\bar{F}^2\frac{\bar{G}^2 M\lambda^{-1}}{N} - 1\right)\left|\mathcal{G}\right\rangle \tag{E.7}$$

   subject to the constraint

$$\langle\mathcal{G}|\mathcal{W}\rangle \propto \bar{F}^2\bar{G}^2\lambda^{-1}\left\langle\mathcal{G}\right|M\left|\mathcal{G}\right\rangle - N = 0 \quad\to\quad \lambda = \bar{F}^2\frac{\bar{G}^2\left\langle\mathcal{G}\right|M\left|\mathcal{G}\right\rangle}{N}. \tag{E.8}$$

2. If $\lambda$ approaches zero as $\chi \to \infty$, then any $\left|\mathcal{W}\right\rangle$ that is orthogonal to $\left|\mathcal{G}\right\rangle$ can be constructed by carefully tuning $\lambda\chi$ to be close to eigenvalues of $M$. These generate the $N-1$ remaining eigenvalues which scale as $\lambda \sim \chi_{\phi\phi}^{-1}$.

Since $\left|\mathcal{G}\right\rangle$ is a normalized vector, $\left\langle\mathcal{G}\right|M\left|\mathcal{G}\right\rangle$ does not scale with $N$, assuming that the memory matrix components $M_{IJ}(\omega, T)$ are at most $\mathcal{O}(1)$ (which can be checked a posteriori). Therefore, the only nonzero eigenvalues satisfy $\lambda = \mathcal{O}(\frac{1}{N})$ or $\lambda = \mathcal{O}(\chi_{\phi\phi}^{-1})$, which is the result stated in Section 4.4.

## E.2 Matrix algebra in the decomposition of $\sigma(\omega)$ into coherent and incoherent parts

Here we evaluate the matrix inverse $(\chi + \mathcal{M})_{\theta\theta'}^{-1}$ en route to the conductivity formula. Recall that

$$\chi + \mathcal{M} = \mathcal{L}\begin{pmatrix} A & B \\ B^T & C \end{pmatrix}\mathcal{L}, \tag{E.9}$$

where the subblocks are defined as

$$\mathcal{L}_{\theta,I;\theta',J} = \begin{pmatrix} L(\theta)\delta_{\theta\theta'} & 0 \\ 0 & \delta_{IJ} \end{pmatrix}, \tag{E.10}$$

$$A_{\theta\theta'} = N\delta_{\theta\theta'} + F(\theta)F(\theta')\bar{G}^2\chi_{\phi\phi}, \quad B_{\theta,L} = F(\theta)G_L\chi_{\phi\phi}, \quad C_{KL} = \chi_{\phi\phi}\delta_{KL} + \mathcal{M}_{KL}. \tag{E.11}$$

Using the explicit inversion formula for $2 \times 2$ block matrices, we immediately deduce that

$$(\chi + \mathcal{M})_{\theta\theta'}^{-1} = L(\theta)^{-1}(A - BC^{-1}B^T)_{\theta\theta'}^{-1}L(\theta')^{-1}. \tag{E.12}$$

To compute these inverses, it is helpful to introduce normalized vectors in the slow subspace $|\mathcal{F}\rangle, |\mathcal{G}\rangle$ such that

$$\langle\theta|\mathcal{F}\rangle = F(\theta)/\bar{F}, \quad \langle I|\mathcal{F}\rangle = 0, \quad \langle\theta|\mathcal{G}\rangle = 0, \quad \langle I|\mathcal{G}\rangle = G_I/\bar{G}. \tag{E.13}$$

In this bra-ket notation, we can then write

$$A = N\mathbb{I} + \chi_{\phi\phi}\bar{G}^2\bar{F}^2\,|\mathcal{F}\rangle\langle\mathcal{F}|, \quad B = \chi_{\phi\phi}\bar{G}\bar{F}\,|\mathcal{F}\rangle\langle\mathcal{G}|, \quad C = \chi_{\phi\phi}\mathbb{I} + \mathcal{M}. \tag{E.14}$$

Using the fact that $|\mathcal{F}\rangle, |\mathcal{G}\rangle$ are normalized, we can easily compute

$$\begin{aligned} A - BC^{-1}B^T &= N\mathbb{I} + \chi_{\phi\phi}\bar{G}^2\bar{F}^2\,|\mathcal{F}\rangle\langle\mathcal{F}| - \chi_{\phi\phi}\bar{G}\bar{F}\,|\mathcal{F}\rangle\langle\mathcal{G}|\,(\chi_{\phi\phi}\mathbb{I} + \mathcal{M})^{-1}\chi_{\phi\phi}\bar{G}\bar{F}\,|\mathcal{G}\rangle\langle\mathcal{F}| \\ &= N\mathbb{I} + \bar{G}^2\bar{F}^2\,\langle\mathcal{G}|\,\mathcal{M}(\mathbb{I} + \chi_{\phi\phi}^{-1}\mathcal{M})^{-1}\,|\mathcal{G}\rangle\,|\mathcal{F}\rangle\langle\mathcal{F}|, \end{aligned} \tag{E.15}$$

$$(A - BC^{-1}B^T)^{-1} = N^{-1}\left[\mathbb{I} - \frac{\bar{G}^2\bar{F}^2\,\langle\mathcal{G}|\,\mathcal{M}(\mathbb{I} + \chi_{\phi\phi}^{-1}\mathcal{M})^{-1}\,|\mathcal{G}\rangle}{N + \bar{G}^2\bar{F}^2\,\langle\mathcal{G}|\,\mathcal{M}(\mathbb{I} + \chi_{\phi\phi}^{-1}\mathcal{M})^{-1}\,|\mathcal{G}\rangle}\,|\mathcal{F}\rangle\langle\mathcal{F}|\right]. \tag{E.16}$$

This formula immediately implies

$$(\chi + \mathcal{M})_{\theta\theta'}^{-1} = N^{-1}\left[\frac{\delta_{\theta\theta'}}{L(\theta)L(\theta')} - \frac{\bar{G}^2\,\langle\mathcal{G}|\,\mathcal{M}(\mathbb{I} + \chi_{\phi\phi}^{-1}\mathcal{M})^{-1}\,|\mathcal{G}\rangle}{N + \bar{G}^2\bar{F}^2\,\langle\mathcal{G}|\,\mathcal{M}(\mathbb{I} + \chi_{\phi\phi}^{-1}\mathcal{M})^{-1}\,|\mathcal{G}\rangle}\frac{F(\theta)F(\theta')}{L(\theta)L(\theta')}\right]. \tag{E.17}$$

Using the above expression in the generalized Drude form and summing over repeated patch indices, we find the conductivity for finite $\chi_{\phi\phi}$ which matches results in Section 4.4 upon

taking $\chi_{\phi\phi} \to \infty$

$$
\begin{aligned}
\sigma^{ij}(\omega) &= \frac{i}{\omega}\chi_{J^i n_\theta}(\chi + \mathcal{M})^{-1}_{\theta\theta'}\chi_{J^j n_{\theta'}} \\
&= \frac{i}{\omega}N v_F^i(\theta)\,L(\theta)^2\, v_F^j(\theta')\,L(\theta')^2 \\
&\quad \times \left[ \frac{\delta_{\theta\theta'}}{L(\theta)L(\theta')} - \frac{\bar{G}^2\bar{F}^2\,\langle\mathcal{G}|\,\mathcal{M}(\mathbb{I}+\chi_{\phi\phi}^{-1}\mathcal{M})^{-1}\,|\mathcal{G}\rangle}{N+\bar{G}^2\bar{F}^2\,\langle\mathcal{G}|\,\mathcal{M}(\mathbb{I}+\chi_{\phi\phi}^{-1}\mathcal{M})^{-1}\,|\mathcal{G}\rangle}\frac{F(\theta)F(\theta')}{\bar{F}^2 L(\theta)L(\theta')} \right] \\
&= \frac{i}{\omega}N\left( \mathrm{Tr}_\theta\left[v_F^i v_F^j\right] - \frac{\mathrm{Tr}_\theta\left[v_F^i f\right]\mathrm{Tr}_\theta\left[v_F^j f\right]}{\mathrm{Tr}_\theta\left[f^2\right]} \right) \\
&\quad + \frac{i}{\omega}N^2\frac{\mathrm{Tr}_\theta\left[v_F^i f\right]\mathrm{Tr}_\theta\left[v_F^j f\right]}{\mathrm{Tr}_\theta\left[f^2\right]\left(N+\bar{G}^2\,\mathrm{Tr}_\theta\left[f^2\right]\langle\mathcal{G}|\,\mathcal{M}(\mathbb{I}+\chi_{\phi\phi}^{-1}\mathcal{M})^{-1}\,|\mathcal{G}\rangle\right)}\,.
\end{aligned}
\tag{E.18}
$$

## E.3  Matrix algebra in the evaluation of the memory matrix

Following the strategy put forth in Section 4.4, we need to evaluate the correlation function $\mathcal{C}_{\phi_I\phi_J}(\omega, T)$ using the memory matrix formalism. For this calculation, it is more convenient to make a change of basis from $\{\tilde{n}_\theta, \phi_I\}$ to $\{n_\theta, \phi_I\}$ so that the susceptibility matrix $\chi$ is block diagonal

$$
\chi = \begin{pmatrix} L(\theta)^2\delta_{\theta\theta'} & 0 \\ 0 & \chi_{\phi\phi}\delta_{IJ} \end{pmatrix}\,.
\tag{E.19}
$$

Using the anomaly equation

$$
\frac{dn_\theta}{dt} = -G_I F(\theta)L(\theta)\frac{d\phi_I}{dt}\,,
\tag{E.20}
$$

one can easily show that

$$
\mathcal{M}_{\theta,I;\theta',J} = \begin{pmatrix} G_I\mathcal{M}_{IJ}G_J F(\theta)L(\theta)F(\theta')L(\theta') & G_I\mathcal{M}_{IJ}F(\theta)L(\theta) \\ F(\theta')L(\theta')\mathcal{M}_{IJ}G_J & \mathcal{M}_{IJ} \end{pmatrix}\,.
\tag{E.21}
$$

Therefore, we can again write

$$
\chi + \mathcal{M} = \mathcal{L}\begin{pmatrix} A & B \\ B^T & C \end{pmatrix}\mathcal{L}\,.
\tag{E.22}
$$

where $\mathcal{L}$ takes the same form as before while

$$
A = N\mathbb{I} + \bar{F}^2\bar{G}^2\,\langle\mathcal{G}|\,\mathcal{M}\,|\mathcal{G}\rangle\,|\mathcal{F}\rangle\,\langle\mathcal{F}|\,, \quad B = \bar{F}\bar{G}\,|\mathcal{F}\rangle\,\langle\mathcal{G}|\,\mathcal{M}\,, \quad C = \chi_{\phi\phi}\mathbb{I} + \mathcal{M}\,.
\tag{E.23}
$$

The advantage of this choice of basis is that the response function now involves only a projection onto the boson subspace

$$
\mathcal{C}_{\phi_I\phi_J}(\omega, T) = -\sum_{A,B\in\mathcal{S}}\chi_{\phi_I A}\,[\chi+\mathcal{M}]^{-1}_{AB}\,\chi_{\phi_J B} = -\chi_{\phi\phi}^2\,[\chi+\mathcal{M}]^{-1}_{IJ}\,.
\tag{E.24}
$$

To evaluate $(\chi + \mathcal{M})^{-1}_{IJ} = C_{IJ} - (B^T A^{-1} B)_{IJ}$, we simply need

$$
\begin{aligned}
(B^T A^{-1} B)_{IJ} &= \frac{\bar{F}^2 \bar{G}^2}{N} \langle I | \mathcal{M} | \mathcal{G} \rangle \langle \mathcal{F} | \left[ \mathbb{I} - \frac{\bar{F}^2 \bar{G}^2 \langle \mathcal{G} | \mathcal{M} | \mathcal{G} \rangle}{N + \bar{F}^2 \bar{G}^2 \langle \mathcal{G} | \mathcal{M} | \mathcal{G} \rangle} | \mathcal{F} \rangle \langle \mathcal{F} | \right] | \mathcal{F} \rangle \langle \mathcal{G} | \mathcal{M} | J \rangle \\
&= \frac{\bar{G}^2 \bar{F}^2 \langle I | \mathcal{M} | \mathcal{G} \rangle \langle \mathcal{G} | \mathcal{M} | J \rangle}{N + \bar{F}^2 \bar{G}^2 \langle \mathcal{G} | \mathcal{M} | \mathcal{G} \rangle} \; .
\end{aligned}
$$
(E.25)

Matching with the Eq. (4.35) and using $D_{IJ}$ to denote the matrix of boson Green's functions $G^R_{\phi_I \phi_J}(\boldsymbol{q} = 0, \omega, T)$, we obtain a nontrivial matrix identity

$$
(\chi_{\phi\phi} - D) \left( \chi_{\phi\phi} + \mathcal{M} - \frac{\bar{G}^2 \bar{F}^2 \mathcal{M} | \mathcal{G} \rangle \langle \mathcal{G} | \mathcal{M}}{N + \bar{F}^2 \bar{G}^2 \langle \mathcal{G} | \mathcal{M} | \mathcal{G} \rangle} \right) = \chi^2_{\phi\phi} \; .
$$
(E.26)

Deep in the IR limit, the boson self energy matrix $\Pi$ is related to the boson Green's function matrix via $D = -\Pi^{-1}$. Multiplying the identity above by $\Pi$ and then taking an expectation value in $| \mathcal{G} \rangle$, we find

$$
\frac{N \chi_{\phi\phi} \langle \mathcal{G} | \Pi \mathcal{M} | \mathcal{G} \rangle + N \langle \mathcal{G} | \mathcal{M} | \mathcal{G} \rangle}{N + \bar{F}^2 \bar{G}^2 \langle \mathcal{G} | \mathcal{M} | \mathcal{G} \rangle} + \chi_{\phi\phi} = 0 \; .
$$
(E.27)

In the $\chi_{\phi\phi} \to \infty$ limit, this identity reduces to the identity quoted in the main text

$$
N \langle \mathcal{G} | \Pi \mathcal{M} | \mathcal{G} \rangle - \bar{F}^2 \bar{G}^2 \langle \mathcal{G} | \mathcal{M} | \mathcal{G} \rangle = -N \; .
$$
(E.28)

# F   RPA calculation of the boson self energy and the conductivity

Following the notation of Ref. [64], we consider a UV Euclidean action describing N species of fermions with a generic inversion-symmetric dispersion $\epsilon(\boldsymbol{k})$ coupled to some order parameter field $\phi^a$

$$
S = S_\phi + S_\psi + S_{\text{int}} \; ,
$$
(F.1)

where the kinetic terms and interaction terms are given by

$$
S_\psi = \int \psi_i^\dagger(\boldsymbol{k}, \tau) \left[ \partial_\tau + \epsilon(\boldsymbol{k}) \right] \psi_i(\boldsymbol{k}, \tau) \quad S_\phi = \frac{1}{2} \int \phi^a(\boldsymbol{q}, \tau) [-\partial_\tau^2 + |q|^{z-1}] \phi^a(-\boldsymbol{q}, \tau) \; , \quad \text{(F.2)}
$$

$$
S_{\text{int}} = \frac{1}{\sqrt{N_f}} \int f^a(\boldsymbol{k}, \hat{\boldsymbol{q}}) \cdot \phi^a(\boldsymbol{q}) \psi_i^\dagger(\boldsymbol{k} + \frac{\boldsymbol{q}}{2}) \psi_i(\boldsymbol{k} - \frac{\boldsymbol{q}}{2}) \; .
$$
(F.3)

The fermion flavor index $i$ runs from $1 \sim N_f$ and the boson flavor index $a$ runs from $1 \sim N_b$. Within RPA, we keep $N_b$ finite and perform a $1/N_f$ expansion. The fermion propagator remains free to leading order in the $1/N_f$ expansion

$$
G(k, i\omega) = \frac{1}{i\omega - \epsilon(\boldsymbol{k})} \; ,
$$
(F.4)

while the effective propagator for the boson field takes the Landau damping form

$$D_{ab}(\boldsymbol{q}, i\nu) = \left[ |\boldsymbol{q}|^{z-1} + \boldsymbol{\Gamma} \frac{|\nu|}{|\boldsymbol{q}|} \right]_{ab}^{-1}, \tag{F.5}$$

where $\boldsymbol{\Gamma}$ is a constant matrix with non-vanishing off-diagonal components to leading order in $\nu/|\boldsymbol{q}|$. Since the conclusions we draw later on will only depend on the transformation law of $f^a(\boldsymbol{k}, \boldsymbol{q})$ under inversion, we restrict our attention to the simplest family of models where $N_b = 1$ and $f^a(\boldsymbol{k}, \boldsymbol{q}) = f(\boldsymbol{k}, \boldsymbol{q})$. In that case, the boson propagator reduces to

$$D(\boldsymbol{q}, i\nu) = \frac{1}{|\boldsymbol{q}|^{z-1} + \gamma_{\hat{q}} \frac{|\nu|}{|\boldsymbol{q}|}}, \tag{F.6}$$

where $\gamma_{\hat{q}}$ is now a scalar that depends on the orientation of $\hat{q}$.

The original calculation in Ref. [64] corresponds to taking $f(\boldsymbol{k}, \boldsymbol{q}) = \vec{v}_F(\boldsymbol{k}) \times \hat{q}$ and taking $\phi(\boldsymbol{q})$ to be the transverse component of the gauge field (the longitudinal component is screened). The standard Ising-nematic setup corresponds to taking $f(\boldsymbol{k}, \boldsymbol{q}) = \cos(k_x) - \cos(k_y)$. In the calculation that follows, we will use the more general notation $f(\boldsymbol{k}, \boldsymbol{q})$ and specialize to specific physical cases only towards the end. In expressions involving the external momentum, we will take $|\boldsymbol{q}| \to 0$ and use the notation $f(\boldsymbol{k}, \hat{q})$ to emphasize that only the orientation $\hat{q}$ matters.

## F.1  Recap of previous results

For both gauge fields and Ising-nematic order parameters, calculations in Refs. [64, 73–76] invariably found the leading frequency scaling of the boson self energy (valid for fixed $\omega$ and at asymptotically large $N_f$) to be

$$\Pi(\omega) = N_f \left[ C_0 + \frac{C_z}{N_f} \omega^{(4-z)/z} + \dots \right], \tag{F.7}$$

where $C_0, C_z$ are constants and $z$ is the boson dynamical critical exponent. Since the most singular frequency scaling in the boson self energy controls the most singular frequency scaling in the conductivity (up to a factor of $\omega$), Eq. (F.7) led to the expectation that

$$\sigma(\omega) = N_f \left[ \frac{\mathcal{D}}{\pi} \frac{i}{\omega} + \frac{\sigma_z}{N_f} \omega^{-2(z-2)/z} + \dots \right], \tag{F.8}$$

where $\mathcal{D}$ is the Drude weight and $\sigma_z$ is a $z$-dependent constant. A precise analysis of the Feynman diagrams contributing to the conductivity reveals that this frequency dependence

is generated entirely by irrelevant operators that invalidate the anomaly constraints [19]. Within the mid-IR formulation of Eq. (3.2), the leading irrelevant operators come from allowing the boson-fermion coupling, $\vec{f}(\theta)$, to vary within patches of the Fermi surface, $\vec{f}(\theta, \boldsymbol{k})$. Indeed, expanding $\vec{f}(\theta, \boldsymbol{k})$ in powers of $\boldsymbol{k}$ around $\boldsymbol{k}_F(\theta)$ and keeping only the linear term, we find that the resulting Feynman diagrams scale as $\omega^{-2(z-2)/z}$.

## F.2 Summary of new results

Although all previous works found the same scaling exponent as in Eq. (F.7), the numerical prefactor $C_z$ has never been explicitly reported. Interestingly, its value depends strongly on the order parameter symmetry. In what follows, we compute $C_z$ for a completely general inversion-symmetric Fermi surface. Surprisingly, we find that the value of $C_z$ is non-zero for inversion-even order parameters ($f^a(\theta) = f^a(\theta + \pi)$), but *vanishes* for generic "loop-current" order parameters ($f^a(\theta) = -f^a(\theta + \pi)$). This means that for any inversion-odd order parameter, it is necessary to include even higher-dimension irrelevant operators to obtain non-trivial frequency scaling in the boson self energy. The ultimate result is

$$\Pi(\omega) = N_f \left[ C_0 + \frac{1}{N_f} \begin{cases} 0, & \vec{f}(\theta) = -\vec{f}(\theta + \pi) \\ C_z\,\omega^{-2(z-2)/z}, & \vec{f}(\theta) = \vec{f}(\theta + \pi) \end{cases} + \text{less singular than } \omega^{-2(z-2)/z} \right].$$
(F.9)

For general order parameters, the diagrams that contribute to the boson self energy are not simply related to those that contribute to the current-current correlators and our results for $\Pi(\omega)$ do not immediately determine the scaling of $\sigma(\omega)$. Nevertheless, after carefully accounting for all diagrams that contribute to the conductivity, we demonstrate that

$$\frac{\sigma(\omega)}{N_f} = \frac{\mathcal{D}}{\pi} \frac{i}{\omega} + \text{less singular than } \omega^{-2(z-2)/z}.$$
(F.10)

Based on the large-$N_f$ expansion, then, it appears that it is more difficult to generate incoherent conductivity from irrelevant operators than previously believed. Note, however, that even at $T = 0$, for any finite $N_f$, these perturbative calculations cannot be trusted down to $\omega = 0$ due to infrared singularities [28]. We also emphasize that the above frequency dependence of $\sigma(\omega)$ in Eq. (F.10) is *not* a property of the infrared fixed point, meaning that it will not fit into scaling function of $\omega/T$ at finite temperature, $T$.

In the rest of the Appendix, we will first calculate the boson self energy to leading two orders in the $1/N_f$ expansion and recover Eq. (F.9). Then we evaluate some additional diagrams to arrive at the conductivity formula in Eq. (F.10).

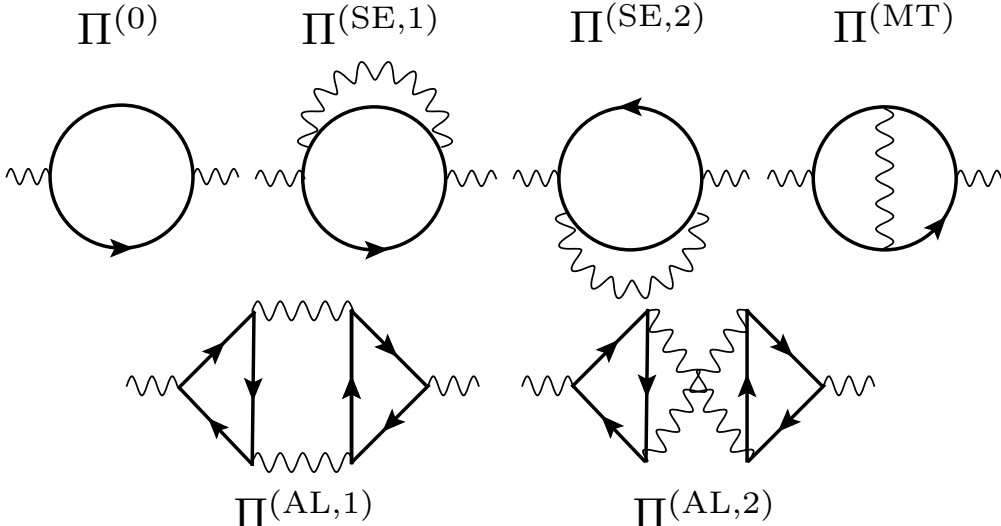

**Figure 6:** The set of diagrams contributing to the boson self energy to leading two orders in the $1/N_f$ expansion. All vertex factors are taken to be $f(\boldsymbol{k}, \boldsymbol{q})$, although analogues of our results hold when the external vertices are not equal to the internal vertices (see Eq. (F.57)). The free fermion bubble $\Pi^{(0)}$ evaluates to a constant in accordance with the general nonperturbative arguments in Section 3. The remaining diagrams give the leading frequency scaling of the boson self energy.

### F.3 Organization of diagrams for the boson self energy

We now proceed to organize the diagrams that we need to compute to extract Eq. (F.9) (see Figure 6). At leading order in $1/N_f$, the boson self energy is just given by a one-loop integral

$$\Pi^{(0)}(\boldsymbol{q}, i\nu) = -N_f \int_{\boldsymbol{k},\omega} f(\boldsymbol{k}, \hat{\boldsymbol{q}})^2 G(\boldsymbol{k} + \frac{\boldsymbol{q}}{2}, i\omega + i\nu) G(\boldsymbol{k} - \frac{\boldsymbol{q}}{2}, i\omega) \,. \tag{F.11}$$

This integral gives a constant contribution $N_f C_0$ that depends on the form factor but not on $\omega$. The leading $1/N_f$ corrections are organized into five diagrams: two fermion self-energy corrections $\Pi^{(\mathrm{SE},1)}, \Pi^{(\mathrm{SE},2)}$, the Maki-Thompson correction $\Pi^{(\mathrm{MT})}$, and the Aslamazov-Larkin corrections $\Pi^{(\mathrm{AL},1)}, \Pi^{(\mathrm{AL},2)}$ (the arguments of these functions are always understood to be $\boldsymbol{q} = 0, i\nu$). It is convenient to split the Maki-Thompson correction into $\Pi^{(\mathrm{MT},1)} + \Pi^{(\mathrm{MT},2)}$ where $\Pi^{(\mathrm{MT},1)}$ excludes irrelevant operator insertions on the external vertex (while keeping irrelevant operators in the internal loops).

### F.4 Precise cancellation between $\Pi^{(\mathrm{SE},1)}$, $\Pi^{(\mathrm{SE},2)}$ and $\Pi^{(\mathrm{MT},1)}$

The contributions $\Pi^{(\mathrm{SE},1)}$, $\Pi^{(\mathrm{SE},2)}$ and $\Pi^{(\mathrm{MT},1)}$ each contains a single fermion loop and comes with a $(-1)$ prefactor. Using $\int_{\boldsymbol{k},\omega}$ to denote $\int \frac{d^2 k d\omega}{(2\pi)^3}$, the one-loop fermion self energies will

involve the following integrals

$$\Sigma(\boldsymbol{k}, i\omega) = \int_{\boldsymbol{q},\nu} f(\boldsymbol{k} + \frac{\boldsymbol{q}}{2}, \hat{\boldsymbol{q}})^2 G(\boldsymbol{k} + \boldsymbol{q}, i\omega + i\nu) D(\boldsymbol{q}, i\nu) \,. \tag{F.12}$$

In terms of these self energy integrals,

$$\begin{aligned}
\Pi^{(\text{SE},1)} &= -\int_{\boldsymbol{k},\omega} f(\boldsymbol{k}, \hat{\boldsymbol{q}})^2 \Sigma(\boldsymbol{k}, i\omega) G(\boldsymbol{k}, i\omega)^2 G(\boldsymbol{k}, i\omega + i\nu) \\
&= -\frac{1}{i\nu} \int_{\boldsymbol{k},\omega} f(\boldsymbol{k}, \hat{\boldsymbol{q}})^2 \Sigma(\boldsymbol{k}, i\omega) G(\boldsymbol{k}, i\omega) \left[ G(\boldsymbol{k}, i\omega) - G(\boldsymbol{k}, i\omega + i\nu) \right] \,, \tag{F.13}
\end{aligned}$$

$$\begin{aligned}
\Pi^{(\text{SE},2)} &= -\int_{\boldsymbol{k},\omega} f(\boldsymbol{k}, \hat{\boldsymbol{q}})^2 \Sigma(\boldsymbol{k}, i\omega + i\nu) G(\boldsymbol{k}, i\omega) G(\boldsymbol{k}, i\omega + i\nu)^2 \\
&= -\frac{1}{i\nu} \int_{\boldsymbol{k},\omega} f(\boldsymbol{k}, \hat{\boldsymbol{q}})^2 \Sigma(\boldsymbol{k}, i\omega + i\nu) G(\boldsymbol{k}, i\omega + i\nu) \left[ G(\boldsymbol{k}, i\omega) - G(\boldsymbol{k}, i\omega + i\nu) \right] \,. \tag{F.14}
\end{aligned}$$

When added together, these diagrams partially cancel each other and we find

$$\Pi^{(\text{SE},1)} + \Pi^{(\text{SE},2)} = \int_{\boldsymbol{k},\omega} f(\boldsymbol{k}, \hat{\boldsymbol{q}})^2 G(\boldsymbol{k}, i\omega) G(\boldsymbol{k}, i\omega + i\nu) \frac{\Sigma(\boldsymbol{k}, i\omega) - \Sigma(\boldsymbol{k}, i\omega + i\nu)}{i\nu} \,. \tag{F.15}$$

We now contrast the above expression with the Maki-Thompson diagram

$$\Pi^{(\text{MT})} = \int_{\boldsymbol{k},\omega} G(\boldsymbol{k}, i\omega) G(\boldsymbol{k}, i\omega + i\nu) f(\boldsymbol{k}, \hat{\boldsymbol{q}}) \Gamma(\boldsymbol{k}, \hat{\boldsymbol{q}}, i\omega, i\nu) \,, \tag{F.16}$$

where

$$\Gamma(\boldsymbol{k}, \hat{\boldsymbol{q}}, i\omega, i\nu) = -\int_{\boldsymbol{q}',\nu'} f(\boldsymbol{k}+\boldsymbol{q}', \hat{\boldsymbol{q}}) f(\boldsymbol{k}+\frac{\boldsymbol{q}'}{2}, \boldsymbol{q}')^2 G(\boldsymbol{k}+\boldsymbol{q}', i\omega+i\nu') G(\boldsymbol{k}+\boldsymbol{q}', i\omega+i\nu+i\nu') D(\boldsymbol{q}', i\nu') \,. \tag{F.17}$$

Pictorially, the Maki-Thompson diagram describes a process in which two fermions exchange a boson and then annihilate. Before and after the exchange, the fermion momenta shift by $\boldsymbol{q}'$, which is small compared to the Fermi momentum. Therefore, in the low energy limit, we can decompose the external vertex factor $f(\boldsymbol{k} + \boldsymbol{q}', \hat{\boldsymbol{q}}) = f(\boldsymbol{k}, \hat{\boldsymbol{q}}) + [f(\boldsymbol{k} + \boldsymbol{q}', \hat{\boldsymbol{q}}) - f(\boldsymbol{k}, \hat{\boldsymbol{q}})]$, anticipating that the second term will give a subleading contribution relative to the first. This motivates the definitions

$$\begin{aligned}
\Gamma^{(1)}(\boldsymbol{k}, \hat{\boldsymbol{q}}, i\omega, i\nu) = -\int_{\boldsymbol{q}',\nu'} f(\boldsymbol{k}, \hat{\boldsymbol{q}}) f(\boldsymbol{k} + \frac{\boldsymbol{q}'}{2}, \boldsymbol{q}')^2 G(\boldsymbol{k} + \boldsymbol{q}', i\omega + i\nu') \\
\cdot G(\boldsymbol{k} + \boldsymbol{q}', i\omega + i\nu + i\nu') D(\boldsymbol{q}', i\nu') \,, \tag{F.18}
\end{aligned}$$

$$\Gamma^{(2)}(\boldsymbol{k},\hat{\boldsymbol{q}},i\omega,i\nu) = -\int_{\boldsymbol{q}',\nu'} [f(\boldsymbol{k}+\boldsymbol{q}',\hat{\boldsymbol{q}}) - f(\boldsymbol{k},\hat{\boldsymbol{q}})] f(\boldsymbol{k}+\frac{\boldsymbol{q}'}{2},\boldsymbol{q}')^2$$

$$\cdot\, G(\boldsymbol{k}+\boldsymbol{q}',i\omega+i\nu')G(\boldsymbol{k}+\boldsymbol{q}',i\omega+i\nu+i\nu')D(\boldsymbol{q}',i\nu')\,. \quad \text{(F.19)}$$

It is now easy to demonstrate that

$$\Pi^{(\mathrm{MT},1)} = -\int_{\boldsymbol{k},\omega} f(\boldsymbol{k},\hat{\boldsymbol{q}})^2 G(\boldsymbol{k},i\omega)G(\boldsymbol{k},i\omega+i\nu)$$

$$\cdot \int_{\boldsymbol{q}',\nu'} f(\boldsymbol{k}+\frac{\boldsymbol{q}'}{2},\boldsymbol{q}')^2 \frac{G(\boldsymbol{k}+\boldsymbol{q}',i\omega+i\nu') - G(\boldsymbol{k}+\boldsymbol{q}',i\omega+i\nu+i\nu')}{i\nu} D(\boldsymbol{q}',i\nu')$$

$$= -\int_{\boldsymbol{k},\omega} f(\boldsymbol{k},\hat{\boldsymbol{q}})^2 G(\boldsymbol{k},i\omega)G(\boldsymbol{k},i\omega+i\nu)\frac{\Sigma(\boldsymbol{k},i\omega) - \Sigma(\boldsymbol{k},i\omega+i\nu)}{i\nu}\,. \quad \text{(F.20)}$$

Comparing with (F.15), we find a perfect cancellation

$$\Pi^{(\mathrm{SE},1)} + \Pi^{(\mathrm{SE},2)} + \Pi^{(\mathrm{MT},1)} = 0\,. \quad \text{(F.21)}$$

## F.5   Reduction of $\Pi^{(\mathrm{MT},2)}, \Pi^{(\mathrm{AL},1)}, \Pi^{(\mathrm{AL},2)}$ to combinations of one-loop diagrams

The cancellations in the previous section show that any frequency dependence in the boson self energy at order $1/N_f$ must come from $\Pi^{(\mathrm{MT},2)}, \Pi^{(\mathrm{AL},1)}, \Pi^{(\mathrm{AL},2)}$. In what follows we will use general symmetry arguments to reduce each diagram to a sum of one-loop integrals which are much simpler to evaluate.

For the Maki-Thompson diagram, via a linear shift of the fermionic momentum $\boldsymbol{k} \to \boldsymbol{k} - \frac{\boldsymbol{q}'}{2}$, we have

$$\Pi^{(\mathrm{MT},2)} = -\int_{\boldsymbol{q}',\nu',\boldsymbol{k},\omega} [f(\boldsymbol{k}+\frac{\boldsymbol{q}'}{2},\hat{\boldsymbol{q}}) - f(\boldsymbol{k}-\frac{\boldsymbol{q}'}{2},\hat{\boldsymbol{q}})] f(\boldsymbol{k}-\frac{\boldsymbol{q}'}{2},\hat{\boldsymbol{q}}) f(\boldsymbol{k},\boldsymbol{q}')^2 D(\boldsymbol{q}',i\nu')$$

$$\times G(\boldsymbol{k}-\frac{\boldsymbol{q}'}{2},i\omega)G(\boldsymbol{k}-\frac{\boldsymbol{q}'}{2},i\omega+i\nu)G(\boldsymbol{k}+\frac{\boldsymbol{q}'}{2},i\omega+i\nu')G(\boldsymbol{k}+\frac{\boldsymbol{q}'}{2},i\omega+i\nu+i\nu')\,. \quad \text{(F.22)}$$

If we make a different change of variables $\boldsymbol{q}' \to -\boldsymbol{q}', \nu' \to -\nu'$ and use $D(\boldsymbol{q}',i\nu') = D(-\boldsymbol{q}',-i\nu')$, we find an alternative expression

$$\Pi^{(\mathrm{MT},2)} = \int_{\boldsymbol{q}',\nu',\boldsymbol{k},\omega} [f(\boldsymbol{k}+\frac{\boldsymbol{q}'}{2},\hat{\boldsymbol{q}}) - f(\boldsymbol{k}-\frac{\boldsymbol{q}'}{2},\hat{\boldsymbol{q}})] f(\boldsymbol{k}+\frac{\boldsymbol{q}'}{2},\hat{\boldsymbol{q}}) f(\boldsymbol{k},\boldsymbol{q}')^2 D(\boldsymbol{q}',i\nu')$$

$$\times G(\boldsymbol{k}-\frac{\boldsymbol{q}'}{2},i\omega)G(\boldsymbol{k}-\frac{\boldsymbol{q}'}{2},i\omega+i\nu)G(\boldsymbol{k}+\frac{\boldsymbol{q}'}{2},i\omega+i\nu')G(\boldsymbol{k}+\frac{\boldsymbol{q}'}{2},i\omega+i\nu+i\nu')\,. \quad \text{(F.23)}$$

Adding up these two expressions for the same diagram, and using the Landau damping form

of $D(\boldsymbol{q}', i\nu')$, we obtain

$$\Pi^{(\mathrm{MT},2)} = \frac{1}{2} \int_{\boldsymbol{q}',\nu',\boldsymbol{k},\omega} [f(\boldsymbol{k}+\frac{\boldsymbol{q}'}{2},\hat{\boldsymbol{q}}) - f(\boldsymbol{k}-\frac{\boldsymbol{q}'}{2},\hat{\boldsymbol{q}})]^2 f(\boldsymbol{k},\boldsymbol{q}')^2 D(\boldsymbol{q}',i\nu')$$

$$\times G(\boldsymbol{k}-\frac{\boldsymbol{q}'}{2},i\omega)G(\boldsymbol{k}-\frac{\boldsymbol{q}'}{2},i\omega+i\nu)G(\boldsymbol{k}+\frac{\boldsymbol{q}'}{2},i\omega+i\nu')G(\boldsymbol{k}+\frac{\boldsymbol{q}'}{2},i\omega+i\nu+i\nu'). \quad (\mathrm{F}.24)$$

Using the Landau damping form of $D(\boldsymbol{q}', i\nu')$, we can rewrite the integral over $\nu'$ as

$$\int_{\nu'} D(\boldsymbol{q}',i\nu')G(\boldsymbol{k}+\frac{\boldsymbol{q}'}{2},i\omega+i\nu')G(\boldsymbol{k}+\frac{\boldsymbol{q}'}{2},i\omega+i\nu+i\nu')$$

$$= \frac{1}{i\nu} \int_{\nu'} D(\boldsymbol{q}',i\nu') \left[ G(\boldsymbol{k}+\frac{\boldsymbol{q}'}{2},i\omega+i\nu') - G(\boldsymbol{k}+\frac{\boldsymbol{q}'}{2},i\omega+i\nu+i\nu') \right]$$

$$= \frac{1}{i\nu} \int_{\nu'} [D(\boldsymbol{q}',i\nu') - D(\boldsymbol{q}',i\nu'+i\nu)] G(\boldsymbol{k}+\frac{\boldsymbol{q}'}{2},i\omega-i\nu')$$

$$= \frac{\gamma_{\hat{\boldsymbol{q}}'}}{i\nu|\boldsymbol{q}'|} \int_{\nu'} D(\boldsymbol{q}',i\nu')D(\boldsymbol{q}',i\nu'+i\nu) \left[|\nu'+\nu| - |\nu'|\right] G(\boldsymbol{k}+\frac{\boldsymbol{q}'}{2},i\omega-i\nu'). \quad (\mathrm{F}.25)$$

Plugging the final line into the expression for $\Pi^{(\mathrm{MT},2)}$, we find

$$\Pi^{(\mathrm{MT},2)} = \frac{1}{2\nu^2} \int_{\boldsymbol{q}',\nu'} D(\boldsymbol{q}',i\nu')D(\boldsymbol{q}',i\nu'+i\nu) \left[|\nu'+\nu| - |\nu'|\right] \frac{\gamma_{\hat{\boldsymbol{q}}'}}{|\boldsymbol{q}'|} \left[\pi^{(\mathrm{MT})}(\boldsymbol{q}',\nu'+\nu) - \pi^{(\mathrm{MT})}(\boldsymbol{q}',\nu')\right],$$

$$(\mathrm{F}.26)$$

where the one loop-integral that we need to evaluate is

$$\pi^{(\mathrm{MT})}(\boldsymbol{q}',\nu') = \int_{\boldsymbol{k},\omega} [f(\boldsymbol{k}+\frac{\boldsymbol{q}'}{2},\hat{\boldsymbol{q}}) - f(\boldsymbol{k}-\frac{\boldsymbol{q}'}{2},\hat{\boldsymbol{q}})]^2 f(\boldsymbol{k},\boldsymbol{q}')^2 G(\boldsymbol{k}-\frac{\boldsymbol{q}'}{2},i\omega)G(\boldsymbol{k}+\frac{\boldsymbol{q}'}{2},i\omega+i\nu').$$

$$(\mathrm{F}.27)$$

Instead of evaluating $\pi^{(\mathrm{MT})}(\boldsymbol{q}',\nu')$ right away, we will massage the Aslamazov-Larkin diagrams to a similar form. By applying the Feynman rules directly to the two diagrams, we get

$$\Pi^{(\mathrm{AL},1)} = \int_{\boldsymbol{q}',\nu'} D(\boldsymbol{q}',i\nu')D(\boldsymbol{q}',i\nu'+i\nu)I(\boldsymbol{q}',\nu,\nu')^2, \quad (\mathrm{F}.28)$$

$$\Pi^{(\mathrm{AL},2)} = \int_{\boldsymbol{q}',\nu'} D(\boldsymbol{q}',i\nu')D(\boldsymbol{q}',i\nu'+i\nu)I(\boldsymbol{q}',\nu,\nu')I(-\boldsymbol{q}',\nu,-\nu'-\nu), \quad (\mathrm{F}.29)$$

where

$$I(\boldsymbol{q}',\nu,\nu') = \int_{\boldsymbol{k},\omega} f(\boldsymbol{k},\hat{\boldsymbol{q}})f(\boldsymbol{k}+\frac{\boldsymbol{q}'}{2},\boldsymbol{q}')^2 G(\boldsymbol{k},i\omega)G(\boldsymbol{k},i\omega+i\nu)G(\boldsymbol{k}+\boldsymbol{q}',i\omega+i\nu+i\nu'). \quad (\mathrm{F}.30)$$

By a simple change of variables $\boldsymbol{q}' \to -\boldsymbol{q}', \nu' \to -\nu-\nu'$, we obtain a succinct representation of the sum of two Aslamazov-Larkin diagrams

$$\Pi^{(\mathrm{AL},1)}+\Pi^{(\mathrm{AL},2)} = \frac{1}{2} \int_{\boldsymbol{q}',\nu'} D(\boldsymbol{q}',i\nu')D(\boldsymbol{q}',i\nu'+i\nu) \left[I(\boldsymbol{q}',\nu,\nu') + I(-\boldsymbol{q}',\nu,-\nu-\nu')\right]^2. \quad (\mathrm{F}.31)$$

In analogy with the manipulations we performed for the Maki-Thompson diagrams, we can define

$$\pi^{(\mathrm{AL})}(\boldsymbol{q}',\nu') = \int_{\boldsymbol{k},\omega} \left[ f(\boldsymbol{k}+\tfrac{\boldsymbol{q}'}{2},\hat{\boldsymbol{q}}) - f(\boldsymbol{k}-\tfrac{\boldsymbol{q}'}{2},\hat{\boldsymbol{q}}) \right] f(\boldsymbol{k},\boldsymbol{q}')^2 G(\boldsymbol{k}+\tfrac{\boldsymbol{q}'}{2},i\omega+i\nu')G(\boldsymbol{k}-\tfrac{\boldsymbol{q}'}{2},i\omega),$$

(F.32)

in terms of which

$$\Pi^{(\mathrm{AL},1)} + \Pi^{(\mathrm{AL},2)} = -\frac{1}{2\nu^2} \int_{\boldsymbol{q}',\nu'} D(\boldsymbol{q}',i\nu')D(\boldsymbol{q}',i\nu'+i\nu) \left[ \pi^{(\mathrm{AL})}(\boldsymbol{q}',\nu+\nu') - \pi^{(\mathrm{AL})}(\boldsymbol{q}',\nu') \right]^2.$$

(F.33)

## F.6 Evaluation of one-loop diagrams and extraction of frequency scaling

One can only avoid explicit integrals for so long. To reach our final conclusion, we have to evaluate $\pi^{(\mathrm{AL})}$ and $\pi^{(\mathrm{MT})}$. It is helpful to consider a more general case

$$\pi_F(\boldsymbol{q}',\nu) = \int_{\boldsymbol{k},\omega} F(\boldsymbol{k},\boldsymbol{q}')G(\boldsymbol{k}+\tfrac{\boldsymbol{q}'}{2},i\omega+i\nu')G(\boldsymbol{k}-\tfrac{\boldsymbol{q}'}{2},i\omega),$$

(F.34)

where $\pi^{(\mathrm{MT})}$ and $\pi^{(\mathrm{AL})}$ correspond to taking $F(\boldsymbol{k},\boldsymbol{q}') = [f(\boldsymbol{k}+\tfrac{\boldsymbol{q}'}{2},\hat{\boldsymbol{q}}) - f(\boldsymbol{k}-\tfrac{\boldsymbol{q}'}{2},\hat{\boldsymbol{q}})]^2 f(\boldsymbol{k},\boldsymbol{q}')^2$ and $F(\boldsymbol{k},\boldsymbol{q}') = [f(\boldsymbol{k}+\tfrac{\boldsymbol{q}'}{2},\hat{\boldsymbol{q}}) - f(\boldsymbol{k}-\tfrac{\boldsymbol{q}'}{2},\hat{\boldsymbol{q}})]f(\boldsymbol{k},\boldsymbol{q}')^2$. We now make a change of variables from $\boldsymbol{k} \to \epsilon(\boldsymbol{k}),\theta(\boldsymbol{k})$ where $\theta$ is an angular coordinate parametrizing the Fermi surface. We assume that the Fermi surface is sufficiently smooth so that the Jacobian $J(\theta,\epsilon)$ of this transformation is nonsingular. This allows us to conclude that

$$\pi_F(\boldsymbol{q}',\nu) = \frac{1}{8\pi^3} \int d\epsilon d\theta \frac{J(\theta,\epsilon)F(\theta,\epsilon,\boldsymbol{q}')}{i\nu - \epsilon(\boldsymbol{k}+\tfrac{\boldsymbol{q}'}{2}) + \epsilon(\boldsymbol{k}-\tfrac{\boldsymbol{q}'}{2})} \int d\omega \left[ G(\boldsymbol{k}-\tfrac{\boldsymbol{q}'}{2},i\omega) - G(\boldsymbol{k}+\tfrac{\boldsymbol{q}'}{2},i\omega+i\nu) \right]$$

$$= \frac{1}{8\pi^2} \int d\epsilon d\theta J(\theta,\epsilon)F(\theta,\epsilon,\boldsymbol{q}') \frac{\mathrm{sgn}\left[\epsilon(\boldsymbol{k}+\tfrac{\boldsymbol{q}'}{2})\right] - \mathrm{sgn}\left[\epsilon(\boldsymbol{k}-\tfrac{\boldsymbol{q}'}{2})\right]}{i\nu - \epsilon(\boldsymbol{k}+\tfrac{\boldsymbol{q}'}{2}) + \epsilon(\boldsymbol{k}-\tfrac{\boldsymbol{q}'}{2})}.$$

(F.35)

The most singular contributions to this integral come from a region in phase space where $\boldsymbol{k}$ is on the Fermi surface and $|\boldsymbol{q}'| \ll |\boldsymbol{k}|$. Hence, to leading order in frequency we can make the approximation $\epsilon(\boldsymbol{k} \pm \tfrac{\boldsymbol{q}'}{2}) \approx \epsilon \pm \tfrac{1}{2}\boldsymbol{q}' \cdot \nabla_{\boldsymbol{k}}\epsilon$. By evaluating $J(\theta,\epsilon)$ and $F(\theta,\epsilon,\boldsymbol{q}')$ exactly at $\epsilon = \epsilon_F$ (and dropping the $\epsilon$ argument from now on), we can perform the $\epsilon$ integral and get

$$\pi_F(\boldsymbol{q}',\nu) = \frac{1}{4\pi^2} \int d\theta J(\theta)F(\theta,\boldsymbol{q}') \frac{\boldsymbol{q}' \cdot \boldsymbol{v}_F(\theta) + \mathcal{O}(q^3)}{i\nu - \boldsymbol{q}' \cdot \boldsymbol{v}_F(\theta) + \mathcal{O}(q^3)}$$

$$= \pi_0(\boldsymbol{q}') + \frac{i\nu}{4\pi^2|\boldsymbol{q}'|} \int d\theta \frac{J(\theta)F(\theta,\boldsymbol{q}')}{i\frac{\nu}{|\boldsymbol{q}'|} - \hat{\boldsymbol{q}}' \cdot \boldsymbol{v}_F(\theta)}$$

$$= \pi_0(\boldsymbol{q}') + \frac{\nu}{4\pi^2|\boldsymbol{q}'|} \int d\theta J(\theta)F(\theta,\boldsymbol{q}') \frac{\frac{\nu}{|\boldsymbol{q}'|} - i\hat{\boldsymbol{q}}' \cdot \boldsymbol{v}_F(\theta)}{\left|\frac{\nu}{|\boldsymbol{q}'|}\right|^2 + |\hat{\boldsymbol{q}}' \cdot \boldsymbol{v}_F(\theta)|^2},$$

(F.36)

where we have isolated a term $\pi_0(\boldsymbol{q}')$ independent of $\nu$ and decomposed the other term into real and imaginary parts. Now let us consider different choices of $F$ in turn.

1. If $F(\theta, \boldsymbol{q}') = f(\theta, \boldsymbol{q}')^2$, then under spatial inversion $\theta \to \theta + \pi$, $J(\theta), F(\theta, \boldsymbol{q}')$ are invariant while $\boldsymbol{v}_F(\theta)$ flips sign. This means the imaginary part vanishes. To evaluate the real part, we recall that the internal bosons have $z = 3$ scaling and $|\nu|/|\boldsymbol{q}'|$ should be regarded as a small parameter $\delta$. Using $\frac{\delta}{\delta^2 + x^2} \approx \pi \delta(x)$, we thus have

$$\pi_F(\boldsymbol{q}', \nu) = \pi_0(\boldsymbol{q}') + \frac{|\nu|}{4\pi|\boldsymbol{q}'|} \int d\theta J(\theta) f(\theta, \boldsymbol{q}')^2 \delta \left[ \hat{\boldsymbol{q}}' \cdot \boldsymbol{v}_F(\theta) \right] = \pi_0(\boldsymbol{q}') + \frac{\gamma_{\hat{\boldsymbol{q}}'} |\nu|}{|\boldsymbol{q}'|}, \quad \text{(F.37)}$$

where we identified the Landau damping coefficient

$$\gamma_{\hat{\boldsymbol{q}}'} = \frac{1}{4\pi} \int d\theta J(\theta) f(\theta, \boldsymbol{q}')^2 \delta \left[ \hat{\boldsymbol{q}}' \cdot \boldsymbol{v}_F(\theta) \right] . \quad \text{(F.38)}$$

2. For the Maki-Thompson diagram, we take

$$F(\boldsymbol{k}, \boldsymbol{q}') = \left[ f\left(\boldsymbol{k} + \frac{\boldsymbol{q}'}{2}, \hat{\boldsymbol{q}}\right) - f\left(\boldsymbol{k} - \frac{\boldsymbol{q}'}{2}, \hat{\boldsymbol{q}}\right) \right]^2 f(\boldsymbol{k}, \boldsymbol{q}')^2 . \quad \text{(F.39)}$$

To leading order in $|\boldsymbol{q}'|/\boldsymbol{k}$,

$$F(\theta, \boldsymbol{q}') \approx |\boldsymbol{q}'|^2 \left| \hat{\boldsymbol{q}}' \cdot \nabla f(\theta, \hat{\boldsymbol{q}}) \right|^2 f(\theta, \boldsymbol{q}')^2 . \quad \text{(F.40)}$$

Under spatial inversion, $J(\theta), F(\theta, \boldsymbol{q}')$ are still invariant while $\boldsymbol{v}_F(\theta)$ flips sign. Like in the previous case, the imaginary part of $\pi_F$ thus vanishes, and we are left with

$$\pi_F(\boldsymbol{q}', \nu) = \pi_0(\boldsymbol{q}') + \frac{|\nu|}{4\pi|\boldsymbol{q}'|} |\boldsymbol{q}'|^2 \int d\theta J(\theta) |\hat{\boldsymbol{q}}' \cdot \nabla f(\theta, \hat{\boldsymbol{q}})|^2 f(\theta, \boldsymbol{q}')^2 \delta \left[ \hat{\boldsymbol{q}}' \cdot \boldsymbol{v}_F(\theta) \right] . \quad \text{(F.41)}$$

For every $\hat{\boldsymbol{q}}'$, there are precisely two antipodal angles $\theta_{\hat{\boldsymbol{q}}'}, \theta_{\hat{\boldsymbol{q}}'} + \pi$ for which $\hat{\boldsymbol{q}}' \cdot \boldsymbol{v}_F(\theta) = 0$. Since $|\hat{\boldsymbol{q}}' \cdot \nabla f(\theta_{\hat{\boldsymbol{q}}'} + \pi, \hat{\boldsymbol{q}})|^2 = |\hat{\boldsymbol{q}}' \cdot \nabla f(\theta_{\hat{\boldsymbol{q}}'}, \hat{\boldsymbol{q}})|^2$, we conclude that the Maki-Thompson diagram have the following structure independent of order parameter symmetries

$$\pi^{(\text{MT})}(\boldsymbol{q}', \nu') = \pi_0(\boldsymbol{q}') + \gamma_{\hat{\boldsymbol{q}}'} |\nu'||\boldsymbol{q}'| |\hat{\boldsymbol{q}}' \cdot \nabla f(\theta_{\hat{\boldsymbol{q}}'}, \hat{\boldsymbol{q}})|^2 . \quad \text{(F.42)}$$

3. For the Aslamazov-Larkin diagrams, we take

$$F(\theta, \boldsymbol{q}') \approx |\boldsymbol{q}'| \hat{\boldsymbol{q}}' \cdot \nabla f(\theta, \hat{\boldsymbol{q}}) f(\theta, \boldsymbol{q}')^2 . \quad \text{(F.43)}$$

Unlike the Maki-Thompson diagram, the Aslamazov-Larkin diagrams are sensitive to the choice of order parameter symmetry. Under a spatial inversion $\theta \to \theta + \pi$, $J(\theta)$

is always even and $\boldsymbol{v}_F(\theta)$ is always odd; but the parity of $F(\theta, \boldsymbol{q}')$ is opposite to the parity of $f(\theta, \boldsymbol{q}')$. Therefore, for an inversion-odd order parameter, $\mathrm{Re}\,\pi_F(\boldsymbol{q}', \nu)$ survives; for an inversion-even order parameter, $\mathrm{Im}\,\pi_F(\boldsymbol{q}', \nu)$ survives. After evaluating the integrals, we find

$$\pi^{(\mathrm{AL})}(\boldsymbol{q}', \nu') = \pi_0(\boldsymbol{q}') + \begin{cases} \gamma_{\hat{\boldsymbol{q}}'} |\nu'| \hat{\boldsymbol{q}}' \cdot \nabla f(\theta_{\hat{\boldsymbol{q}}'}, \hat{\boldsymbol{q}}) & \text{inversion-odd} \\ -i\frac{\nu'}{4\pi^2}\mathcal{I}(\boldsymbol{q}', \nu') & \text{inversion-even} \end{cases}, \quad \text{(F.44)}$$

where $\mathcal{I}$ is an undetermined real function that will not be too important.

We now plug the above expressions for $\pi^{(\mathrm{MT})}$ and $\pi^{(\mathrm{AL})}$ into the Maki-Thompson and Aslamazov-Larkin diagrams to get

$$\Pi^{(\mathrm{MT},2)} = \frac{1}{2\nu^2} \int_{\boldsymbol{q}',\nu'} D(\boldsymbol{q}', i\nu') D(\boldsymbol{q}', i\nu' + i\nu) \left[|\nu' + \nu| - |\nu'|\right]^2 \gamma_{\hat{\boldsymbol{q}}'}^2 |\hat{\boldsymbol{q}}' \cdot \nabla f(\theta_{\hat{\boldsymbol{q}}'}, \hat{\boldsymbol{q}})|^2, \quad \text{(F.45)}$$

$$\Pi^{(\mathrm{AL},1)} + \Pi^{(\mathrm{AL},2)} \quad \text{(F.46)}$$

$$= -\frac{1}{2\nu^2} \int_{\boldsymbol{q}',\nu'} D(\boldsymbol{q}', i\nu') D(\boldsymbol{q}', i\nu' + i\nu) \begin{cases} [\gamma_{\hat{\boldsymbol{q}}'} \hat{\boldsymbol{q}}' \cdot \nabla f(\theta_{\hat{\boldsymbol{q}}'}, \hat{\boldsymbol{q}})]^2 \left[|\nu' + \nu| - |\nu'|\right]^2 & \text{inversion-odd} \\ \left[\frac{i\nu}{4\pi^2}\mathcal{I}(\boldsymbol{q}', \nu')\right]^2 & \text{inversion-even} \end{cases}. \quad \text{(F.47)}$$

By comparing (F.46) and (F.45), we immediately see that when the order parameter is odd under inversion,

$$\Pi^{(\mathrm{MT},2)} + \Pi^{(\mathrm{AL},1)} + \Pi^{(\mathrm{AL},2)} = 0. \quad \text{(F.48)}$$

On the other hand, when the order parameter is even under inversion, since $D(\boldsymbol{q}', i\nu')$ is positive, the integrand for $\Pi^{(\mathrm{MT},2)}$ and $\Pi^{(\mathrm{AL},1)} + \Pi^{(\mathrm{AL},2)}$ are positive functions of $\boldsymbol{q}', \nu'$ that add together. Therefore, without explicitly computing $\mathcal{I}(\boldsymbol{q}', \nu')$, we can already conclude that

$$\Pi^{(\mathrm{MT},2)} + \Pi^{(\mathrm{AL},1)} + \Pi^{(\mathrm{AL},2)} \neq 0. \quad \text{(F.49)}$$

Finally, to estimate the leading frequency scaling, we can evaluate the above expressions in the special case with rotational invariance. We find, in agreement with Kim et al., that

$$\Pi^{(\mathrm{MT},2)}, \Pi^{(\mathrm{AL},1)} + \Pi^{(\mathrm{AL},2)} \sim |\nu|^{\frac{4-z}{z}}. \quad \text{(F.50)}$$

Hence, neglecting corrections with higher powers of $1/N_f$ and $\omega$, our final results for the boson self energy for generic dispersion and form factor symmetry are

$$\boxed{\Pi(\boldsymbol{q} = 0, i\nu) \approx N_f C_0 + \begin{cases} 0, & \text{inversion-odd} \\ C_z |\nu|^{\frac{4-z}{z}}, & \text{inversion-even} \end{cases}.} \quad \text{(F.51)}$$

The above formula exactly agrees with Eq. (F.9). From here, it is simple to realize that an even stronger result (which we will later need for the conductivity) holds: Suppose we define a new function $\Pi_{VV}$ that has the same diagrammatic expansion as $\Pi$, but with external vertices $f(\boldsymbol{k}, \boldsymbol{q})$ replaced by some different $V(\boldsymbol{k}, \boldsymbol{q})$. After going through the same manipulations as before, we find that

$$\Pi_{VV}^{(\text{SE},1)} + \Pi_{VV}^{(\text{SE},2)} + \Pi_{VV}^{(\text{MT},1)} = 0\,, \tag{F.52}$$

for all $V$, generalizing Eq. (F.21). Similarly, we can relate $\Pi_{VV}^{(\text{MT},2)}, \Pi_{VV}^{(\text{AL},1)}, \Pi_{VV}^{(\text{AL},2)}$ to one loop integrals

$$\Pi_{VV}^{(\text{MT},2)} = \frac{1}{2\nu^2} \int_{\boldsymbol{q}',\nu'} D(\boldsymbol{q}', i\nu') D(\boldsymbol{q}', i\nu'+i\nu) \left[|\nu' + \nu| - |\nu'|\right] \frac{\gamma_{\hat{\boldsymbol{q}}'}}{|\boldsymbol{q}'|} \left[\pi_{VV}^{(\text{MT})}(\boldsymbol{q}', \nu' + \nu) - \pi_{VV}^{(\text{MT})}(\boldsymbol{q}', \nu')\right]\,, \tag{F.53}$$

$$\Pi_{VV}^{(\text{AL},1)} + \Pi_{VV}^{(\text{AL},2)} = -\frac{1}{2\nu^2} \int_{\boldsymbol{q}',\nu'} D(\boldsymbol{q}', i\nu') D(\boldsymbol{q}', i\nu' + i\nu) \left[\pi_{VV}^{(\text{AL})}(\boldsymbol{q}', \nu + \nu') - \pi_{VV}^{(\text{AL})}(\boldsymbol{q}', \nu')\right]^2\,, \tag{F.54}$$

where we have new definitions for the one-loop integrals

$$\pi_{VV}^{(\text{MT})}(\boldsymbol{q}', \nu') = \int_{\boldsymbol{k},\omega} [V(\boldsymbol{k} + \tfrac{\boldsymbol{q}'}{2}, \hat{\boldsymbol{q}}) - V(\boldsymbol{k} - \tfrac{\boldsymbol{q}'}{2}, \hat{\boldsymbol{q}})]^2 f(\boldsymbol{k}, \boldsymbol{q}')^2 G(\boldsymbol{k} - \tfrac{\boldsymbol{q}'}{2}, i\omega) G(\boldsymbol{k} + \tfrac{\boldsymbol{q}'}{2}, i\omega + i\nu')\,, \tag{F.55}$$

$$\pi_{VV}^{(\text{AL})}(\boldsymbol{q}', \nu') = \int_{\boldsymbol{k},\omega} \left[V(\boldsymbol{k} + \tfrac{\boldsymbol{q}'}{2}, \hat{\boldsymbol{q}}) - V(\boldsymbol{k} - \tfrac{\boldsymbol{q}'}{2}, \hat{\boldsymbol{q}})\right] f(\boldsymbol{k}, \boldsymbol{q}')^2 G(\boldsymbol{k} + \tfrac{\boldsymbol{q}'}{2}, i\omega + i\nu') G(\boldsymbol{k} - \tfrac{\boldsymbol{q}'}{2}, i\omega)\,. \tag{F.56}$$

When evaluating these integrals, we find that the inversion parity of $f$ doesn't play a role and the inversion parity of $V$ alone determines the final result. Hence, we arrive at the more general result

$$\boxed{\Pi_{VV}(\boldsymbol{q} = 0, i\nu) \approx N_f C_{0,V} + \begin{cases} 0\,, & V \text{ is inversion-odd} \\ C_{z,V} |\nu|^{\frac{4-z}{z}}\,, & V \text{ is inversion-even} \end{cases}}\,. \tag{F.57}$$

## F.7   From boson self energy to conductivity

Finally, we relate the boson self energy to the conductivity. For a general order parameter form factor $f(\boldsymbol{k}, \boldsymbol{q})$, it is useful to decompose the current operator at zero momentum $\boldsymbol{J}(t)$ in the following way (see Section 3.2) with implicit summation over repeated fermion flavor indices,

$$\boldsymbol{J}(t) = \boldsymbol{J}_0(t) + \frac{1}{\sqrt{N_f}} \boldsymbol{J_1}(t)\,, \tag{F.58}$$

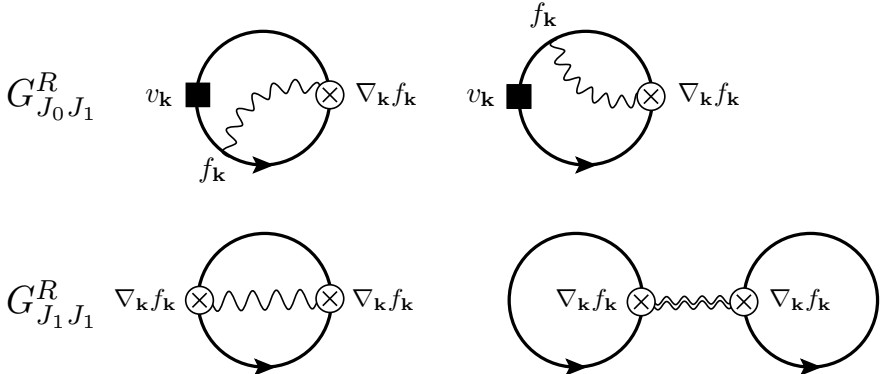

**Figure 7:** Additional diagrams that contribute to the conductivity but not the boson self energy. The solid square is a source for $J_0(\boldsymbol{q}=0,\Omega)$ and the cross is a source for $J_1(\boldsymbol{q}=0,\Omega)$. The vertex factors are marked explicitly for clarity. In the lower right diagram, the double wiggly line corresponds to a geometric sum of one-loop bubble diagrams that all contribute to the same order in $1/N_f$.

$$\boldsymbol{J}_0(t) = \int_{\boldsymbol{k}} \boldsymbol{v}(\boldsymbol{k})\psi_i^\dagger(\boldsymbol{k})\psi_i^\dagger(\boldsymbol{k})\,, \quad \boldsymbol{J}_1(t) = \int_{\boldsymbol{k},\boldsymbol{q}} [\partial_{\boldsymbol{k}} f(\boldsymbol{k},\boldsymbol{q})]\,\phi(\boldsymbol{q})\psi_i^\dagger(\boldsymbol{k}+\frac{\boldsymbol{q}}{2})\psi_i(\boldsymbol{k}-\frac{\boldsymbol{q}}{2})\,. \quad \text{(F.59)}$$

Using the standard Kubo formula, we can relate the conductivity to the current-current correlation function

$$\sigma^{ij}(\omega) = \frac{i}{\omega}\left[\mathfrak{D} - G_{J^i J^j}^R(\boldsymbol{q}=0,\omega)\right]\,. \quad \text{(F.60)}$$

The diamagnetic term $\mathfrak{D}$ contributes to the Drude weight but not to the incoherent conductivity. Therefore, we focus on $G_{J^i J^j}^R$. Using the decomposition of the current operator above, we can write

$$G_{J^i J^j}^R(\boldsymbol{q}=0,\omega) = G_{J_0 J_0}^R(\boldsymbol{q}=0,\omega) + \frac{2}{\sqrt{N_f}}G_{J_0 J_1}^R(\boldsymbol{q}=0,\omega) + \frac{1}{N_f}G_{J_1 J_1}^R(\boldsymbol{q}=0,\omega)\,. \quad \text{(F.61)}$$

The diagrams in $G_{J_1 J_1}^R(\boldsymbol{q}=0,\omega)$ and $G_{J_0 J_1}^R(\boldsymbol{q}=0,\omega)$ that contribute to the conductivity to leading two orders in the $1/N_f$ expansion are shown in Figure 7. By scaling arguments, one can demonstrate that the frequency dependent parts of $G_{J_0 J_1}^R(\boldsymbol{q}=0,\omega)$ and $G_{J_1 J_1}^R(\boldsymbol{q}=0,\omega)$ are both less singular than $\omega^{(4-z)/z}$ for all values of $2 < z \le 3$. Finally we evaluate $G_{J_0 J_0}^R(\boldsymbol{q}=0,\omega)$ to leading two orders in the $1/N_f$ expansion. The leading order term is $\mathcal{O}(N_f)$ and corresponds to a geometric series of one-loop bubbles that contribute to the Drude weight. The subleading $\mathcal{O}(1)$ term comes from diagrams that are structurally identical to Figure 6 but with external vertices $f(\boldsymbol{k},\boldsymbol{q})$ replaced by velocities $v_{\boldsymbol{k}} = \nabla_{\boldsymbol{k}}\epsilon(\boldsymbol{k})$. Since $v_{\boldsymbol{k}}$ is inversion-odd independent of the order parameter symmetry, we can simply apply Eq. (F.57). We therefore conclude that to leading two orders in the $1/N_f$ expansion, all diagrams contributing to the conductivity have a frequency scaling less singular than $\omega^{-2(z-2)/z}$. This leads to the conclusion in Eq. (F.10).

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
