# Peer review of "Loop current fluctuations and quantum critical transport"

_SciPost Physics_

## Round 1 · Referee Report · Anonymous (Referee 1) · 2022-10-31

Report

In the manuscript “Loop current fluctuations and quantum critical transport”, the authors extend their previous study on anomaly-assisted calculation of optical conductivity in a clean quantum critical system to a random large N theory. In their previous work, the authors claim that for the Hertz-Millis theory of clean non-Fermi liquids, the incoherent part of the optical conductivity vanishes for a special reason, leaving only the coherent Drude peak without fine tuning, and therefore other kinds of mechanism(s) are required to account for the existence of incoherent transport properties observed in many experiments. It is the goal of the current manuscript to find such models which have modified anomaly structure to admit finite incoherent optical conductivity. Here they make use of a Yukawa-SYK type large N theory and calculate the optical conductivity at large but finite N case, by means of memory matrix approach. In my opinion, this work, together with the previous one by the same authors Ref[19], have a strong claim against previous established knowledge, which may reshape our understanding of the non-Fermi liquid phase in strongly correlated electronic systems. Thus I recommend publication after my following questions have been addressed.

1. As the authors stated, there is a discrepancy between N>1 and N=1. Is this a sharp discontinuity? The condition for Eq.(1.5), the main result of this paper, is $\omega^{1/3}N\ll 1$ (take $z=3$ for example). So it seems that even when N is order 1 the incoherent part is still there. 
2. In Sec.3.2 the authors claim that the source of low energy anomaly is due to the necessity of taking bosons into account for calculating the current vertex, where the form factor plays an important role. Eq.(3.3) is obtained by extending $k$ to $k+A$ and expanding to linear order in $A$. But in cases when the form factor $g(k)$ does not contain linear in $k$ term, does this anomaly argument fail?
3. From Eq.(3.3) and (3.4), the current operator from the fermions are approximated by $v_F(\theta)\tilde{n}_\theta$. Will the conclusions be altered if keeping momentum dependence in $v_{F}$ in addition to the patch index? Because such complete k dependence in $v(k)$ is important in using Ward identity analysis.
4. Is it possible to perform some numerical calculations for the patch model at $N=1$ to justify the claim by the authors?
  • validity: -
  • significance: -
  • originality: -
  • clarity: -
  • formatting: -
  • grammar: -

Author:  Zhengyan Shi  on 2023-01-04  [id 3203]

(in reply to Report 1 on 2022-10-31)
Category:
answer to question
correction

Please see attached.

Attachment:

loopcurrent_response_ref1.pdf

---

## Round 1 · Referee Report · Anonymous (Referee 2) · 2022-11-1

Strengths

see report

Weaknesses

see report

Report

This paper continues the pursuit of an understanding of the conductivity of strange metals in terms of clean quantum critical points. Previous recent work by the same authors has established strong constraints on the frequency dependence of the conductivity in a popular class of models of non-Fermi liquids. In particular, they showed that in the standard clean Hertz-Millis fixed-point theory, the conductivity is exactly a delta function at zero frequency.

The goal of the present paper is to construct models (not necessarily realistic or robust) that exhibit a nonzero incoherent (finite-frequency) conductivity, as a proof of principle and for development of technique. The key idea is to introduce a number ($N$) of flavors of fermion species, and couple them to the critical boson by a random (but spatially-uniform) coupling tensor; because the boson now couples to a combination of fermion bilinears that is not a conserved density, the anomaly constraints are weakened.

The narrative of the paper negotiates many subtleties, generally arising from long-lived modes that overlap wth the current. In particular, the authors explicitly demonstrate that the large-$N$ and low-frequency limits do not commute. They derive this result in two (sort of) independent ways, and develop useful technology along the way.

The paper, though long and complicated, is clearly written, and an effort has been made to help the reader along.

-- I applaud the inclusion of the demonstration of self-averaging of the leading two orders of the large-$N$ expansion.

-- I applaud the honesty about the presence of many ($N^2$!) relevant perturbations.

-- In the summary above I spoke as if there were a single boson. Actually the bosons in the paper come with two indices $\phi_I^a$. The $a$ index is eventually dropped as not necessary. Is the $I$ index on the boson necessary?

-- at the top of page 13 there is an extra "the".

-- I didn't like the paragraph after equation (3.6), about the regulator-dependence of the expressions for various densities in terms of the fields. The reader has just been introduced to these quantities $n_\theta$ and $\tilde n_\theta$ and to immediately be told that the concrete expressions involving them are so mercurial is disheartening. It would be nice to rephrase this paragraph to first say more things that are robustly true about these quantities.

-- At various points there is a close parallel between the coupling to a loop current order parameter and to an abelian gauge field. I understand that this is a target for future work, but it might be nice to comment about where the analyses depart. (For example, does the discussion on page 16 apply to the gauge field case?)

-- The paper, as a sort of bonus, contains a reanalysis of the calculation of Ref 64, which regards the inclusion of $O(N_f)$-symmetric couplings of $N_f$ Fermi surfaces to a gauge field, including irrelevant couplings. As currently presented, this might be a bit of a distraction from the main focus of the paper. I think it is a nice thing to include, but perhaps more can be done to sequester it.

Requested changes

1- please think about the comments in the report.

  • validity: -
  • significance: -
  • originality: -
  • clarity: -
  • formatting: -
  • grammar: -

Author:  Zhengyan Shi  on 2023-01-04  [id 3204]

(in reply to Report 2 on 2022-11-01)
Category:
answer to question
correction

Please see attached.

Attachment:

loopcurrent_response_ref2.pdf

---

## Round 1 · Referee Report · Anonymous (Referee 3) · 2022-11-7

Report

The paper studies the optical conductivity of quantum critical metals by leveraging non-perturbative constraints imposed by the 't Hooft anomaly under the assumption that the emergent symmetry includes the fermion number conservation within each patch. It is argued that a large N model with random coupling in the flavor space and the original N=1 model belong to different universality classes because of different anomaly structures. Consequently, the large-N model can have an incoherent conductivity while the original model only has the Drude peak. This is an interesting work that sheds new light on transport in strongly coupled systems that are usually hard to understand in a controlled way. Before publication, however, it will be nice to include a discussion on the possible effects of large-angle scatterings. Generically, the low-energy theory can include the four-fermion coupling that decays slowly in momentum, thus evading the anomaly constraint. Even if the four-fermion coupling is `irrelevant' under the scaling, its effect may not be small because the effective number of patches connected by large-angle scatterings grows as the low-energy limit is taken. Then, is it, in principle, possible that the four-fermion coupling generates an incoherent conductivity for the N=1 model that is comparable to that of the large N model, or is the contribution of the four-fermion coupling expected to be generically sub-leading?
  • validity: -
  • significance: -
  • originality: -
  • clarity: -
  • formatting: -
  • grammar: -

Author:  Zhengyan Shi  on 2023-01-04  [id 3205]

(in reply to Report 3 on 2022-11-07)
Category:
answer to question

We thank the referee for raising this important subtlety. In this paper and the previous one, [SciPost Phys. 13 (2022) 102], we are mainly concerned with the conductivity arising from the fixed point theory, since this is what has the potential to exibhit the $\omega/T$ scaling form seen in materials. It is true that one must be careful to define the "fixed point theory" properly using a suitable IR limit. As the referee points out, since the number of patches generally approaches infinity as one takes the IR limit, it is possible that terms which are formally irrelevant (such as large-angle scattering) can have finite contribution in the "fixed point theory". In selecting which terms we believe are part of the "fixed point theory", we have been motivated by the conventional wisdom among practitioners, which admittedly is not based on a rigorous calculation but instead on tree level scaling in each patch. Certainly this is something that should warrant further exploration in the future, although we do not expect such irrelevant operators to dramatically affect the most singular contributions to transport. In any case, our work still demonstrates the distinctions between the $N=1$ and random-flavor large-$N$ regimes in the sense that the behavior when keeping the same set of terms in the Lagrangian ends up being qualitatively different.

---

## Editorial Decision

resubmitted